# The NDNF-like factor Nord is a Hedgehog-induced extracellular BMP modulator that regulates *Drosophila* wing patterning and growth

Shu Yang[1†], Xuefeng Wu[1†], Euphrosyne I Daoutidou[2], Ya Zhang[1], MaryJane Shimell[2], Kun-Han Chuang[1], Aidan J Peterson[2], Michael B O'Connor[2], Xiaoyan Zheng[1]*

[1]Department of Anatomy and Cell Biology and the GW Cancer Center, George Washington University School of Medicine and Health Sciences, Washington, United States; [2]Department of Genetics, Cell Biology & Development and the Developmental Biology Center, University of Minnesota, Minneapolis, United States

*For correspondence:
xzheng@gwu.edu

†These authors contributed equally to this work

Competing interest: The authors declare that no competing interests exist.

**Abstract** Hedgehog (Hh) and Bone Morphogenetic Proteins (BMPs) pattern the developing *Drosophila* wing by functioning as short- and long-range morphogens, respectively. Here, we show that a previously unknown Hh-dependent mechanism fine-tunes the activity of BMPs. Through genome-wide expression profiling of the *Drosophila* wing imaginal discs, we identify *nord* as a novel target gene of the Hh signaling pathway. Nord is related to the vertebrate Neuron-Derived Neurotrophic Factor (NDNF) involved in congenital hypogonadotropic hypogonadism and several types of cancer. Loss- and gain-of-function analyses implicate Nord in the regulation of wing growth and proper crossvein patterning. At the molecular level, we present biochemical evidence that Nord is a secreted BMP-binding protein and localizes to the extracellular matrix. Nord binds to Decapentaplegic (Dpp) or the heterodimer Dpp-Glass-bottom boat (Gbb) to modulate their release and activity. Furthermore, we demonstrate that Nord is a dosage-dependent BMP modulator, where low levels of Nord promote and high levels inhibit BMP signaling. Taken together, we propose that Hh-induced Nord expression fine-tunes both the range and strength of BMP signaling in the developing *Drosophila* wing.

## Editor's evaluation

This manuscript is of broad interest to readers in the field of Bone Morphogenetic Proteins (BMPs) and *Drosophila* wing development. The identification of a Nord as an indirect regulator of BMP diffusion is a new contribution to our understanding of regulation of BMP function. A combination of genetics, transcriptomics, and biochemistry provide support for many of the conclusions in the paper.

## Introduction

Morphogens are conserved, secreted signaling molecules that pattern organs and tissues by eliciting graded responses from cells surrounding a localized source (*Lawrence and Struhl, 1996*; *Wolpert, 1969*). Secreted signaling proteins of the Epidermal Growth Factor (EGF), Fibroblast Growth Factor (FGF), Hedgehog (Hh), Notch, Transforming Growth Factor β (TGF-β)/Bone Morphogenetic Protein (BMP), and Wnt/Wingless (Wg) families have been shown to act as morphogens either at a short

range or over a long distance. How these morphogens and their corresponding signaling pathways coordinate to control patterning and growth in various developmental systems is a central question in developmental biology.

The developing appendages of *Drosophila* provide a valuable model system to genetically and molecularly identify morphogens and members of their corresponding signaling pathways. They have also assisted in revealing the fundamental features and interplay between different morphogens during pattern formation. The *Drosophila* Hh and BMP signals function as short- and long-range morphogens, respectively, and together organize patterning along the anteroposterior (A/P) axis of the wing imaginal discs (*Lecuit et al., 1996*; *Mullor et al., 1997*; *Nellen et al., 1996*; *Strigini and Cohen, 1997*; *Zecca et al., 1995*).

In the case of Hh signaling, the quiescent state of the pathway is maintained by the Hh receptor Patched (Ptc)-dependent inhibition of Smoothened (Smo) (*Ingham et al., 2011*). This inhibition is lifted by Hh binding to Ptc and its co-receptors (*Allen et al., 2011*; *Izzi et al., 2011*; *McLellan et al., 2008*; *Zheng et al., 2010*), which releases Smo from repression and thus activates intracellular signal transduction (*Beachy et al., 2010*; *Lum and Beachy, 2004*; *Varjosalo and Taipale, 2008*). In the wing discs, Hh is secreted in the posterior (P) compartment and spreads toward the anterior (A) compartment (*Basler and Struhl, 1994*; *Capdevila et al., 1994*; *Tabata and Kornberg, 1994*). Hh signaling does not occur in P compartment cells because they do not express critical components of the Hh pathway, such as the major transcriptional effector Ci (*Eaton and Kornberg, 1990*). Conversely, A compartment cells can receive and respond to Hh but are unable to produce Hh. Hh synthesized by P cells diffuses into the A compartment adjacent to the A/P boundary, where it is sequestered by the Ptc-Ihog/Boi receptor complex (*Chen and Struhl, 1996*; *Zheng et al., 2010*), thus forming a short-range gradient that activates the localized expression ofseveral target genes, including *ptc* and *decapentaplegic* (*dpp*) (*Basler and Struhl, 1994*; *Capdevila et al., 1994*; *Chen and Struhl, 1996*; *Ingham et al., 1991*; *Tabata and Kornberg, 1994*).

Dpp is the homolog of vertebrate TGF-β superfamily ligands BMP2/4, which binds to the BMP type I receptor thickveins (Tkv) or Saxophone (Sax) and type II receptor Punt (*Parker et al., 2004*). Upon ligand binding, Tkv or Sax phosphorylates the transcription factor Mad (referred to as pMad; an indication of active Dpp signal transduction). Phosphorylated Mad then forms a trimeric complex with the co-Smad Medea and is translocated to the nucleus where it regulates downstream target genes (*Moustakas and Heldin, 2009*). In contrast to the Hh signal, Dpp diffuses into both the A and P compartments and acts as a long-range morphogen (*Brook et al., 1996*; *Lawrence and Struhl, 1996*). Dpp has multiple roles both in determining the wing size by promoting growth and epithelial morphogenesis during larval stages (*Gibson and Perrimon, 2005*; *Schwank et al., 2008*; *Shen and Dahmann, 2005*), and in patterning the longitudinal veins (LV) as well as the anterior crossvein (ACV) and posterior crossvein (PCV) during pupal development (*de Celis et al., 1996*; *Ralston and Blair, 2005*; *Serpe et al., 2005*; *Shimmi et al., 2005*; *Sotillos and De Celis, 2005*; *Vilmos et al., 2005*).

The diverse signaling outputs of Dpp are influenced by the formation of ligand heterodimers and by their binding to various extracellular factors. In *Drosophila*, two other characterized BMP family ligands, the BMP5/6/7/8 homolog Glass-bottom boat (Gbb), and the more distantly related Screw (Scw) (*Arora et al., 1994*; *Padgett et al., 1987*; *Wharton et al., 1991*), can form heterodimers with Dpp to augment the level and increase the range of BMP signaling in different cells and tissues. Other secreted or membrane-binding BMP-binding proteins, including Short gastrulation (Sog) (*Ralston and Blair, 2005*; *Serpe et al., 2005*), Twisted gastrulation (Tsg), Crossveinless (Cv) (*Shimmi et al., 2005*; *Vilmos et al., 2005*), Crossveinless-2 (Cv-2) (*Conley et al., 2000*; *Serpe et al., 2008*), Larval translucida (ltl) (*Szuperák et al., 2011*), Pentagone (Pent) (*Vuilleumier et al., 2010*), Kekkon5 (Kek5) (*Evans et al., 2009*), Dally and Dally-like protein (Dlp) (*Akiyama et al., 2008*; *Belenkaya et al., 2004*), have been identified as modulators that enhance or inhibit BMP signaling. Additionally, Tkv, the main type I Dpp receptor in *Drosophila*, critically affects Dpp tissue distribution through ligand trapping and internalization, and thus globally shapes the BMP activity gradient (*Crickmore and Mann, 2006*; *Lecuit and Cohen, 1998*; *Tanimoto et al., 2000*). Prominent among the many determinants that impact proper establishment and maintenance of the Dpp activity gradient is the action of the Hh signal, which simultaneously induces Dpp expression and lowers responsiveness to Dpp in the same cells at the A/P border by repressing transcription of the *tkv* receptor gene (*Tanimoto et al., 2000*). The coordinated activation and attenuation of BMP signaling activity by

Hh illustrates the complex regulatory interactions of different morphogens in pattern and growth in developing systems.

Here, we show that the activity of BMPs is further fine-tuned by a previously unknown Hh-dependent mechanism involving induction of *nord,* which encodes a Neuron-Derived Neurotrophic Factor (NDNF)-like factor. Absence of *nord* results in reduced medial BMP signaling near the source of Dpp in the larval wing imaginal discs but increases long-range BMP activity surrounding the primordial PCV in the pupal wing. Thus, adult wings from *nord* mutant flies exhibit seemingly opposite phenotypes of reduced wing size and ectopic PCV. Expression and co-immunoprecipitation studies in cultured S2 cells demonstrate that Nord is secreted, associates with the extracellular matrix, and binds to Dpp or the heterodimer of Dpp-Gbb. We further show that Nord is a dosage-dependent BMP modulator, where low levels of Nord promote and high levels inhibit BMP signaling. Taken together, we propose that Hh-induced Nord expression modulates the distribution and level of BMP signaling in the developing *Drosophila* larval and pupal wing to obtain proper size and crossvein patterning.

## Results

### Genome-wide expression profiling in *Drosophila* wing imaginal discs identifies *nord* as a novel target gene of the Hh signaling pathway

The imaginal discs of *Drosophila melanogaster*, where most known Hh signaling target genes are expressed with a restricted pattern (*Strigini and Cohen, 1997*), offer an accessible model system for identifying novel targets of the Hh signaling pathway. In wing discs, cells near the A/P compartment boundary (B: *ptc+*) receive the highest level of Hh stimulation while A cells (A: *hh-*), located further from the border, receive lower levels of stimulation. In contrast, P cells (P: *hh+*) do not respond to Hh due to lack of the receptor Ptc and the transcription factor Ci (*Eaton and Kornberg, 1990*; *Figure 1A*, *Figure 1—figure supplement 1*). To identify target genes whose expression is controlled, directly or indirectly, by Hh signaling activity, we performed a systematic comparison of gene expression profiles among the three cell types. After genetic labeling, wing discs were dissected, cells dissociated, and then sorted using fluorescence-activated cell sorting (FACS). RNA was extracted from the sorted cell populations and subjected to microarray analysis (*Figure 1A*, *Figure 1—figure supplement 1*). Previously, by comparing genes differentially expressed in the A/P boundary adjacent cells (B: *ptc+*) and P cells (P: *hh+*), we identified an unknown Hh pathway target dTRAF1/TRAF4 and established that the Hh signal mediates JNK activity by regulating the expression of dTRAF1/TRAF4 in developmental organ size control (*Willsey et al., 2016*; *Figure 1B*). Here, we modified the gene expression analysis method by including additional transcriptome comparisons between the A/P boundary adjacent cells (B: *ptc+*) and A cells (A: *hh-*), and between A cells (A: *hh-*) and P cells (P: *hh+*). Genes whose expression is not only higher in A cells than P cells ($Fold_{A/P} > 1.2$), but also higher in the A/P boundary adjacent cells than general A cells ($Fold_{B/A} > 1.5$) were selected as potential Hh-induced target genes (*Figure 1B and C*). Hh-responsive genes known to be differentially expressed in the wing discs were found, including *ptc* and *dpp*, (*Figure 1B–D*, *Supplementary file 1*). We then focused on *nord* (FlyBase, CG30418), one of the top-ranking A/P boundary-enriched genes, with no previously characterized expression pattern or functional analysis (*Figure 1B–D*). We first verified the differential expression of *nord* across the *Drosophila* wing imaginal discs via quantitative reverse transcription PCR using RNA isolated from FACS sorted A, B, and P cells (*Figure 1E*). We also performed *in situ* hybridization to localize *nord* transcripts in the wing imaginal discs (*Figure 1F*). Like the known Hh signaling target gene *ptc* (*Figure 1E and F*), *nord* transcripts were absent from the P compartment and primarily detected in the A cells adjacent to the A/P compartment boundary (*Figure 1E and F*). Of note, unlike *ptc* and most other Hh pathway target genes, *nord* expression was not detected in the central wing pouch (*Figure 1F*). Collectively, these results suggested that *nord* is a potential target gene of the *Drosophila* Hh signaling pathway.

### Hh signaling regulates *nord* expression in the wing discs

To further analyze the expression pattern and investigate the function of Nord, we identified and characterized two Minos-Mediated Integration Cassette (MiMIC) lines from the *Drosophila* Gene Disruption Project (GDP) collection (*Venken et al., 2011*), one a gene-trap *nord* allele *Mi{MIC}nord*[MI06414] and the second a protein-trap *nord* allele *Mi{PT-GFSTF.2}nord*[MI06414-GFSTF.2]. We also generated an additional

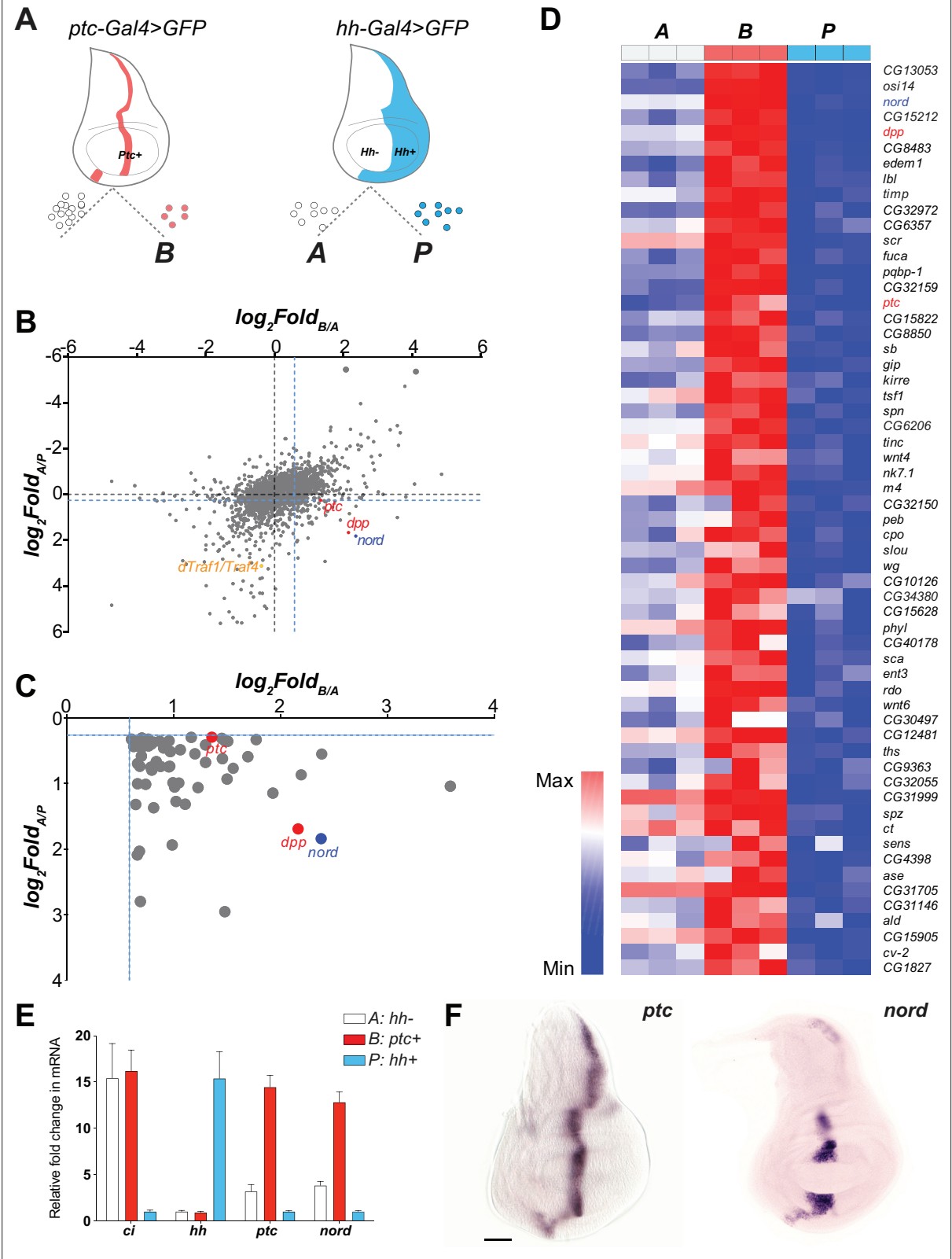

**Figure 1.** *nord* is a novel target gene of the *Drosophila* Hedgehog (Hh) signaling pathway. (**A**) Schematic diagram of *Drosophila* wing imaginal disc: posterior compartment (P: *hh+*), anterior compartment (A: *hh-*), and anterior compartment cells adjacent to the A/P boundary (B: *ptc+*). A, P, and B cells from wing imaginal discs of third instar larvae carrying *hh-Gal4* or *ptc-Gal4*-driven *UAS-mCD8-GFP* were dissociated and sorted by fluorescence-activated cell sorting (FACS). RNA was isolated and hybridized to microarrays. Differentially expressed genes were identified. (**B**) Dot plot shows all

*Figure 1 continued on next page*

*Figure 1 continued*

14,448 annotated probe sets. Each dot indicates one probe set. The x-axis represents the log$_2$ fold change of each gene in B vs. A cells, and the y-axis represents the log$_2$ fold change of each gene in A vs. P cells. The blue dash lines indicated the threshold used to select differentially expressed genes in each group. Colored dots indicate known or novel Hh pathway targets *dTraf1/Traf4* (orange), *dpp* (red), *ptc* (red), and *nord* (blue) in the graph. (**C**) Zoomed view of the bottom-right corner in panel (**B**) to show 59 differentially expressed genes, whose expression is significantly increased in B (*ptc+*) cells but decreased in P (*hh+*) cells compared with A (*hh-*) cells. (**D**) Heatmap shows the expression level of the 59 top-ranking differentially expressed genes in A, B, and P cells. The fold change in B vs. A cells was used to rank the order. (**E**) Fold changes of *ci, hh, ptc,* and *nord* mRNA expression, measured by quantitative reverse transcription PCR and normalized by the expression of the housekeeping gene *pkg*, in FACS-sorted A, B, or P cells. (**F**) *In situ* hybridization of *ptc* and *nord* in the third instar larval *Drosophila* wing discs. Scale bar, 50 µm.

The online version of this article includes the following figure supplement(s) for figure 1:

**Figure supplement 1.** Workflow to isolate RNA from different cell populations sorted from the *Drosophila* wing imaginal discs.

protein-trap *nord* allele *Mi{PT-RFPHA.2}nord*$^{MI06414-RFPHA.2}$. The *nord* gene-trap allele *Mi{MIC}nord*$^{MI06414}$ contains a MiMIC consisting of a splice acceptor site followed by stop codons in all three reading frames. This transposon was inserted into the first coding intron of *nord* and thus interrupted transcription and translation of *nord* (***Figure 2—figure supplement 1***). This *nord* gene-trap allele was used in the functional analysis of Nord. The two protein-trap *nord* alleles *Mi{PT-GFSTF.2}nord*$^{MI06414-GFSTF.2}$ and *Mi{PT-RFPHA.2}nord*$^{MI06414-RFPHA.2}$ were derived from the *Mi{MIC}nord*$^{MI06414}$ line by replacing the original gene-trap cassette with either an EGFP-FlAsH-StrepII-3xFLAG (GFSTF) or a TagRFP-T-3xHA (RFPHA) protein-trap cassette using Recombinase-Mediated Cassette Exchange strategies (***Nagarkar-Jaiswal et al., 2015***; ***Figure 2A***). In these *nord* protein-trap alleles, the GFSTF or RFPHA tag was inserted in the appropriate orientation and reading frame of *nord* between amino acids 103R and 104F (***Figure 2—figure supplement 2***), which permitted visualization of the Nord protein localization *in vivo*. Immunostaining of wing imaginal discs from the *nord* protein-trap alleles showed that Nord is expressed along the A/P boundary within the hinge and notum but avoids the central wing pouch (***Figure 2B and C***), identical to the pattern of endogenous *nord* transcripts revealed by *in situ* hybridization (***Figure 1F***).

The selective upregulation of *nord* in the wing imaginal disc A cells adjacent to the A/P boundary indicated that *nord* expression may be controlled by Hh signaling activity.

We next examined this issue directly by following endogenous Nord expression in the protein-trap *nord* allele *Mi{PT-GFSTF.2}nord*$^{MI06414-GFSTF.2}$ (hereafter referred to as *nord-GFP*). We used the *heat shock* (*hs*)-*Gal4* driver (***Halfon et al., 1997***) either to activate Hh signaling by ectopically expressing *UAS-Hh* or inactivate Hh signaling activity by overexpressing *UAS-Ptc*. When compared to the wing discs carrying *hs-Gal4* alone, Nord-GFP expression was expanded anteriorly following ectopic Hh expression by *hs-Gal4* driving *UAS-Hh* (***Figure 2—figure supplement 3A and B***). Conversely, overexpression of Ptc, which keeps the Hh pathway silenced in the wing imaginal discs, dramatically reduced the Nord-GFP expression domain along the A/P boundary (***Figure 2—figure supplement 3A and C***).

To investigate whether Hh signaling activity can cell-autonomously induce the expression pattern of *nord*, we performed clonal analyses using the flip-out technique to ectopically activate the Hh signaling pathway (***Ito et al., 1997***). The expression of endogenous Nord was examined in the third instar wing discs carrying the protein-trap *nord* allele *Mi{PT-RFPHA.2}nord*$^{MI06414-RFPHA.2}$ (hereafter referred to as *nord-RFP*) and flip-out clones expressing *UAS-mCD8-GFP* alone or in combination with a constitutively active form of Smo (*UAS-SmoGlu*) (***Zhang et al., 2004***; ***Figure 2C and D***). We found that activation of Hh signaling via the constitutively active *SmoGlu* autonomously induced ectopic *nord* expression in the A compartment in clones that flanked the wing pouch (***Figure 2D and D'***). Consistent with the observation that endogenous *nord* expression is restricted to the A compartment and mostly absent from the center of the wing pouch (***Figures 1F, 2B and C***, ***Figure 2—figure supplement 3A***), we noted that little or no ectopic Nord-RFP was detected in *SmoGlu*-expressing flip-out clones located in the central wing pouch region or those located in the P compartment, respectively (***Figure 2D" and D'"***). Furthermore, ectopic Hh, which is sufficient to activate the expression of the high-threshold target *ptc,* failed to induce *nord* expression in the center of the wing pouch (***Figure 2—figure supplement 3B***), indicating another mechanism excludes *nord* expression from this region. Nevertheless, these data demonstrated that *nord* expression in the hinge, the notum, and the edge of the wing pouch is regulated by Hh signaling activity. We thus identified *nord* as a novel target gene of the *Drosophila* Hh signaling pathway.

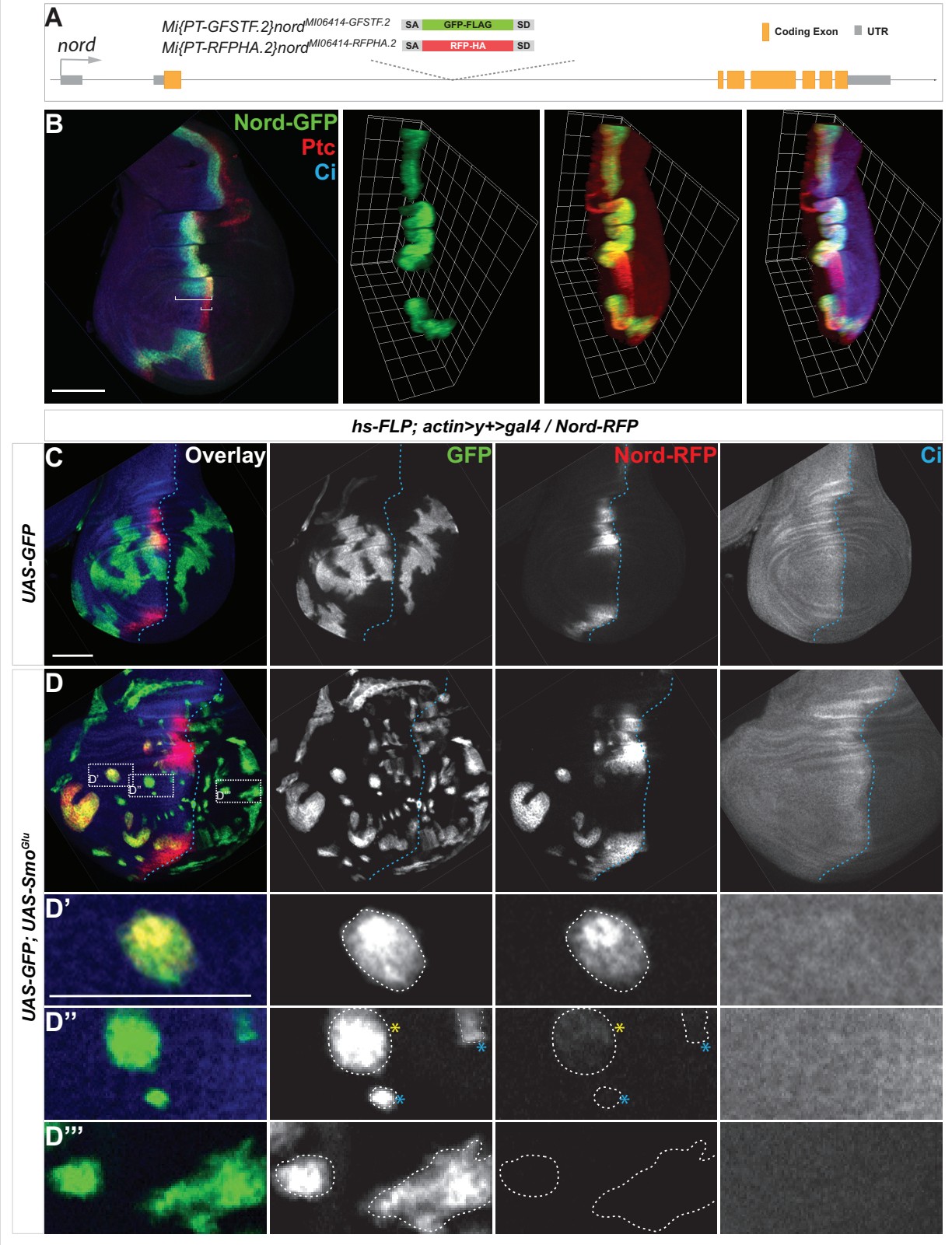

**Figure 2.** Hedgehog (Hh) signaling regulates *nord* expression in the *Drosophila* wing imaginal disc. (**A**) Schematic diagram of the wild-type *nord* locus and the protein-trap alleles of *nord*. The EGFP-FlAsH-StrepII-3xFLAG (GFSTF) or TagRFP-T-3xHA (RFPHA) tag was inserted in the appropriate orientation and reading frame of *nord*, which permitted visualization of the Nord protein localization *in vivo*. (**B**) Wing imaginal discs from late third instar larvae carrying *nord-GFP* (*Mi{PT-GFSTF.2}nord^{MI06414-GFSTF.2}/+*) were immunostained for GFP (green), Ptc (red), and Ci (blue). Maximum intensity

*Figure 2 continued on next page*

*Figure 2 continued*

z-projection and 3D reconstruction from a confocal image stack show *nord* expression in a representative wing imaginal disc. White brackets indicate the expression range of Ptc or Nord-GFP. (**C, D**) Wing imaginal discs from late third instar larvae carrying *nord-RFP* (*Mi{PT-RFPHA.2}nord*^MI06414-RFPHA.2^/+) and flip-out clones expressing the indicated *UAS*-transgenes were immunostained for HA (Nord-RFP, red), GFP (flip-out clones, green), and Ci (A compartment, blue). (**D'–D"'**) Zoomed view of the indicated area from panel (**D**). Note that ectopic *nord-RFP* is induced in *UAS-SmoGlu*-expressing clones located in the A compartment flanking the wing pouch (**D'**), but not in the P compartment (**D"'**). In the central wing pouch (**D"**), little (yellow star) or none (blue star) ectopic Nord-RFP was detected in *SmoGlu*-expressing flip-out clones. Dashed white lines indicate the clone boundary; dashed blue lines indicate the A/P compartment boundary, which is determined by the expression of endogenous Ci. Scale bar, 50 μm.

The online version of this article includes the following figure supplement(s) for figure 2:

**Figure supplement 1.** Schematic diagram of the wild-type *nord* locus, the gene-trap, and the protein-trap alleles of *nord*.

**Figure supplement 2.** Alignment of Nord, Nord-GFP, and Nord-RFP encoded by transcripts from the wild-type or *nord* protein-trap alleles.

**Figure supplement 3.** Hedgehog (Hh) signaling regulates *nord* expression in the *Drosophila* wing disc.

## Nord belongs to an evolutionarily conserved family of secreted proteins

The *Drosophila nord* gene is associated with only one protein-coding transcript and one polypeptide containing 587 amino acids. It has a single homolog in the mouse genome, called NDNF (*Kuang et al., 2010*). Nord and NDNF belong to a family of evolutionarily conserved secreted proteins with a predicted signal peptide followed by one or two fibronectin type III-like repeats (FN3) and a domain of unknown function (pfam10179: DUF2369). Analyses of the genome and EST sequences from various organisms suggest that nearly all bilaterian animals have either single or multiple orthologous genes for Nord/Ndnf (*Figure 3A*). We cloned the full-length cDNAs from *Drosophila* Nord and NDNF from several different species into a mammalian expression vector and found that both Nord and various NDNF proteins were secreted into the medium after transient transfection into HEK 293T cells (*Figure 3B*). We immunostained unpermeabilized HEK 293T cells that were co-transfected with plasmids expressing Myc-tagged Nord or various NDNFs together with cytoplasmic-localized GFP proteins and detected a significant amount of Nord/NDNF proteins on the surface of the transfected cells (marked by cytoplasmic GFP expression), as well as the surrounding extracellular matrix within several cell diameters (*Figure 3C*, arrows). Consistent with the observations in cultured cells, when wing imaginal discs from third instar larvae carrying *nord-GFP* were immunostained in the absence of detergent, secreted Nord-GFP was noticed both flanking the A/P boundary and throughout the wing disc albeit at a much lower level (*Figure 3D*). Furthermore, when flip-out clones expressing an HA-tagged Nord were induced in the third instar wing imaginal discs, secreted Nord was detected outside of flip-out clones (*Figure 3E*). Together, these data demonstrated that Nord and its homolog NDNF belong to a family of secreted proteins, which likely exist in two spatially distinct pools: diffusible Nord/NDNF proteins that can reach a longer distance and membrane/matrix-associated Nord/NDNF proteins near the source cells.

## Nord is required for proper growth and crossvein patterning of the *Drosophila* wing

To investigate the function of Nord, we generated and characterized several *nord* loss-of-function alleles by using different genetic strategies. We first analyzed the phenotypes of two *nord* mutant alleles, *nord*^3D^ and *nord*^22A^, created by the CRISPR/Cas9 strategy. Both alleles carry short deletions in the fourth coding exon that cause frameshifts and premature stop codons within the DUF2369 domain of Nord (*Figure 4A*, *Figure 4—figure supplement 1*). Animals homozygous or trans-heterozygous for *nord*^22A^ and *nord*^3D^ are viable and fertile. As homozygous or trans-heterozygotes, they show no obvious defects in the shape of the wing blade. However, a consistent and significant decrease in wing size was observed in both male and female flies (*Figure 4—figure supplement 2*). We next characterized the *nord* gene-trap allele *Mi{MIC}nord*^MI06414^ (hereafter referred to as *nord*^MI06414^) from the GDP collection (*Venken et al., 2011*), in which the transcription and translation of *nord* are interrupted after the first coding exon, resulting in a predicted truncated peptide containing only the first 103 amino acids without the conserved DUF2369 and FN3 domains (*Figure 2—figure supplement 1*, *Figure 4A*, *Figure 4—figure supplement 1*). Similar to the CRISPR-derived *nord* mutant alleles, the

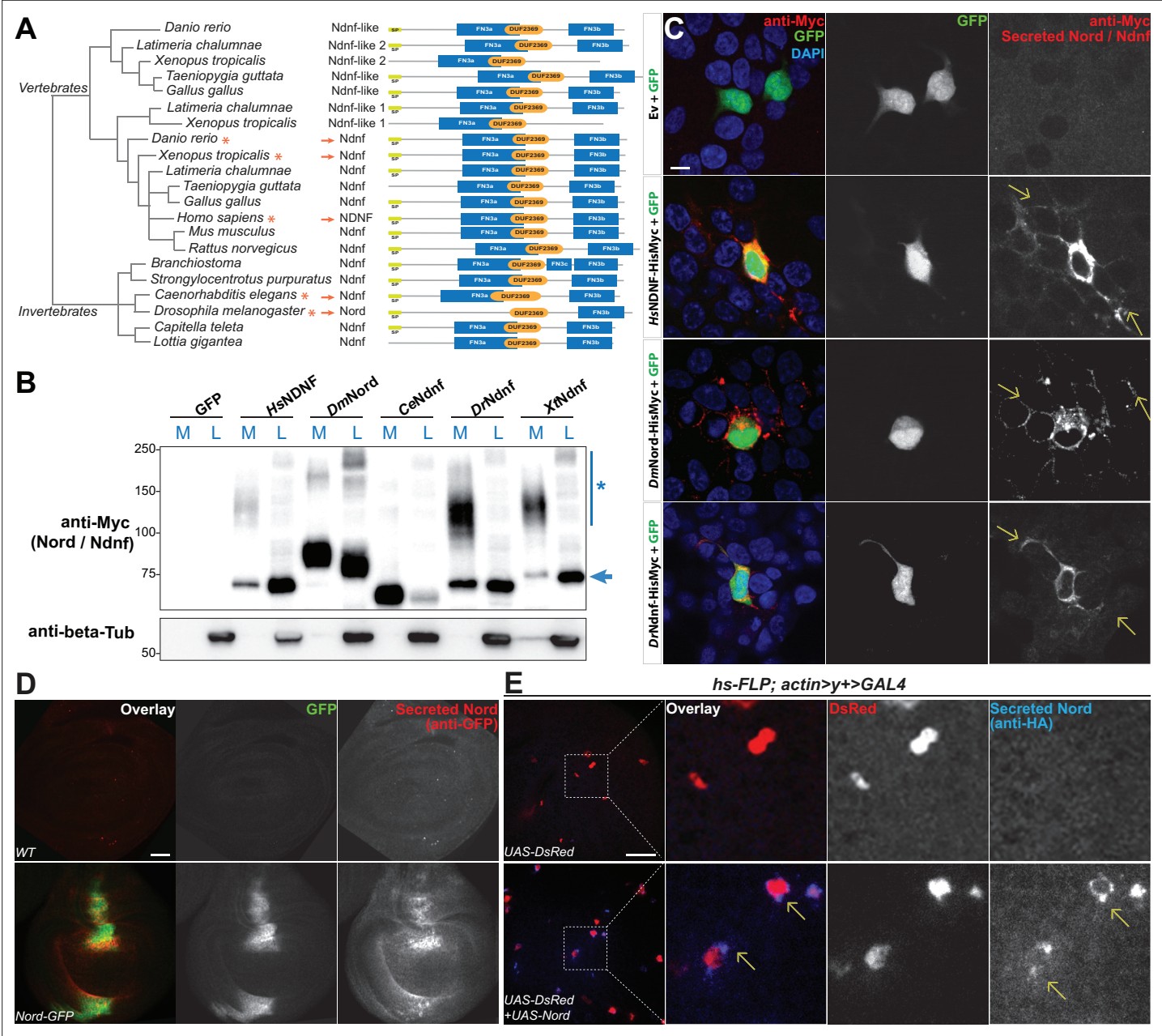

**Figure 3.** Nord belongs to a family of secreted proteins. (**A**) Phylogenetic analysis of Nord homologs from different species. The phylogenetic analysis of Nord homologs was performed with eggNOG (http://eggnogdb.embl.de/#/app/home). The phylogenetic tree is shown with domains predicted using Pfam (https://www.ebi.ac.uk/interpro/). The protein diagrams were then drawn proportional in length to the number of residues. Red stars (*) and red arrows (→) indicate the Ndnf/Nord proteins selected for further analysis. SP, signal peptide; FN3, fibronectin type 3 domain; DUF2369, domain of unknown function 2369. (**B**) Western blot analysis of HisMyc-tagged Nord or Ndnf protein in medium (M) and cell lysate (L) from transiently expressed in HEK-293 cells. Smeared band (*) indicates a portion of slow migrating Ndnf/Nord protein that may undergo extensive post-translational modification. Blue arrow (→) indicates Ndnf/Nord proteins that migrate as their predicted size. *Hs, Homo sapiens; Dm, Drosophila melanogaster; Ce, Caenorhabditis elegans; Dr, Danio rerio; Xt, Xenopus tropicalis.* (**C**) HEK-293 cells co-transfected with GFP and HisMyc-tagged Nord or Ndnf followed by cell-surface staining with anti-Myc antibody to reveal secreted Nord or Ndnf protein (yellow arrows). Note that secreted Nord-Myc and Ndnf-Myc (anti-Myc, red) were also detected several cell diameters away from the expressing cells (GFP, green). Scale bar, 10 μm. (**D**) Wing imaginal discs from third instar WT or *nord-GFP (Mi{PT-GFSTF.2}nord^{MI06414-GFSTF.2}/+)* larvae were immunostained for GFP (red) without detergent treatment to label the secreted pool of Nord-GFP proteins. The total Nord-GFP proteins were detected by fluorescence from the GFP tag (green). Scale bar, 20 μm. (**E**) Wing imaginal discs from third instar larvae carrying flip-out clones expressing *UAS-DsRed* alone or in combination with *UAS-Nord*. The discs were immunostained in the absence of detergent to label the secreted pool of HA-tagged *UAS-Nord* (anti-HA, blue). Flip-out clones were detected by fluorescence from the *UAS-DsRed* transgene (red). Scale bar, 20 μm.

*Figure 3 continued on next page*

*Figure 3 continued*

The online version of this article includes the following source data for figure 3:

**Source data 1.** Uncropped Western blot for *Figure 3*.

*nord^{MI06414}* homozygous flies are viable and fertile, and both male and female *nord* mutants showed a significant wing size reduction when compared to their wild-type counterpart (*Figure 4C*).

Along with the growth defects in the wing, we also noted that 61% of females (59/97) and 21% (17/82) of males of the *nord^{22A}* allele showed ectopic venation in the vicinity of the PCV of one or both wings (*Figure 4D and E*). The ectopic vein tissue was also found emanating from the PCV in the wing of adult flies homozygous for the *nord^{MI06414}* allele. Like *nord^{22A}*, the ectopic venation displayed higher penetrance in females than in males, with 86% of mutant females and 36% of mutant males showing additional vein material at one or both wings (*Figure 4D and E*). This ectopic venation phenotype was not observed in *nord^{3D}* homozygotes.

To further address whether the ectopic venation phenotype is caused by loss of *nord*, we performed genetic complementation tests between *nord^{MI06414}*, *nord^{3D}*, and *nord^{22A}* with various deficiency lines covering the *nord* locus (*Figure 4—figure supplement 3*). All available deficiency lines in which the *nord* locus was entirely removed did not rescue the ectopic PCV phenotype of *nord^{M106414}*, whereas adjacent deficiency lines with an intact *nord* locus fully rescued the crossvein defects in the *nord^{MI06414}* adult wings (*Figure 4D and E*, *Supplementary file 2*). We next tested *Df(2R)BSC155 in trans* to either *nord^{22A}* and *nord^{3D}* and once again found that *nord^{22A}* exhibited high-frequency ectopic venation (*Figure 4D and E*; 69% females and 26% males), while *nord^{3D}* showed a modest interaction (14% females and <1% males). Similar to the homozygotes, both *nord^{22A}* and *nord^{3D} in trans* to *Df(2R)BSC155* also produced smaller wings compared to controls (*Figure 4—figure supplement 4*). To examine whether this size difference was the result of reduced cell proliferation or a smaller cell size, we examined wing trichome density as a proxy for cell size (*Brummel et al., 1999*; *Dobzhansky, 1929*; *Martín-Castellanos and Edgar, 2002*). We found no difference for *nord^{MI06414}/Df(2R)BSC155* or *nord^{22A}/Df(2R)BSC155* compared to heterozygous (*nord^{MI06414}/+* or *nord^{22A}/+*) controls (*Figure 4—figure supplements 5 and 6*), suggesting that the alteration in wing size is caused by reduced cell proliferation. Together, these experiments indicate that Nord is required in wing imaginal discs for both proper growth and crossvein patterning and that the *nord^{3D}* allele may retain some function.

## Spatial-temporal overlapping expression of *nord* and *dpp* in the developing *Drosophila* wing

Both wing growth and crossvein patterning require precisely controlled BMP signaling activity (*Affolter and Basler, 2007*; *De Celis, 2003*). During wing development, BMP signaling activity is an output of the combined action of two BMP ligands, the *Drosophila* BMP2/4 homolog Dpp and the BMP5/6/7/8 homolog Gbb (*Bangi and Wharton, 2006*). Gbb is broadly and uniformly expressed in the larval and pupal wing, while Dpp, a well-known target of Hh signaling, is expressed in a stripe of cells in the anterior compartment along the A/P compartmental boundary of the larval wing imaginal disc. From this source, Dpp protein is thought to spread and form a concentration gradient to control the patterning and growth of the wing imaginal disc. In agreement with our finding that like *dpp*, *nord* is a target gene of the Hh signaling pathway in the wing imaginal discs (*Figures 1 and 2*), we observed a spatial-temporal correspondence between *nord* and *dpp* expression. Both Nord protein indicated by the Nord-GFP fusion derived from the *nord-GFP* protein-trap allele and Dpp precursor protein detected via an anti-Dpp prodomain antibody were present along the A/P boundary flanking the central wing pouch through the larval stage (*Figure 5A*). In the pupal wing, both Nord and Dpp expression does not change during the first 8 hr after pupation (AP) (*Figure 5A*). Subsequently, *dpp* disappears from the A/P boundary and commences expression in the differentiating LVs (deCelis, 1997; *Yu et al., 1996*); however, the Hh-dependent expression of *nord* along the A/P boundary remains for about 30 hr after pupation and diminishes after Dpp expression was detectable in both the LV and PCV regions (*Figure 5A*, *Figure 5—figure supplement 1*). Taken together, Nord proteins secreted from the A/P boundary stripe are expressed together or in close proximity with Dpp through the larval and early pupal (0–30 hr AP) stages.

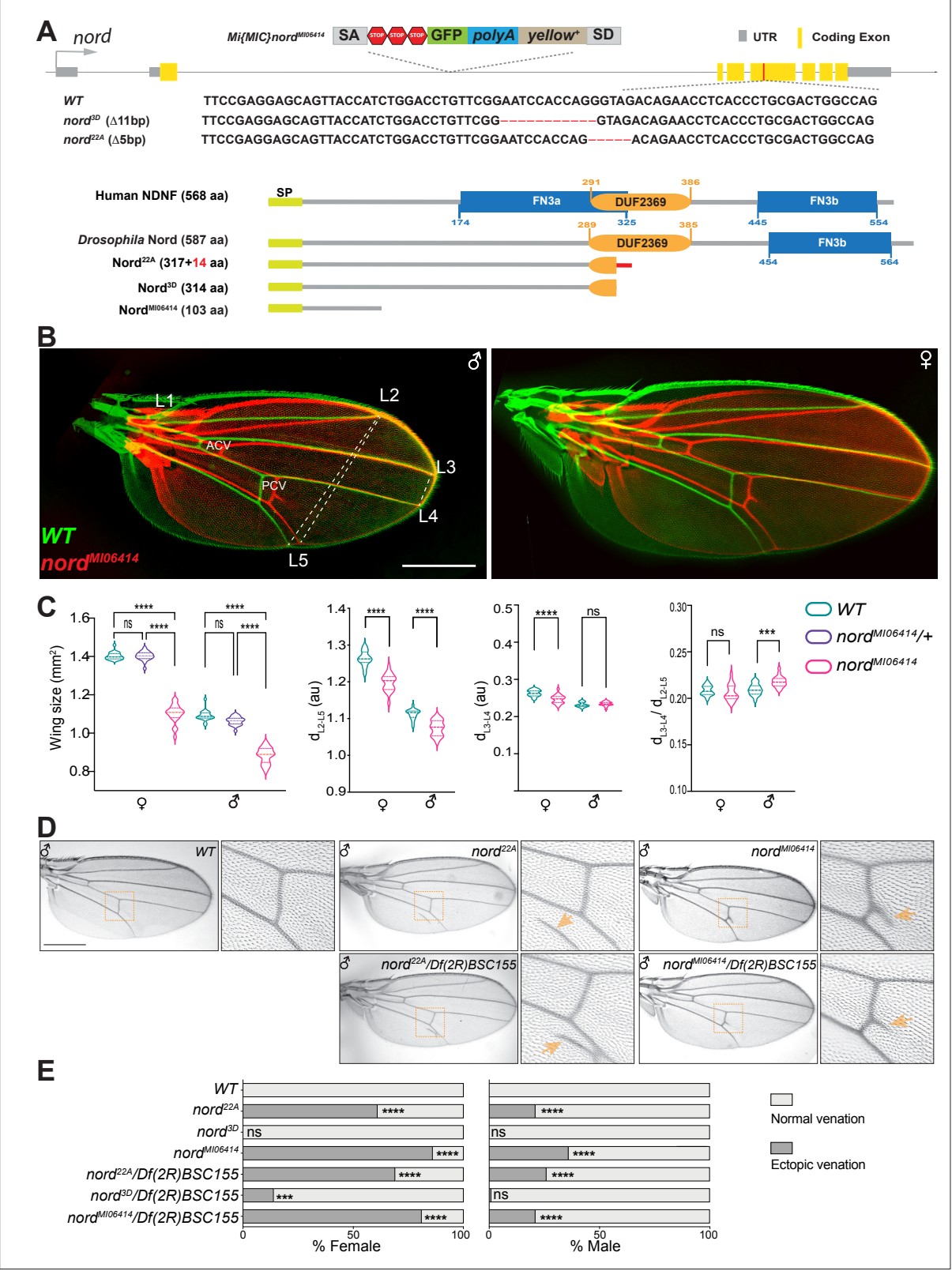

**Figure 4.** Nord is required for proper crossvein patterning and growth of the *Drosophila* wing. (**A**) Upper panel: schematic diagram of the wild-type *nord* locus, the gene-trap, and the CRISPR alleles of *nord*. A Minos-Mediated Integration Cassette (MiMIC) cassette consisting of a splice acceptor site followed by stop codons in all three reading frames was inserted into the first coding intron of *nord* in the *nord* gene-trap allele *nord*^MI0641^. Detailed view of the deleted regions in the *nord* mutant alleles generated by the CRISPR/Cas9 system. Lower panel: schematic diagram of the human Neuron-

*Figure 4 continued on next page*

*Figure 4 continued*

Derived Neurotrophic Factor (NDNF) protein, *Drosophila* Nord protein, and the predicted polypeptide products from the indicated *nord* mutant alleles. (**B**) Adult wings obtained from male or female flies with indicated genotypes were pseudo-colored and overlapped to show the size difference. ACV, anterior crossvein; PCV, posterior crossvein; LV, longitudinal veins (L1–L5). (**C**) Quantification of wing size, distance between distal ends of LVs L2 and L5 ($d_{L2-L5}$), L3 and L4 ($d_{L3-L4}$), and the ratio of $d_{L3-L4}/d_{L2-L5}$. Each bar shows the mean ± SD from n = 20 wings. All flies were grown at 25°C. One-way ANOVA followed by Sidak's multiple comparison test or unpaired two-tailed *t*-test was used for statistical analysis. ***p<0.001, ****p<0.0001, ns, not significant; au, arbitrary units. (**D**) Adult wings of flies with the indicated genotypes. Yellow arrowhead indicates ectopic vein near posterior crossvein (PCV) or L5. (**E**) Quantification of the ectopic venation phenotype in adult wings from flies with the indicated genotypes. n > 50. Two-sided Fisher's exact tests were used for statistical analysis. ***p<0.001, ****p<0.0001, ns, not significant. Scale bar, 500 μm.

The online version of this article includes the following figure supplement(s) for figure 4:

**Figure supplement 1.** Alignment of wild-type and truncated Nord proteins from the wild-type and *nord* mutant alleles.

**Figure supplement 2.** Nord is required for proper growth of the *Drosophila* wing.

**Figure supplement 3.** Location of deficiency lines used in the complementation test.

**Figure supplement 4.** *nord22A* and *nord3D* in trans to *Df(2R)BSC155* produced smaller wings compared to controls.

**Figure supplement 5.** Effects of loss of *nord* on wing trichome density.

**Figure supplement 6.** Illustration of wing trichome measurements.

## Opposing effects of Nord in modulating BMP signaling activity during wing growth and PCV patterning

Both the wing growth and patterning defects observed in *nord* mutant animals and the overlapping expression patterns of Nord and the BMP ligand Dpp point to a possible role of Nord in mediating Dpp/BMP signal transduction (*Figures 4 and 5A*). We, therefore, asked whether elimination of Nord alters the level of phosphorylated Mad (pMad), the primary downstream signal transducer of BMP signaling in the wing disc. We quantified pMad signal intensity in *nord* mutant wing discs and compared it to that of wild-type controls. We found that the pMad intensity was slightly reduced in *nord* mutant wing discs, and this reduction of pMad levels was more evident in the A compartment where Nord is expressed (*Figure 5B and C*, arrow). In contrast, by measuring the expression of high-threshold target gene *ptc* and low-threshold gene *dpp*, we found no obvious difference in the Hh signaling activity in the wild-type and *nord* mutant discs (*Figure 5B and C*, *Figure 5—figure supplement 2*).

Of note, besides BMPs, Hh signaling also patterns the developing *Drosophila* wing. Hh short-range activity is responsible for patterning the central L3–L4 region and determining the distance between L3 and L4 LVs (*Mullor et al., 1997*; *Strigini and Cohen, 1997*). When quantifying the distance between distal ends of LVs L3 and L4 ($d_{L3-L4}$) and that of L2 and L5 ($d_{L2-L5}$), we found no specific reduction in the L3–L4 region indicated by comparable or higher $d_{L3-L4}/d_{L2-L5}$ ratio in the female and male *nord* mutant wings (*Figure 4B and C*), which is consistent with normal Hh signaling activity during the development of *nord* mutant wing imaginal discs (*Figure 5B and C*). Together, these findings suggested a positive role of endogenous Nord in augmenting BMP signaling activity to promote wing growth.

During pupal wing development, BMP signaling is activated in the prospective CV regions prior to the appearance of other known vein promoting signals (*O'Connor et al., 2006*; *Ralston and Blair, 2005*), and abnormal BMP signaling can selectively affect the PCV and leave the LVs largely or entirely intact (*de Celis, 1997*; *Haerry et al., 1998*; *Khalsa et al., 1998*; *Nguyen et al., 1998*; *Ralston and Blair, 2005*; *Ray and Wharton, 2001*; *Wharton et al., 1999*; *Yu et al., 1996*). Therefore, we assessed the possible role of Nord in Dpp/BMP signal transduction during crossvein patterning. It is known that pMad becomes gradually refined to a narrow strip of precursor cells that form the future PCV during the first 24–28 hr AP (*Conley et al., 2000*; *Shimmi et al., 2005*). To examine whether the ectopic PCV in *nord* mutants is a direct consequence of enhanced BMP signaling, we quantified pMad signal intensity at the presumptive PCV region in pupal wings at various time points after pupation. In contrast to the rather restricted pMad domain in wild-type pupal wings, we detected a broadened pMad domain around the presumptive PCV in *nord* mutants from 19 to 20 hr AP (*Figure 5D and E*). Gradually, the ectopic pMad accumulation became an expanded patch adjacent to the presumptive PCV, indicating that BMP signaling was abnormally elevated in this region (*Figure 5D and E*).

During early pupal stage, it is notable that Nord expression was seen neither in the LV nor primordia PCV (*Figure 5A*, *Figure 5—figure supplement 1*). In agreement, we did not notice any ectopic PCV

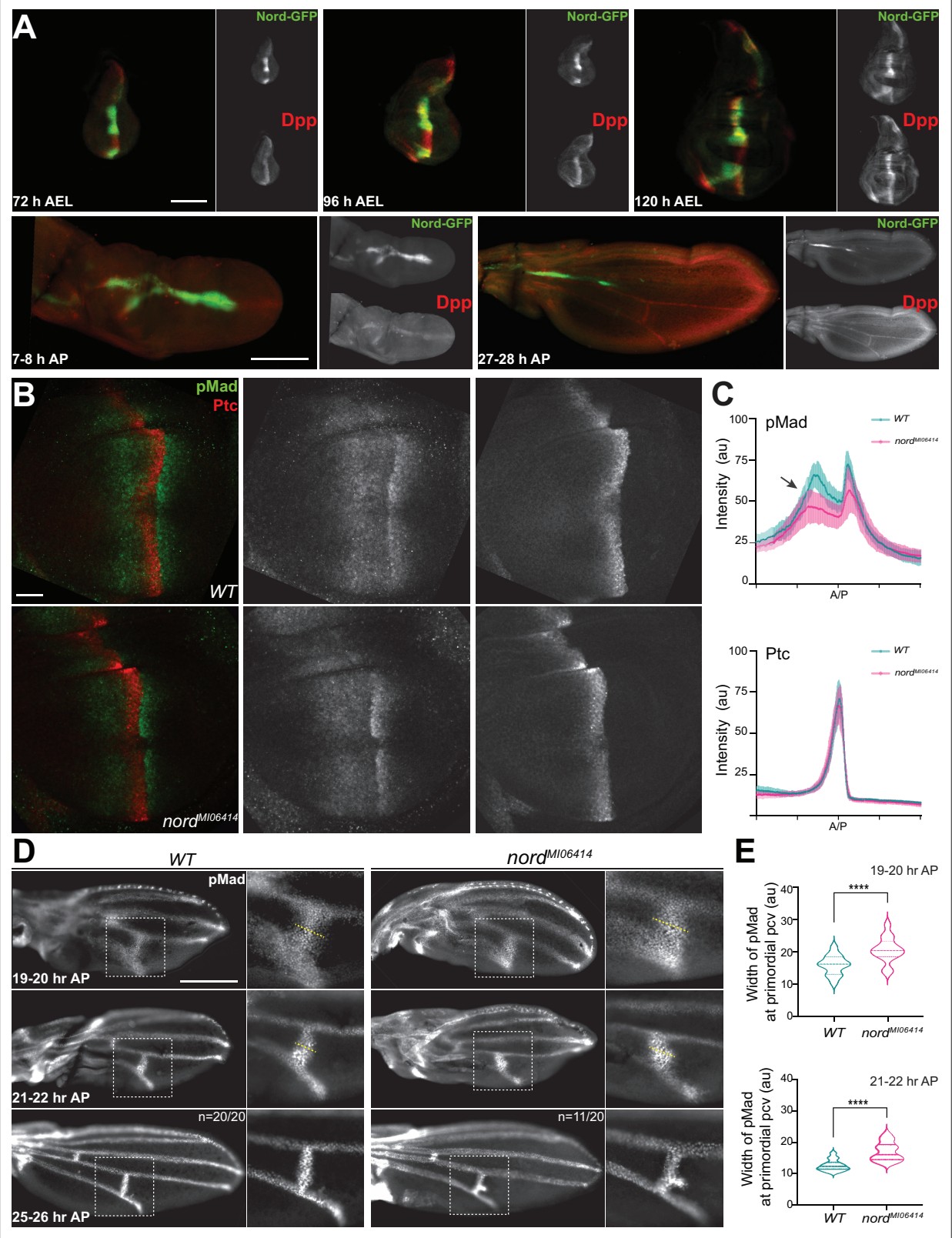

**Figure 5.** Nord modulates Bone Morphogenetic Protein (BMP) signaling at or above the level of Mad phosphorylation. (**A**) Expression of Nord in the wing discs through the third instar larval and early pupal stage. Upper panel: the third instar wing discs from *nord-GFP* (*Mi{PT-GFSTF.2}nord^{MI06414-GFSTF.2}/+*) larvae were collected at indicated time points after egg laying (AEL) and were immunostained for GFP (Nord, green) and the pro-domain of Dpp (red). Bottom panel: pupal wings from pupae carrying both *nord-GFP* were collected at indicated time points after pupation (AP) and were

*Figure 5 continued on next page*

*Figure 5 continued*

immunostained for GFP (Nord, green) and the pro-domain of Dpp (red). Scale bar, 100 µm. (**B**) Wing imaginal discs from third instar wild-type and *nord* mutant (*nord*^MI06414^) larvae were immunostained for pMad (green) and Ptc (red). Scale bar, 50 µm. (**C**) Plotted pixel intensity of pMad or Ptc as a function of A/P position. Each point shows the mean ± SD. n = 15. au, arbitrary units. (**D**) Anti-pMad staining in wild-type and *nord*^MI06414^ pupal wings at indicated hours AP. Scale bar, 100 µm. (**E**) Quantification of the width (yellow dashed line in panel **D**) of pMad signal at primordial posterior crossvein (PCV) at indicated time points. Each bar shows the mean ± SD from n > 10 pupal wings. The unpaired two-tailed *t*-test was used for statistical analysis. ****p<0.0001.

The online version of this article includes the following figure supplement(s) for figure 5:

**Figure supplement 1.** Expression of Nord in the wing discs during the early pupal stage.

**Figure supplement 2.** Loss of *nord* has little effect on the expression of Dpp in wing imaginal discs.

**Figure supplement 3.** Venation phenotype in adult wings associated with *nord* knocking down.

**Figure supplement 4.** Quantification of the ectopic venation phenotype in adult wings associated with *nord* knocking down.

in the adult flies when *UAS-nord-RNAi* was selectively expressed in the P compartment of the larval and pupal wing via the *hh-Gal4* driver (***Figure 5—figure supplements 3 and 4***). Given that Nord is a secreted protein (***Figure 3***), Nord proteins secreted from the A/P boundary would likely play a role in reducing excessive BMP signaling in the L4–L5 intervein region to prevent ectopic venation. To test this possibility, we analyzed the ectopic venation phenotype in flies carrying *ptc-Gal4*-driven expression of *UAS-nord-RNAi*. Although ectopic venation phenotype was observed when *nord* was selectively knocked down in the Hh-responding cells (***Figure 5—figure supplement 3***), the frequency of flies carrying ectopic PCV was lower when compared to *nord* mutant flies (***Figure 4D and E***, ***Figure 5—figure supplement 4***). This weaker phenotype likely resulted from residual Nord protein either due to incomplete *ptc-Gal4>UAS-nord-RNAi*-mediated *nord* knock down in the cells flanking the A/P boundary or due to possible Nord expression and secretion from other tissues. Nevertheless, opposite to the positive role of enhancing BMP signaling activity to promote growth of the larval wing discs, endogenous Nord also plays a negative role in inhibiting BMP signaling activity in the early pupal wing to prevent the formation of ectopic crossveins in the posterior compartment.

## Nord is a biphasic modulator of BMP signaling during wing growth

To better understand the role of Nord in modulating BMP signaling *in vivo*, we generated transgenic flies carrying a *Gal4*-inducible *UAS-Nord* transgene. Under control of the ubiquitous wing blade driver *nub-Gal4*, ectopic Nord expression resulted in reduced range and level of the pMad gradient in the third instar wing imaginal discs (***Figure 6A and B***, 25°C), and accordingly decreased wing size in both adult males and females (***Figure 6C and D***, 25°C; ***Figure 6—figure supplement 1***). In flies, minimal Gal4 activity is present at 16°C, while 29°C provides a balance between maximal Gal4 activity with minimal effects on fertility and viability due to growth at a high temperature (***Duffy, 2002***). Taking advantage of the temperature-dependent nature of Gal4 activity in *Drosophila*, we compared the dosage effect of Nord on pMad intensity in larvae raised at two different temperatures, 25 and 29°C. Indeed, in animals raised at 29°C expressing higher levels of exogenous Nord, we detected both a much reduced pMad gradient in the larval wing discs and more severely decreased wing size in the adult flies (***Figure 6A–D***, ***Figure 6—figure supplement 1***). In contrast, Hh signaling activity is relatively normal in the wing discs expressing ectopic Nord. Although the Ptc-expressing domain became narrower, we did not notice any obvious decrease in the levels of Ptc expression. (***Figure 6A and B***). Consistently, ectopic Nord expression did not cause any specific reduction in the distance of L3 and L4 based on the $d_{L3–L4}/d_{L2–L5}$ ratio (***Figure 6D***), suggesting that the wing growth defect caused by ectopic Nord is unlikely due to inhibition of Hh signaling. Interestingly, partial L5 and PCV loss was also noticed in the wings with more dramatic size reduction (***Figure 6C***, arrowheads and arrows), and the frequency of disrupted L5 and PCV was more dramatic when the flies were raised at higher temperatures and expressed higher levels of ectopic Nord (***Figure 6C***). Therefore, our observations indicate that ectopic Nord attenuates BMP signaling, leading to inhibition of wing growth and vein patterning. Along with the positive role of endogenous Nord in enhancing BMP signaling to promote wing growth, we propose a model that Nord has both positive and negative effects in modulating BMP signaling activity, where low (endogenous) levels of Nord enhance and high (ectopic) levels of Nord inhibit BMP signaling.

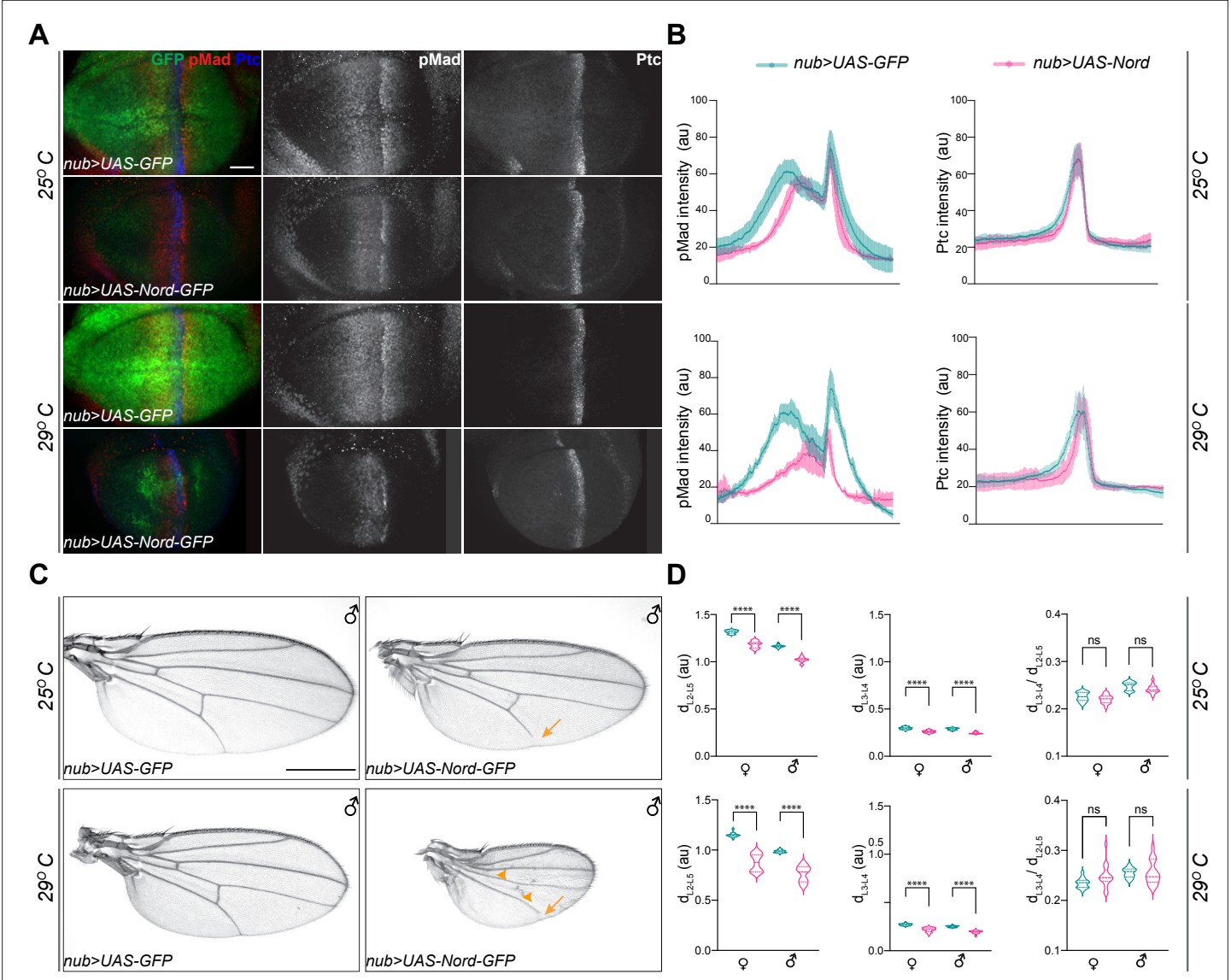

**Figure 6.** Ectopic expression of Nord attenuates Bone Morphogenetic Protein (BMP) signaling *in vivo*. (**A**) Larvae expressing the indicated transgenes driven by *nub-Gal4* were raised at 25 or 29°C. Wing imaginal discs were collected from these larvae in the late third instar stage, and then immunostained for anti-pMad (red), anti-Ptc (blue), and anti-GFP (green). Scale bar, 20 µm. (**B**) Plotted pixel intensity of pMad or Ptc as a function of anteroposterior (A/P) position. Each point shows the mean ± SD. 25°C: n = 12. 29°C: n = 7. (**C**) Adult wings from male flies, which expressed the indicated transgenes driven by *nub-Gal4* and were raised at 25 and 29°C. Scale bar, 500 µm. Arrows indicate reduced L5, and arrowheads indicate reduced crossveins. (**D**) Quantification of distance between distal ends of longitudinal veins L2 and L5 ($d_{L2-L5}$), L3 and L4 ($d_{L3-L4}$), and the ratio of $d_{L3-L4}$ / $d_{L2-L5}$. Each bar shows the mean ± SD from n > 14 wings. au, arbitrary units. The unpaired two-tailed *t*-test was used for statistical analysis. ****p<0.0001, ns, not significant.

The online version of this article includes the following figure supplement(s) for figure 6:

**Figure supplement 1.** Ectopic Nord expression leads to reduction of the wing size.

## Nord is a dosage-dependent modulator of BMP signaling during PCV patterning

To further examine the model that Nord is a dosage-dependent modulator of BMP signaling, we sought to manipulate Nord levels during pupal wing development in the posterior compartment and examine the effects on PCV formation since with this structure it is possible to assay both positive and negative roles by looking for an ectopic versus a loss of crossvein formation. Accordingly, we used both *hh-Gal4* and *en-Gal4* to drive different levels of ectopic Nord in the P compartment of the

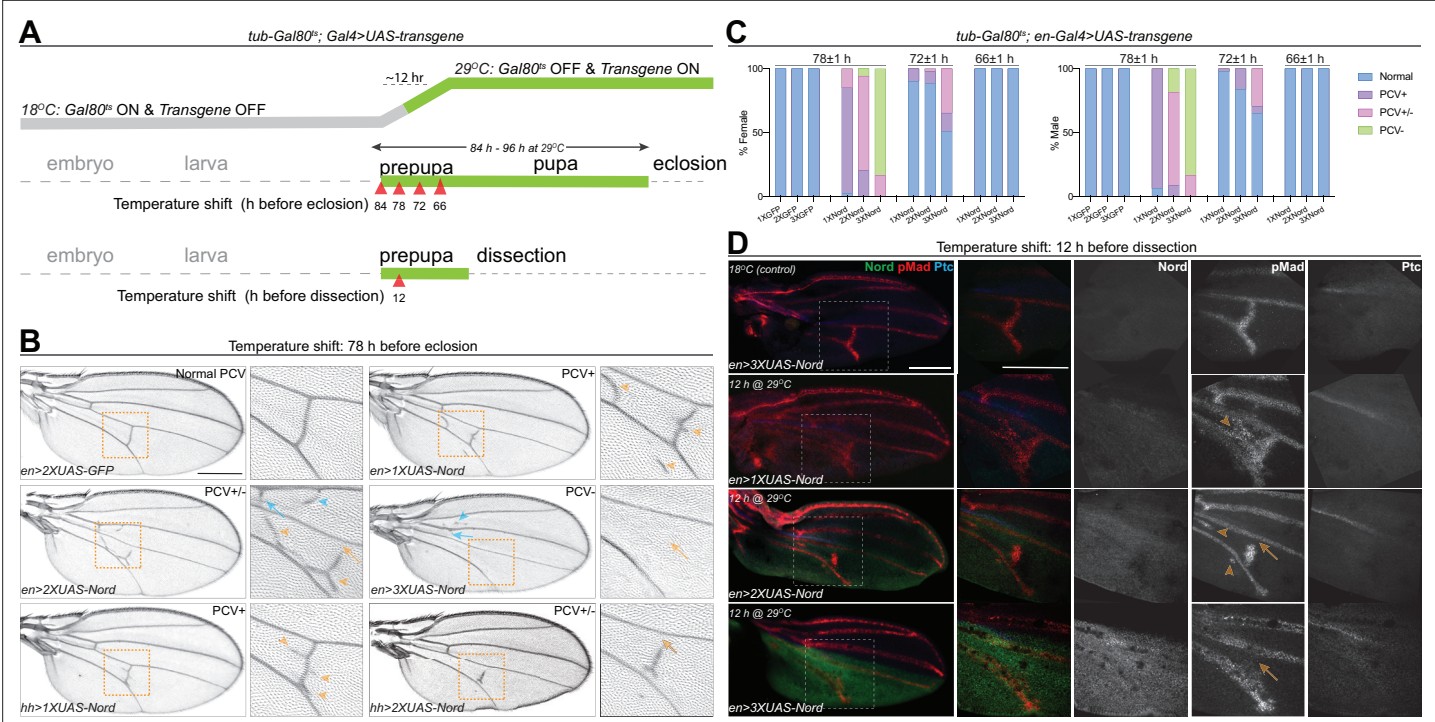

**Figure 7.** Nord is a dosage-dependent modulator of the Bone Morphogenetic Protein (BMP) signaling. (**A**) Upper row: schematic diagram of temporal *UAS-Nord* expression in the posterior compartment of wing discs using the temperature-sensitive Gal80 (Gal80^ts) system. At the permissive temperature (18°C), Gal4 activity is blocked by Gal80^ts. At the restrictive temperature (29°C), Gal80^ts is unable to repress Gal4, which then induces expression of *the UAS-transgenes*. Middle and lower rows: timing of temperature shift. Embryos and larvae are grown at 18°C, and prepupae are transferred to 29°C at the indicated time points before eclosion (middle row) or before dissection (lower row). (**B**) Representative adult wings from flies that carry the indicated transgenes under the control of *tub-Gal80^ts* together with *hh-Gal4* or *en-Gal4*. The animals were grown at 18°C till prepupa stage, and then transferred to 29°C at 78 hr before eclosion. Yellow arrowhead indicates ectopic posterior crossvein (PCV); yellow arrow indicates reduced PCV; blue arrowhead indicates ectopic anterior crossvein (ACV); blue arrow indicates reduced ACV. Scale bar, 500 μm. (**C**) Quantification of PCV phenotypes in adult wings of female and male flies with indicated genotypes. The animals were grown at 18°C till prepupa stage, and then transferred to 29°C around 78, 72, or 66 (±1) hr before eclosion. n > 30 wings for each genotype at a given temperature shift time point. (**D**) Representative pupal wing from larvae that carry the indicated transgenes under the control of *tub-Gal80^ts* together with *en-Gal4*. The animals were grown at 18°C till prepupa stage, and then transferred to 29°C at 12 hr before dissection. The collected pupal wings were immunostained for anti-pMad (red), anti-Ptc (blue), and anti-GFP (Nord-GFP, green). Yellow arrowhead indicates ectopic pMad around the primordial PCV; yellow arrow indicates reduced pMad signal around the primordial PCV. Scale bar, 100 μm.

The online version of this article includes the following figure supplement(s) for figure 7:

**Figure supplement 1.** Spaciotemporal expression of exogenous Nord in the pupal wing leads to abnormal posterior crossvein (PCV) patterning.

**Figure supplement 2.** Ectopic Nord secreted from the posterior compartment induces ectopic crossvein growth in the anterior compartment.

wing disc, including the PCV primordia where Nord is not normally expressed. To avoid the influence of prior larval stage Nord expression on the role of BMP signaling specifically during PCV pupal development, we used *Gal4* together with *tub-Gal80^ts* (a temperature-sensitive version of Gal80) to temporally control *UAS-Nord* expression. At a low temperature (18°C), Gal80^ts represses the function of Gal4 bound to a *UAS* sequence but is unable to do so at the restrictive temperature (29°C) (*McGuire et al., 2003*). We performed temperature-shift experiments (from 18 to 29°C) to initiate ectopic Nord expression at different times during pupal development and characterized the impact on PCV patterning in the resulting adult wings. Consistent with a previous report (*Roberts, 1998*), the length of pupal period became shorter after the temperature was raised from 18 to 29°C due to temperature-dependent effects on the growth rate, and we found that shifting the temperature to 29°C right after pupation led to eclosion ~84–96 hr later (*Figure 7A*). We found that activation of *UAS-Nord* expression by temperature shifts at 78 hr or earlier before eclosion caused the most severe PCV phenotype, but activation at 66 hr before eclosion or later resulted in essentially normal PCV (*Figure 7B and C*, *Figure 7—figure supplement 1*). More importantly, we found that the abnormal

PCV phenotypes relied on the levels of exogenous Nord: expression of a lower level of Nord (*1XUAS-Nord*) led to ectopic PCV, moderate levels (*2XUAS-Nord*) yielded a mixed phenotype with both ectopic and slightly reduced PCV, whereas high levels (*3XUAS-Nord*) gave rise to nearly complete loss of PCV (*Figure 7B and C*, *Figure 7—figure supplement 1*). Of note, consistent with the fact that Nord is a secreted protein, exogenous Nord expressed within the *hh-Gal4* or *en-Gal4*-expressing domains was also able to influence crossvein patterning in the A compartment although with a lower frequency (*Figure 7B*, blue arrows and blue arrowheads, *Figure 7—figure supplement 2*).

The critical time window in which the primordial PCV responds to exogenously expressed Nord (66–78 hr before eclosion at 29°C) coincides with the stage during which BMP signaling induces PCV formation. We next tested whether the PCV phenotypes caused by ectopic Nord were correlated with alterations of BMP signaling activity. As shown in *Figure 7D*, adult PCV defects and pMad patterns in pupal wings showed a correlation with the level of ectopic Nord, where the temperature shift occurred 12 hr after the start of pupal development. In pupal wings expressing a low level of exogenous Nord (*1XUAS-Nord*), ectopic pMad and ectopic crossveins were detected, while moderate overexpression (*2XUAS-Nord*) led to a mixed phenotype of partial pMad and crossvein vein loss or wider pMad and ectopic veins. When high Nord levels (*3XUAS-Nord*) were expressed in the pupal wings, pMad and the PCV were largely absent. Although the temporal resolution is somewhat limited, the results clearly indicate that the level of Nord influences the outcome of PCV patterning during the early pupal development where a lower level of exogenous Nord resulted in enhanced BMP signaling and ectopic PCV, while higher levels of exogenous Nord inhibited BMP signaling and caused disrupted or depleted PCVs. Taken together, these results demonstrated that Nord is a dosage-dependent modulator of BMP signaling both in wing growth and crossvein patterning.

## Nord binds to Dpp and interferes with BMP signaling *in vitro*

The activity of BMPs is modulated by a large variety of binding proteins that can either enhance or inhibit their signaling in a context-dependent manner (*Chang, 2016*; *Umulis et al., 2009*). Given the spatial-temporal overlapping expression of *nord* and *dpp* and the dosage-dependent modulation of Nord on BMP signaling in wing growth and crossvein patterning, we assessed whether Nord modulates BMP signaling via binding to either of the two BMP ligands, Dpp and Gbb, that are expressed in the developing *Drosophila* wing. We turned to an *in vitro* model to examine possible interactions of Nord with Dpp and Gbb by carrying out co-immunoprecipitation assays. We added a GFP tag to the C-terminus of Nord and expressed the fusion protein in *Drosophila* S2 cells. The conditioned medium from Nord-GFP-expressing cells was collected and mixed with medium from cells transfected with FLAG-tagged Dpp and HA-tagged Gbb alone or in combination. The mixed-media were then incubated with anti-FLAG or anti-HA antibody-coupled beads to precipitate the BMP ligands. We found that Nord co-precipitated with Dpp and, to a lesser extent, with Gbb (*Figure 8A*). Additionally, we observed an increased level of Nord proteins co-precipitated with Dpp or Gbb when Dpp and Gbb were co-expressed (*Figure 8A*), indicating that Nord may have a higher affinity for Dpp-Gbb heterodimers formed in cells co-expressing Dpp and Gbb.

We next collected conditioned medium from S2 cells expressing Dpp or Dpp-Gbb with or without co-expressed Nord (source cell) and compared the levels of processed Dpp or Dpp-Gbb within the different conditioned media. We found that far less cleaved Dpp protein was released into the conditioned medium when Nord was co-expressed (*Figure 8B*, compare lane M3 with M2). Likewise, the same negative effect of Nord was observed when the medium was collected from cells expressing both Dpp and Gbb (*Figure 8B*, compare lane M5 with M4), albeit the increased total amount of ligands likely reflects the release of Dpp-Gbb heterodimers (*Figure 8B*, compare lane M4 with M2). Thus, the presence of Nord affected the release into the media of both Dpp and Dpp-Gbb, likely via binding to the BMP ligand. We then determined the activity of the collected conditioned media in an S2 cell-based signaling assay. Because endogenous levels of Mad protein in S2 cells are low, we established Mad-S2 responding cells that stably express a FLAG epitope-tagged Mad transgene (FLAG-Mad) (*Ross et al., 2001*). Upon incubation of the Mad-S2 responding cells with conditioned medium collected from the source cells, BMP signaling activity was monitored by measuring the pMad signal intensity. Conditioned medium containing either Dpp or Dpp-Gbb, but not that containing Nord alone, was able to induce Mad phosphorylation (*Figure 8C*). In agreement with the dramatically reduced amount of Dpp or Dpp-Gbb ligands (*Figure 8B*), the conditioned medium collected from

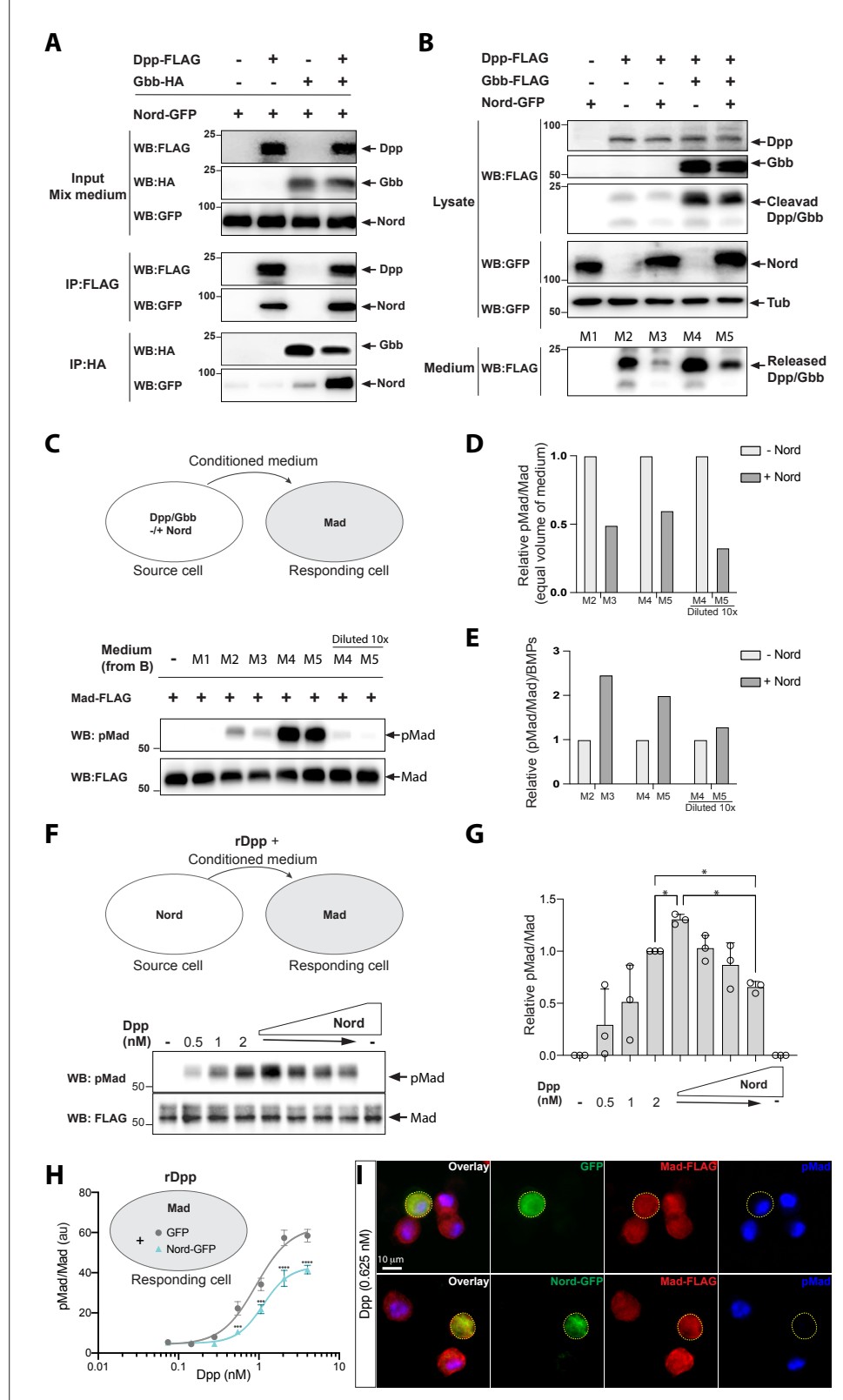

**Figure 8.** Nord binds to Decapentaplegic (Dpp) and attenuates Bone Morphogenetic Protein (BMP) signaling *in vitro*. (**A**) Co-immunoprecipitation of Nord with the BMP ligands Dpp and Glass-bottom boat (Gbb). Medium from S2 cells transfected for expression of GFP-tagged Nord were mixed with medium from cells expressing FLAG-tagged Dpp and HA-tagged Gbb alone or in combination, followed by incubation with anti-FLAG or

*Figure 8 continued on next page*

*Figure 8 continued*

anti-HA antibody-coupled beads overnight at 4°C. Precipitated proteins were analyzed by Western blotting with indicated antibodies. Nord was immunoprecipitated with Dpp and, to a lesser extent, with Gbb. The amount of immunoprecipitated Nord was increased when Dpp and Gbb were co-transfected. (**B**) S2 cells were transfected for expression of FLAG-tagged Dpp or Gbb with or without GFP-tagged Nord (source cell). Both cell lysate and conditioned medium from the source cells were collected and followed by Western blot analysis. Loading was controlled by probing the blot for tubulin. The amount of Dpp or Dpp-Gbb ligands released into the medium was reduced when Nord was co-expressed in the source cells. (**C**) Comparison of BMP signaling activities of conditioned media in a cell-based signaling assay. After incubating the conditioned media collected in (**B**) with S2 cells stably expressing the FLAG-Mad transgene (Mad-S2, responding cell) for 1 hr at room temperature, the responding cells were washed and lysed. The lysates were probed with anti-pMad and anti-FLAG antibodies to detect both the phosphorylated Mad and total Mad protein, respectively. (**D, E**) Quantification of the Western blot data in panels (**B**) and (**C**). The levels of the secreted BMP ligands from the medium (anti-FLAG in panel **B**), phosphorylated Mad (anti-pMad in panel **C**), and total Mad (anti-FLAG in panel **C**) were measured based on the band intensity. The signaling activity from an equal volume of conditioned medium was determined by the ratio of pMad and the corresponding total Mad (pMad/Mad) in panel (**C**), which was then normalized to the condition without the exogenous Nord-GFP to calculate a relative signaling activity (**D**), or normalized to the secreted ligand amount to calculate the relative ligand activity ([pMad/Mad]/BMPs) (**E**). (**F**) Mad-S2 cells were treated with a recombinant Dpp peptide (rDpp) in the absence or presence of conditioned medium containing raising levels of Nord for 1 hr at room temperature, the responding cells were washed and lysed. The lysates were probed with anti-pMad and anti-FLAG antibodies to detect both the phosphorylated Mad and total Mad protein, respectively. (**G**) Quantification of the Western blot data in panel (**F**). The phosphorylated Mad (anti-pMad) and total Mad (anti-FLAG) levels were measured based on the band intensity. The signaling activity from each conditioned medium was determined by the ratio of pMad and the corresponding total Mad (pMad/Mad), which was then normalized to the condition with 2 nM Dpp but no additional Nord. Panel (**F**) is representative of n = 3 independent experiments. Each point shows the mean ± SD. One-way ANOVA test with Tukey's multiple comparison was used for statistical analysis, and a significant difference was considered by *$p<0.05$. (**H**) Mad-S2 cells were transiently transfected for expression of GFP or Nord-GFP. 48 hr after transfection, the cells were treated with recombinant Dpp peptides for 1 hr. Upon treatment, the cells were washed, fixed, and stained by anti-FLAG to detect total Mad and anti-pMad to detect phosphorylated Mad. The average pMad levels were measured and normalized to the total Mad levels, and then plotted against different Dpp concentrations (0–5 nM). Each point shows the mean ± SD, n > 10. au, arbitrary units. The unpaired two-tailed *t*-test was used for statistical analysis. ***$p<0.001$, ****$p<0.0001$. (**I**) Representative images of Mad-S2 cells treated by 0.625 nM Dpp. Scale bar, 10 μm.

The online version of this article includes the following source data for figure 8:

**Source data 1.** Uncropped Western blot for *Figure 8*.

source cells co-expressing Nord showed lower pMad signal intensity compared to that collected from source cells lacking Nord co-expression (*Figure 8C and D*). Of note, Nord is not a membrane-tethered protein (*Figure 3A*), but we observed noticeable amounts of Nord deposited on the surface of source cells, as well as the surrounding extracellular matrix (*Figure 3C and D*). Thus, the matrix-associated Nord may sequester Dpp and Dpp-Gbb ligands to the source cells and thereby reduce ligand level in the media, which in turn leads to decreased BMP signaling activity in the responding cells (*Figure 8C and D*).

## Dosage-dependent modulation of BMP signaling by Nord *in vitro*

*In vivo*, our loss- and gain-of-function analyses suggested that Nord is a dosage-dependent modulator of BMP signaling in wing growth and crossvein patterning. In the *in vitro* signaling assay, ectopically expressing Nord in the source cells led to reduced BMP ligand release and thus decreased the signaling activity (pMad/Mad) from an equal volume of conditioned medium (*Figure 8D*). However, when the signaling activity was further normalized to the ligand amount (*Figure 8B*), the relative ligand activity ([pMad/Mad]/BMPs) from the medium without Nord is lower than that with Nord (*Figure 8E*), suggesting that while ectopic Nord expressed in the source cells reduced the levels of released ligand, that soluble ligand appears to have a higher signaling activity perhaps as a result of an association with Nord (*Figure 8E*). To further assess whether Nord directly modulates the signaling activity of released BMP ligands in a dosage-dependent manner, we treated Mad-S2 cells with a recombinant Dpp peptide (rDpp) in the absence or presence of conditioned medium containing increasing amounts of Nord (*Figure 8F*). Consistent with our *in vivo* analyses, *in vitro*, we also observed a dosage-dependent

signaling profile of rDpp in the presence of increasing concentrations of Nord where the lowest tested level of Nord enhanced signaling while higher Nord levels reduced Dpp signaling activity (*Figure 8F and G*).

Notably, the highest level of Nord supplied from conditional medium failed to completely inhibit the free rDpp-induced BMP signaling. We next transiently transfected Mad-S2 cells for expression of Nord-GFP. Similar to Nord supplied from conditioned medium, we found that transient expression of Nord-GFP in the Mad-S2 cells partially inhibited the average Mad phosphorylation induced by exogenous rDpp (*Figure 8H*). Remarkably, immunofluorescent staining revealed that cells expressing Nord-GFP exhibited much lower pMad when compared to the surrounding Mad-S2 cells lacking Nord-GFP expression (*Figure 8I*). In contrast, cells expressing GFP did not cause a noticeable reduction in pMad level in rDpp-treated Mad-S2 cells (*Figure 8I*). The much stronger cell-autonomous inhibition of Nord on BMP signaling is likely due to a much higher level of ectopic Nord when over-expressed in the Mad-S2 cells (*Figure 8I*). Taken together, both our *in vivo* and *in vitro* assays demonstrated that Nord not only can sequester BMP ligands, thereby impeding their release from the source cells, but may also directly modulate the activity of released BMP ligands in a dosage-dependent manner, where low levels promote and high doses attenuate BMP signaling.

## Discussion

In *Drosophila*, the short-range morphogen Hh and the long-range morphogen BMP function together to organize wing patterning (*Lecuit et al., 1996*; *Mullor et al., 1997*; *Nellen et al., 1996*; *Strigini and Cohen, 1997*; *Zecca et al., 1995*). It has been previously shown that the Hh signal shapes the activity gradient of BMP by both inducing the expression of Dpp and simultaneously downregulating the Dpp receptor Tkv, resulting in lower responsiveness to Dpp in cells at the A/P compartment border (*Tanimoto et al., 2000*). In this study, we showed that the activity of BMP is further fine-tuned by another previously unknown Hh-dependent mechanism. Using a genome-wide expression profiling of the *Drosophila* wing imaginal discs, we identified *nord* as a novel target gene of the Hh signaling pathway (*Figures 1 and 2*). Nord and its homolog NDNF belong to a family of secreted proteins that can exist in two distinct pools: diffusible Nord/NDNF proteins that can reach a longer distance and membrane/matrix-associated Nord/NDNF proteins spreading within a short distance from the source cells (*Figure 3*). During larval and early pupal wing development, Nord is expressed together or in close proximity with the BMP ligand Dpp along the A/P compartment boundary (*Figure 5A*, *Figure 5—figure supplement 1*). Elimination of *nord* caused a reduction of overall wing size and resulted in ectopic PCV formation. Both of these phenotypes are attributable to alterations of BMP signaling activity as monitored by the level of Mad phosphorylation, yet in opposite directions: loss of *nord* led to decreased pMad in larval wing discs, whereas ectopic pMad surrounded the primordial PCV in *nord* mutant pupal wings (*Figures 4 and 5*). Moreover, expressing exogenous Nord at different levels and during different developmental stages and contexts showed that Nord is a dosage-dependent modulator of BMP signaling both in wing growth and crossvein patterning (*Figures 6 and 7*). At the molecular level, we further demonstrated that Nord is a BMP-binding protein that directly enhances or inhibits BMP signaling in cultured S2 cells (*Figure 8*).

Combining the genetic and biochemical evidence, we propose that Nord mediates BMP signaling activity through binding of the BMP ligands Dpp and Dpp-Gbb (*Figure 9*). Depending on the levels of Nord proteins and the source/types of BMP ligands, Nord-mediated binding of Dpp and Dpp-Gbb may promote or repress BMP signaling activity. Additionally, the existence of two spatially distinct pools of diffusible and membrane/matrix-associated Nord proteins may introduce further complications in Nord-mediated BMP signaling regulation. In the wild-type wing discs, expressed in a subset of Dpp-secreting cells along the A/P boundary, Nord binds and enhances the local BMP signaling activity by augmenting ligand concentration near the Nord/Dpp-secreting cells. Meanwhile, Nord also impedes the mobilization of Dpp, especially the long-range BMP signaling mediator Dpp-Gbb heterodimer (*Figure 9A*). Loss of *nord* simultaneously led to reduced local BMP and increased long-range BMP activities, and therefore gave rise to the seemingly opposite phenotypes of reduced wing size and ectopic PCV (*Figure 9B*). In contrast, low levels of ectopic Nord in the P compartment autonomously increased BMP signaling activity (*Figure 9C*), whereas high levels of Nord, either in the P compartment or throughout the wing pouch, inhibited BMP signaling activity likely through interfering with the normal BMP reception (*Figure 9D and E*). Taken together, we propose that Hh-induced Nord

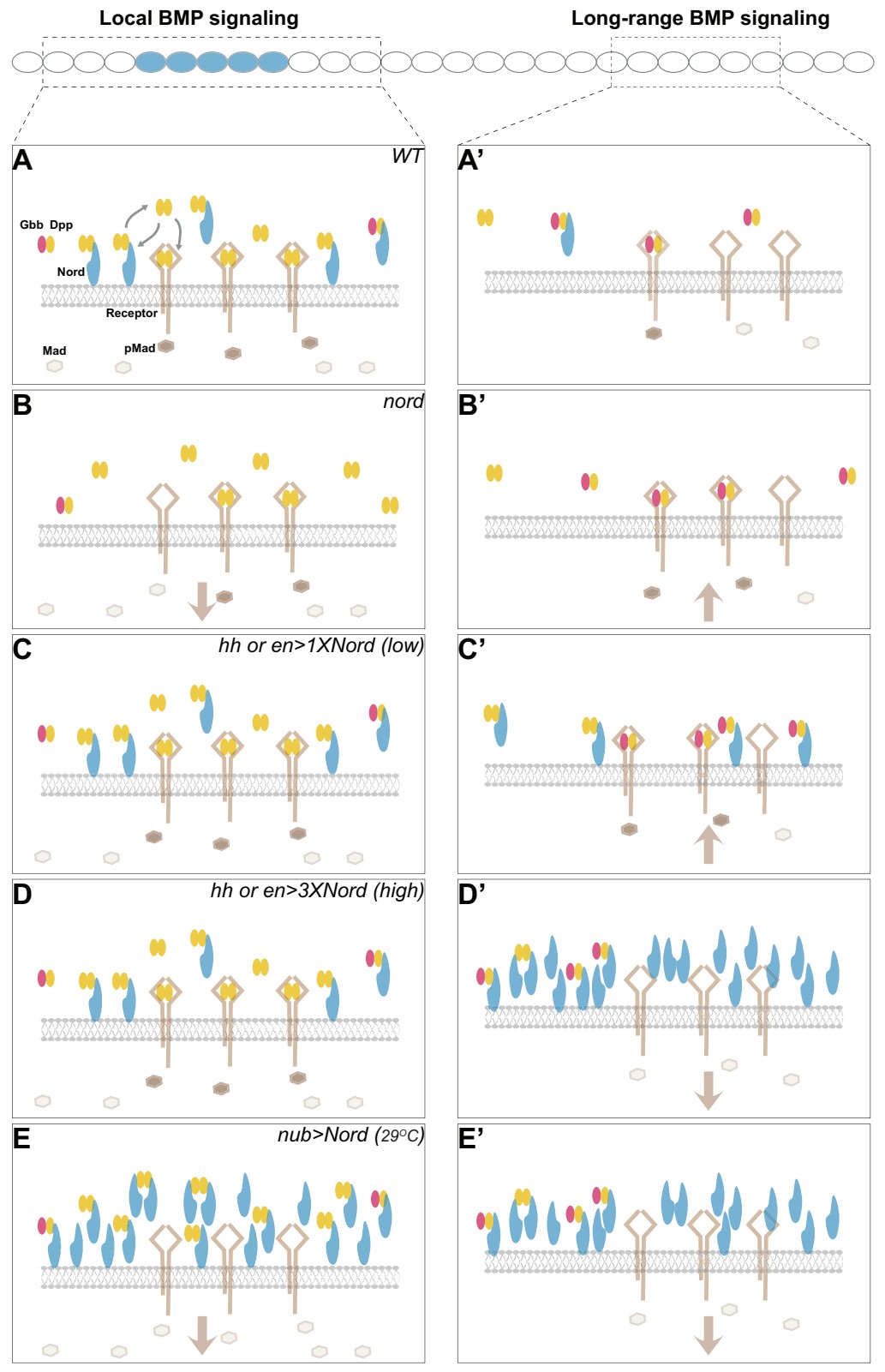

**Figure 9.** A model for dosage-dependent modulation of Bone Morphogenetic Protein (BMP) signaling by the Hedgehog (Hh) target gene *nord*. Illustration of potential concentration- and location-dependent mechanism of Nord as a dosage-dependent BMP signaling modulator. The source cells of Nord (and Decapentaplegic [Dpp]) are labeled in blue, and the responding cells in focus are indicated by the dashed line. Note that Nord could be either

*Figure 9 continued on next page*

*Figure 9 continued*

diffusible or membrane/matrix-associated. (**A, A′**) In the wild-type wing discs, *nord* is induced by the Hh signal along the anteroposterior (A/P) compartment boundary flanking the central wing pouch, and thus is expressed in a subset of Dpp-secreting cells. We propose that membrane/matrix-associated Nord-mediated binding of Dpp and Dpp-Gbb both enhances the local BMP signaling activity by augmenting ligand concentration near the Nord/Dpp-secreting cells and impedes the mobilization of Dpp, especially the long-range BMP signaling mediator Dpp-Gbb heterodimer. The diffusible pool of Nord is more likely to interfere with BMP reception. (**B, B′**) Loss of *nord* simultaneously leads to reduced local BMP and increased long-range BMP activities, and therefore gives rise to the seemingly opposite phenotypes of reduced wing size and ectopic posterior crossvein (PCV), both of which are attributable to alteration of BMP activity at the level of Mad phosphorylation (pMad). (**C–E, C′-E′**) Nord misexpression experiments. Low levels of ectopic Nord in the P compartment increase BMP signaling activity (**C, C′**), whereas high levels of Nord, either in the P compartment (**D, D′**) or throughout the wing pouch (**E, E′**), inhibit BMP signaling activity. The altered BMP signaling activities are reflected by the pMad levels.

expression provides an exquisite regulation of the strength and range of BMP signaling in the developing *Drosophila* wing.

## Nord is a novel, multifunctional BMP-binding protein

The activity of TGF-β type factors, including the BMP subfamily, is modulated by a large variety of binding proteins that can either enhance or inhibit their signaling in a context-dependent manner (*Chang, 2016*; *Umulis et al., 2009*). These modulator proteins vary broadly in structure, location, and mechanism of action. Well-known extracellular and freely diffusible proteins include Noggin, Tsg, Follistatin, the CR (cysteine-rich) domain containing proteins such as Chordin/Sog, and the Can family named after two founding members, Dan and Cerberus (*Chang, 2016*). With the exception of Tsg and Tsg/Sog or Tsg/Chordin complexes that in some cases can promote BMP signaling, all of these factors behave as antagonists, where BMP binding prevents association of the ligand with the receptor complex.

The other broad category of BMP-binding proteins includes membrane-bound or matrix-associated proteins and, in contrast to the highly diffusible class of BMP-binding factors, these proteins often act as either agonists or antagonists depending on context. These proteins are also structurally diverse, but to date, none contain FN3 or DUF2369 domains that are characteristic of Nord and NDNF, its vertebrate counterpart. From a mechanistic point of view, perhaps the two most instructive *Drosophila* members of this class of modulators are the heparan sulfate proteoglycan (HSPG) Dally and the CR-containing protein Cv-2. HSPGs are well characterized as modulators of growth factor signaling (*Nakato and Li, 2016*). In the case of FGFs, HSPGs act as true co-receptors in which they form a tripartite complex with ligand and FGFR, the signaling receptor (*DiGabriele et al., 1998*; *Schlessinger et al., 2000*). However, they can also mediate signaling in other ways. Analysis of *dally* loss-of-function clones in imaginal discs demonstrates that it has both cell-autonomous and non-autonomous effects with respect to BMP signaling (*Akiyama et al., 2008*; *Belenkaya et al., 2004*; *Fujise et al., 2003*). In general, low levels tend to promote signaling while high doses attenuate signaling. Many models have been put forth to explain these opposing effects and often come down to balancing ligand sequestration and diffusion properties. For instance, in the absence of HSPGs, Dpp may more freely diffuse away from the disc epithelial cell surface. In this case, HSPG acts to enhance signaling by keeping Dpp tethered to the cell surface where it can engage its signaling receptors. On the other hand, a high level of HSPG may compete with signaling receptors for BMP binding and thereby reduce signal (*Nakato and Li, 2016*).

The situation with respect to signal modulation becomes even more complex for factors such as Nord that bind both HSPGs (*Akiyama et al., 2021*) and BMPs (this report). An instructive example to consider is Cv-2, a secreted factor that, like Nord, binds both to HSPGs and BMPs and is also induced by BMP signaling (*Serpe et al., 2008*). Like Dally, Cv-2 also has dose-dependent effects on signaling in wing imaginal discs, where low levels enhance while high levels inhibit BMP signaling. By virtue of being bound to HSPGs, it may simply function as an additional tethering molecule that keeps BMPs localized near the cell surface. However, Cv-2 has the unique property that it is also able to bind Tkv, a *Drosophila* BMPR type I receptor (*Serpe et al., 2008*). This has led to speculation that it could act as an exchange factor that aids in handing off a BMP ligand from the HSPG pool to the type I receptor.

Mathematical modeling showed that this mechanism can produce a biphasic signal depending on affinities of the various BMP-binding proteins involved and their concentrations (*Serpe et al., 2008*).

In the case of Nord, its mechanism of action is likely compatible with a variety of these and/or alternative models. While we have shown that Nord is a BMP-binding protein and *Akiyama et al., 2021* have shown that it also binds HSPGs, it is not clear whether the BMP and HSPG-binding sites overlap or are distinct and where they are positioned relative to the FN3 and DUF2369 domains. This is an important issue to consider with respect to the two CRISPR mutants that we generated that truncate Nord within the DUF2369 domain. Interestingly, the *nord³ᴰ* allele appears to retain some function since it does not generate ectopic crossveins as do the *nordᴹᴵ⁰⁶⁴¹⁴* or *nord²²ᴬ* alleles, yet *nord³ᴰ* still produces small wings in transheterozygous combination with a deficiency or *nord²²ᴬ*, consistent with having lost the BMP growth-promoting ability. The discrepancy in crossvein patterning between the different *nord* alleles may be explained by a difference in residual function of the various truncated Nord protein products (*Figure 4*). Because the *nordᴹᴵ⁰⁶⁴¹⁴* allele yields a much shorter predicted Nord peptide compared to the two CRISPR alleles, it is likely to behave as a protein null with a stronger phenotype. The two *nord* CRISPR alleles, although similar in the sequence deleted from the C-terminus, differ in how many non-*nord* encoded amino acids occur between the frameshift and the stop codon. The *nord²²ᴬ* allele has additional 14 amino acids relative to *nord³ᴰ*. Perhaps this extension of the truncated fragment destabilizes or interferes with residual function found in the *nord³ᴰ* allele. Additional biochemical studies defining the BMP and HSPG-binding sites, the stability of truncated Nord fragments, and whether Nord can also associate with either the type I or II receptors will aid in formulating a more precise mechanistic model.

## Is Nord structure and function conserved across species?

Nord shows some sequence similarity to the NDNF family of proteins (*Figure 3A*). Based on a very recent study, like many other neurotrophic factors, NDNF arose in the ancestor of bilaterians or even later (*Heger et al., 2020*). In agreement, by analyzing the genome and EST sequences from various organisms, we found that nearly all bilaterian animals have either single or multiple orthologous genes for Nord/Ndnf (*Figure 3A*). Of note, we did not identify any Ndnf homologs in the flatworm *Planarian*, but these factors are highly conserved across vertebrates (*Kuang et al., 2010*). All vertebrate family members contain a signal peptide, two FN3-like repeats, and a domain of unknown function (DUF2369) that is now referred to as the NDNF domain. The NDNF domain partially overlaps with the first FN3 but shows some additional conservation that extends between the two FN3 domains (*Figure 3A*). The FN3 module is quite diverse in sequence but is thought to exhibit a common fold that is used as an interaction surface or spacer (*Campbell and Spitzfaden, 1994*; *Koide et al., 1998*). The function of the NDNF domain is not clear, but it may also provide a protein interaction surface.

Although the vertebrate NDNFs are highly conserved throughout the entire protein length, the *Caenorhabditis elegans* and *Drosophila* relatives are quite divergent in primary sequence and show little conservation beyond a few key residues that define the second FN3 and NDNF domains (*Kuang et al., 2010*). Notably, the *Drosophila* protein is missing the first FN3 domain, and therefore it is not clear the extent to which Nord and the vertebrate NDNFs may exhibit functional conservation. Ironically, the original human NDNF clone was identified on the basis of domain structure conservation with *Drosophila* Nord, which was identified via enhancer trapping to be a gene expressed in mushroom bodies and whose loss leads to defects in olfactory learning and memory (*Dubnau et al., 2003*). Unfortunately, that particular *LacZ* enhancer trap line that disrupted the *nord* locus is no longer available. The use of our new alleles should prove helpful for either confirming or eliminating the involvement of Nord as a modulator of learning and memory and/or other neuronal functions in larva and adult *Drosophila*.

In the mouse, NDNF is highly expressed in many neurons of the brain and spinal cord (*Boyle et al., 2011*; *Kuang et al., 2010*; *Schuman et al., 2019*). Studies using cultured mouse hippocampal neurons revealed that it promotes neuron migration and neurite outgrowth, hence its name (*Kuang et al., 2010*). In later studies, NDNF was also found to be upregulated in mouse endothelial cells in response to hindlimb ischemia, where it promotes endothelial cell and cardiomyocyte survival through integrin-mediated activation of AKT/endothelial NOS signaling (*Ohashi et al., 2014*; *Joki et al., 2015*). Additionally, recent studies have shown that NDNF expression is significantly downregulated in human

lung adenocarcinoma (LUAD) and renal cell carcinoma (RCC), indicating that NDNF may also provide a beneficial function as a tumor suppressor (*Xia et al., 2019*; *Zhang et al., 2019*).

Taken together, these studies have suggested some possible functions for vertebrate NDNF. However, they have primarily relied on *in vitro* cell culture models, and only recently have *in vivo* loss-of-function studies been reported (*Messina et al., 2020*). Remarkably, NDNF mutants were discovered in the genomes of several probands with congenital hypogonadotropic hypogonadism (CHH), a rare genetic disorder that is characterized by absence of puberty, infertility, and anosmia (loss of smell) (*Boehm et al., 2015*; *Lima Amato et al., 2017*). This phenotype is very similar to that produced by loss of the *anos1*, which also encodes an FN3 superfamily member and is responsible for Kallmann syndrome, a condition that similarly presents with CHH and anosmia due to lack of proper GnRH and olfactory neuron migration (*Stamou and Georgopoulos, 2018*). Although *in vitro* studies indicated that NDNF modulates FGFR1 signaling after FGF8 stimulation, the *in vivo* molecular mechanism responsible for the neuronal migration defects is not clear (*Messina et al., 2020*). The results of our study on the function of *Drosophila* Nord raise the issue of whether any of the ascribed vertebrate NDNF functions could involve alterations in BMP signaling. In the case of angiogenesis and EMT, BMPs, as well as other TGF-β family members, participate at many levels (*Goumans et al., 2018*; *Kahata et al., 2018*). At present, however, no involvement of BMP or TGF-β signaling has been implicated in migration of the GnRH neurons, although BMP signaling does define neurogenic permissive areas in which the olfactory placode forms (*Forni and Wray, 2015*). A clear objective for the future is to determine if the vertebrate NDNF factors bind BMPs and/or HSPG proteins such as Dally-like glypicans to modulate BMP signaling activity. On the *Drosophila* side, additional non-BMP-modulating roles for Nord should also be examined.

# Materials and methods

## Key resources table

| Reagent type (species) or resource | Designation | Source or reference | Identifiers | Additional information |
|---|---|---|---|---|
| Genetic reagent (*Drosophila melanogaster*) | hs-FLP | *Golic and Lindquist, 1989* | FBti0000785 | Chr X |
| Genetic reagent (*D. melanogaster*) | Actin>y+>Gal4 | *Ito et al., 1997* | FBti0009983 | Chr 2 |
| Genetic reagent (*D. melanogaster*) | ptc-Gal4 | *Hinz et al., 1994* | FBal0287777 | Chr 2 |
| Genetic reagent (*D. melanogaster*) | hh-Gal4 | BDSC | RRID:BDSC_67046 | Chr 3 |
| Genetic reagent (*D. melanogaster*) | en-Gal4 | BDSC | RRID:BDSC_30564 | Chr 2 |
| Genetic reagent (*D. melanogaster*) | nub-Gal4 | BDSC | RRID:BDSC_25754 | Chr 2 |
| Genetic reagent (*D. melanogaster*) | MS1096-Gal4 | BDSC | RRID:BDSC_8860 | Chr X |
| Genetic reagent (*D. melanogaster*) | MS1096-Gal4; UAS-Dcr-2 | BDSC | RRID:BDSC_25706 | Chr X; Chr 2 |
| Genetic reagent (*D. melanogaster*) | A9-Gal4 | BDSC | RRID:BDSC_8761 | Chr X |
| Genetic reagent (*D. melanogaster*) | hs-Gal4 | *Halfon et al., 1997* | FBtp0065595 | Chr 2 |
| Genetic reagent (*D. melanogaster*) | tub-Gal80$^{ts}$ | BDSC | RRID:BDSC_7019 | Chr 2 |
| Genetic reagent (*D. melanogaster*) | tub-Gal80$^{ts}$ | BDSC | RRID:BDSC_7018 | Chr 3 |
| Genetic reagent (*D. melanogaster*) | UAS-Dicer-2 | BDSC | RRID:BDSC_24650 | Chr 2 |

*Continued on next page*

*Continued*

| Reagent type (species) or resource | Designation | Source or reference | Identifiers | Additional information |
|---|---|---|---|---|
| Genetic reagent (*D. melanogaster*) | UAS-Dicer-2 | BDSC | RRID:BDSC_24651 | Chr 3 |
| Genetic reagent (*D. melanogaster*) | UAS-GFP | BDSC | RRID:BDSC_1521 | Chr 2 |
| Genetic reagent (*D. melanogaster*) | UAS-GFP | BDSC | RRID:BDSC_1522 | Chr 3 |
| Genetic reagent (*D. melanogaster*) | UAS-mCD8-GFP | BDSC | RRID:BDSC_5137 | Chr 2 |
| Genetic reagent (*D. melanogaster*) | UAS-DsRed | BDSC | RRID:BDSC_6282 | Chr 3 |
| Genetic reagent (*D. melanogaster*) | UAS-Ptc | *Johnson et al., 1995* | | Chr 3 |
| Genetic reagent (*D. melanogaster*) | UAS-Hh | *Lee et al., 1992* | | Chr 3 |
| Genetic reagent (*D. melanogaster*) | UAS-SmoGlu | *Zhang et al., 2004* | | Chr 3 |
| Genetic reagent (*D. melanogaster*) | UAS-nord-RNAi-1 | VDRC | RRID:VDRC_v38151 | Chr 2 |
| Genetic reagent (*D. melanogaster*) | UAS-nord-RNAi-2 | VDRC | RRID:VDRC_v38152 | Chr 2 |
| Genetic reagent (*D. melanogaster*) | w[1,118]; Df(2R)BSC770/SM6a | BDSC | RRID:BDSC_26867 | Chr 2 |
| Genetic reagent (*D. melanogaster*) | [1,118]; Df(2 R)BSC356/SM6a | BDSC | RRID:BDSC_24380 | Chr 2 |
| Genetic reagent (*D. melanogaster*) | y(1) w[1,118]; Df(2R)BSC155/CyO-Df(2R)B80, y[+] | BDSC | RRID:BDSC_9691 | Chr 2 |
| Genetic reagent (*D. melanogaster*) | w[1,118]; Df(2R)BSC780/SM6a | BDSC | RRID:BDSC_27352 | Chr 2 |
| Genetic reagent (*D. melanogaster*) | w[1,118]; Df(2R)BSC603/SM6a | BDSC | RRID:BDSC_25436 | Chr 2 |
| Genetic reagent (*D. melanogaster*) | w[1,118]; Df(2R)ED4061, P{w[+ mW.Scer\FRT.hs3] = 3'. RS5 + 3.3'}ED4061/SM6a | BDSC | RRID:BDSC_9068 | Chr 2 |
| Genetic reagent (*D. melanogaster*) | w1118; PBac{y[+ mDint2]=vas-Cas9}VK00027 | BDSC | RRID:BDSC_51324 | Chr 3 |
| Genetic reagent (*D. melanogaster*) | y w | BDSC | RRID:BDSC_1495 | Chr X |
| Genetic reagent (*D. melanogaster*) | y w; Mi{MIC}nord$^{MI06414}$ | BDSC | RRID:BDSC_42389 | Chr 2 |
| Genetic reagent (*D. melanogaster*) | y w; Mi{PT-GFSTF.2}nord$^{MI06414-GFSTF.2}$/CyO | BDSC | RRID:BDSC_60250 | Chr 2 |
| Genetic reagent (*D. melanogaster*) | w$^{1118}$; PBac{y[+ mDint2]=vas-Cas9}VK00027 | BDSC | RRID:BDSC_51324 | Chr 3 |
| Genetic reagent (*D. melanogaster*) | y w; Mi{PT-RFPHA.2}nord$^{MI06414-RFPHA.2}$ | This paper | | Chr 2 |
| Genetic reagent (*D. melanogaster*) | UAS-Nord-HA-GFP (Chr.2) | This paper | | Chr 2 |
| Genetic reagent (*D. melanogaster*) | UAS-Nord-HA-GFP (Chr.3) | This paper | | Chr 3 |
| Genetic reagent (*D. melanogaster*) | w; nord$^{p2A}$ | This paper | | Chr 2 |

*Continued on next page*

*Continued*

| Reagent type (species) or resource | Designation | Source or reference | Identifiers | Additional information |
|---|---|---|---|---|
| Genetic reagent (*D. melanogaster*) | *w; nord³ᴰ* | This paper | | Chr 2 |
| Antibody | Anti-Ci (rat monoclonal) | DSHB | Cat# 2A1; RRID:AB_2109711 | IF: (1:50) |
| Antibody | Anti-Ptc (mouse monoclonal) | DSHB | Cat# Apa 1; RRID:AB_528441 | IF: (1:50) |
| Antibody | Anti-Dpp prodomain (rabbit polyclonal) | *Akiyama and Gibson, 2015* | A gift from M. Gibson | IF: (1:100) |
| Antibody | Anti-GFP (rabbit polyclonal) | Molecular Probes | Cat# A-11122, RRID:AB_221569 | IF: (1:2000) |
| Antibody | Anti-GFP (chicken polyclonal) | Abcam | Cat# ab13970, RRID:AB_300798 | IF: (1:2000) |
| Antibody | Anti-beta tubulin (mouse monoclonal) | DSHB | Cat#E7; RRID:AB_2315513 | WB: (1:5000) |
| Antibody | Anti-HA.11 (mouse monoclonal, 16B12) | Covance | Cat# MMS-101P-1000; RRID: AB_291259 | IF: (1:1000) |
| Antibody | Anti-HA (rabbit polyclonal, SG77) | Thermo Fisher Scientific | Cat# 71-5500; RRID:AB_2533988 | WB: (1:1000) |
| Antibody | Anti-FLAG (mouse monoclonal, M2) | Sigma-Aldrich | Cat# F3165; RRID:AB_259529 | WB: (1:200) |
| Antibody | Anti-FLAG (rabbit polyclonal) | Sigma-Aldrich | Cat# F7425; RRID:AB_439687 | WB: (1:200) |
| Antibody | Anti-Myc (mouse monoclonal) | Santa Cruz Biotechnology | Cat# sc-40; RRID:AB_627268 | WB: (1:200) |
| Antibody | Anti-anti-phospho-Smad1/5 (rabbit polyclonal, Ser463/465) | Cell Signaling Technology | Cat# 9516; RRID:AB_491015 | IF: (1:100) |
| Antibody | Anti-FLAG (mouse monoclonal, M2) Magnetic Beads | Sigma-Aldrich | Cat# M8823; RRID:AB_2637089 | IP |
| Antibody | Anti-HA (rat monoclonal, 3F10) Affinity Matrix | Roche | Cat#11815016001;RRID:AB_390914 | IP |
| Antibody | Fluorophore-conjugated secondary antibodies | Jackson ImmunoResearch Lab | NA | IF: (1:500) |
| Antibody | HRP-conjugated secondary antibodies | Jackson ImmunoResearch Lab | NA | WB: (1:10,000) |
| Other | DAPI | MilliporeSigma | Cat# D9542 | (1 μg/mL) |
| Other | Fetal bovine serum | Omega Scientific | Cat# FB-02 | |
| Other | Schneider's *Drosophila* Media | Invitrogen | Cat# 21720 | |
| Other | Dulbecco's Modification of Eagle's Medium (DMEM) | Corning | Cat# 10-013CM | |
| Other | Blasticidin S HCl | Thermo Fisher Scientific | Cat# R21001 | |
| Other | Penicillin-Streptomycin-Glutamine (100×) | Thermo Fisher Scientific | Cat# 10378016 | |
| Other | Antifade mounting media | VECTASHIELD | Cat# H-1000 | |
| Other | FuGENE HD transfection reagent | Promega | Cat# E2311 | |
| Other | 16% paraformaldehyde aqueous solution | Electron Microscopy Sciences | Cat# 15710 | |
| Other | Dissociation buffer | Sigma | Cat# C-1544 | |
| Other | Recombinant Dpp | R&D Systems | Cat# 59-DP-020 | |
| Other | Elastase | Sigma | Cat# E-0258 | |
| Other | Propidium iodide | Invitrogen | Cat# P3566 | |
| Other | DNAse I | Thermo Fisher Scientific | Cat# AM2222 | |
| Other | Maxima Reverse Transcriptase | Thermo Fisher Scientific | Cat# EP0742 | |
| Other | SYBR Green Supermix | Bio-Rad | Cat# 1708880 | |
| Other | RNeasy Mini | QIAGEN | Cat# 74104 | |

| Reagent type (species) or resource | Designation | Source or reference | Identifiers | Additional information |
|---|---|---|---|---|
| Recombinant DNA reagent | pAcSV-Nord-GFP | *Supplementary file 3* | | Nord coding sequence (NM_138056) was fused in frame with C-terminal GFP tag and cloned into pAcSV vector |
| Recombinant DNA reagent | pcDNA3.1-Nord-HisMyc | *Supplementary file 3* | | Nord coding sequence (NM_138056) was fused in frame with C-terminal HisMyc tag and cloned into pcDNA3.1 vector |
| Recombinant DNA reagent | pcDNA3.1-HsNDNF-HisMyc | *Supplementary file 3* | | HsNDNF coding sequence (NM_024574) was fused in frame with C-terminal HisMyc tag and cloned into pcDNA3.1 vector |
| Recombinant DNA reagent | pcDNA3.1-CeNdnf-HisMyc | *Supplementary file 3* | | CeNdnf coding sequence (NM_067881) was fused in frame with C-terminal HisMyc tag and cloned into pcDNA3.1 vector |
| Recombinant DNA reagent | pcDNA3.1-DrNdnf-HisMyc | *Supplementary file 3* | | DrNdnf coding sequence (XM_684842) was fused in frame with C-terminal HisMyc tag and cloned into pcDNA3.1 vector |
| Recombinant DNA reagent | pcDNA3.1-XtNdnf-HisMyc | *Supplementary file 3* | | XtNdnf coding sequence (NM_001122800) was fused in frame with C-terminal HisMyc tag and cloned into pcDNA3.1 vector |
| Recombinant DNA reagent | pBRAcpA- Dpp-FLAG | *Supplementary file 3* | | |
| Recombinant DNA reagent | pBRAcpA- Gbb-FLAG | *Supplementary file 3* | | |
| Recombinant DNA reagent | pBRAcpA-Gbb-HA | *Supplementary file 3* | | |
| Recombinant DNA reagent | pBRAcpA-FLAG-Mad | *Supplementary file 3* | | |
| Sequence-based reagent | Primer: *pkg* Forward: GTCCCAAGACCCGTGAGCTCTTCGC | This paper | | Primers used in *Figure 1* for quantitative reverse transcription PCR |
| Sequence-based reagent | Primer: *pkg* Reverse: TCCGTGTTCCACTTGGCGCAGCAAG | This paper | | Primers used in *Figure 1* for quantitative reverse transcription PCR |
| Sequence-based reagent | Primer: *ci* Forward: CGACCACCAGGAGGAAGTAT | This paper | | Primers used in *Figure 1* for quantitative reverse transcription PCR |
| Sequence-based reagent | Primer: *ci* Reverse: AATCGGAATAAGGCGATGAC | This paper | | Primers used in *Figure 1* for quantitative reverse transcription PCR |
| Sequence-based reagent | Primer: *hh* Forward: GGATTCGATTGGGTCTCCTA | This paper | | Primers used in *Figure 1* for quantitative reverse transcription PCR |
| Sequence-based reagent | Primer: *hh* Reverse: GAATCTGACTTGACGGAGCA | This paper | | Primers used in *Figure 1* for quantitative reverse transcription PCR |

*Continued on next page*

*Continued*

| Reagent type (species) or resource | Designation | Source or reference | Identifiers | Additional information |
|---|---|---|---|---|
| Sequence-based reagent | Primer: *ptc* Forward: CGATGTGGTCGATGAGAAAT | This paper | | Primers used in *Figure 1* for quantitative reverse transcription PCR |
| Sequence-based reagent | Primer: *ptc* Reverse: CGAGGTGGGACTGGAATACT | This paper | | Primers used in *Figure 1* for quantitative reverse transcription PCR |
| Sequence-based reagent | Primer: *nord* Forward: CACCGCAAAAGTGTCCTTCG | This paper | | Primers used in *Figure 1* for quantitative reverse transcription PCR |
| Sequence-based reagent | Primer: *nord* Reverse: CAGGTTCAGCACAAATCGCT | This paper | | Primers used in *Figure 1* for quantitative reverse transcription PCR |
| Sequence-based reagent | Primer: *ptc* Forward: atggaccgcgacagcctccca | *Hsia et al., 2017* | Anti-sense probe (600 bp) for *ptc* | Primers used in *Figure 1* for generating *in situ* probes |
| Sequence-based reagent | Primer: *ptc* Reverse: TAATACGACTCACTATAGGG gaggtggcgcaggatctgctc | *Hsia et al., 2017* | Anti-sense probe (600 bp) for *ptc* | Primers used in *Figure 1* for generating *in situ* probes |
| Sequence-based reagent | Primer: *nord* Forward: gaaatccgggtgaagctgctacg | This paper | Anti-sense probe (666 bp) for *nord* | Primers used in *Figure 1* for generating *in situ* probes |
| Sequence-based reagent | Primer: *nord* Reverse: TAATACGACTCACTATAGGG atgcagcgaagctttgggtatgg | This paper | Anti-sense probe (666 bp) for *nord* | Primers used in *Figure 1* for generating *in situ* probes |
| Sequence-based reagent | P1: nord exon 1 Forward: GCAAGTGGCAAGAGCTGAAC | This paper | | Primers used in *Figure 2— figure supplement 1* for confirming the predicted transcripts from different Nord alleles |
| Sequence-based reagent | P2: nord exon 1/2 Reverse: GTGTTCTGCGGTTTTGCCTG | This paper | | Primers used in *Figure 2— figure supplement 1* for confirming the predicted transcripts from different Nord alleles |
| Sequence-based reagent | P3: nord exon 2 Forward: CGCACTCAGAGGTTGTTTCA | This paper | | Primers used in *Figure 2— figure supplement 1* for confirming the predicted transcripts from different Nord alleles |
| Sequence-based reagent | P4: nord exon 4/5 Reverse: GCTCCTTTCCCACTTGACGA | This paper | | Primers used in *Figure 2— figure supplement 1* for confirming the predicted transcripts from different Nord alleles |
| Sequence-based reagent | P5: nord exon 6 Forward: AGGCTCTGTTCCGGGATTTG | This paper | | Primers used in *Figure 2— figure supplement 1* for confirming the predicted transcripts from different Nord alleles |
| Sequence-based reagent | P6: nord exon 8 Reverse: AAATGCAGCGAAGCTTTGGG | This paper | | Primers used in *Figure 2— figure supplement 1* for confirming the predicted transcripts from different Nord alleles |

*Continued on next page*

*Continued*

| Reagent type (species) or resource | Designation | Source or reference | Identifiers | Additional information |
|---|---|---|---|---|
| Sequence-based reagent | Gapdh Forward: GCCACCTATGACGAAATCAAGGCTA | This paper | | Primers used in *Figure 2—figure supplement 1* for confirming the predicted transcripts from different Nord alleles |
| Sequence-based reagent | Gapdh Reverse: GGAGTAACCGAACTCGTTGTCGTAC | This paper | | Primers used in *Figure 2—figure supplement 1* for confirming the predicted transcripts from different Nord alleles |
| Sequence-based reagent | 5'-GGACCTGTTCGGAATCCACC-3' | This paper | | Guide RNA sequence used to generate different Nord alleles using CRISPR/Cas9 system |
| Sequence-based reagent | 5'-GGGTGAGGTTCTGTCTACCC-3 | This paper | | Guide RNA sequence used to generate different Nord alleles using CRISPR/Cas9 system |
| Cell line (*D. melanogaster*) | *S2* | DGRC | S2-DGRC | |
| Cell line (*Homo sapiens*) | *embryonic kidney cell line HEK 293* | ATCC | CRL-1573 | |
| Software, algorithm | Fiji | NIH | RRID:SCR_002285 | |
| Software, algorithm | GraphPad Prism | GraphPad software | RRID:SCR_002798 | |

## *Drosophila* maintenance

Animals were grown on standard food containing molasses at room temperature unless otherwise indicated. The *hs-FLP* and *actin>y+>*Gal4 (*Ito et al., 1997*) driver was used to generate random flip-out clones expressing various *UAS-transgenes*. The *hs-Gal4* was used to induce random ectopic expression of *UAS-Hh* or *UAS-Ptc* (*Figure 2—figure supplement 3*). Larvae of the corresponding genotypes were incubated at 37°C for 30–60 min during the second instar larval stage (*Figure 2*, *Figure 2—figure supplement 3*) or 15–20 min in the mid-third larval stage (*Figure 3*) to induce flip-out clones. Wing imaginal discs were dissected from the larvae containing flip-out clones or *hs-Gal4*-expressing clones at the wondering larva stage. The *hh-Gal4* or *en-Gal4* driver together with *tub-Gal80^{ts}* (*McGuire et al., 2003*) was used for transient expression of transgenic constructs. Fly crosses, embryos, and larvae were maintained at 18°C, and the *Gal80^{ts}* repressor was inactivated for the indicated number of hours at restrictive temperature (29°C) before adult fly eclosion or dissection (see *Figure 7* legend for details). The genotypes of larvae, pupae, or adult flies used in each figure are listed in *Supplementary file 3*. *Drosophila* stocks used in this study are listed in the Key resources table.

## Dissociation and sorting of imaginal disc cells

Wing imaginal discs were dissected from wandering third instar larvae of the genotypes *hh-Gal4; UAS-mCD8-GFP* or *ptc-Gal4; UAS-mCD8-GFP*. Discs were stored in Schneider's *Drosophila* Media (21720, Invitrogen) plus 10% fetal bovine serum (FBS) (10438, Invitrogen) on ice for less than 2 hr prior to cell dissociation. Discs were washed twice with 1 mL cell dissociation buffer (Sigma, C-1544). Elastase (Sigma, E-0258) was diluted to 0.4 mg/mL in fresh cell dissociation buffer once discs were ready. Discs were incubated for 20 min at room temperature in 0.4 mg/mL elastase with stirring by a magnetic micro-stir bar. Undissociated tissue was spun out, cell viability was measured (>80%), and cells were immediately isolated using the BD FACSAria system. Dead cells labeled with propidium iodide (P3566, Invitrogen) were excluded during FACS, and purity of sorted cells was greater than 99% by post-sorting FACS analysis. Total RNA was extracted from sorted cells (RNeasy, QIAGEN) and stored at −80°C. Quality was assessed with the Agilent Bioanalyzer 2100 (RIN > 7.0).

## Identification of target genes of the Hh signaling pathway

As described in *Willsey et al., 2016*, using total RNA extracted from sorted A (*hh-*), B (*ptc+*), and P (*hh+*) cells (see details in 'Dissociation and sorting of imaginal disc cells'), we acquired three primary

transcriptome datasets via the Affymetrix *D. mel* GeneChip Genome 2.0 microarrays. The raw microarray data were deposited to the Gene Expression Omnibus public repository (https://www.ncbi.nlm.nih.gov/geo/query/acc.cgi?acc=GSE180120; Gene Expression Omnibus series no. GSE180120). In our previous analysis (*Willsey et al., 2016*), we focused on two datasets and searched for genes differentially expressed in the A/P boundary adjacent cells (B: *ptc+*) and P cells (P: *hh+*). Here, we modified the gene expression analysis method by including the third dataset from A cells (*hh-*) and additional transcriptome comparisons between the A/P boundary adjacent B cells (*ptc+*) and A cells (*hh-*), and between A cells (*hh-*) and P cells (*hh+*).

Briefly, all analyses were conducted in R version 4.0.2. Expression values were determined using the affy package (*Gautier et al., 2004*), available from BioConductor (http://bioconductor.org). The *Drosophila* 2.0 CDF environment was utilized. Probe-level data from the CEL files were imported using the function *ReadAffy* and converted to expression values using *rma* with default settings. This method implemented the multi-array average (RMA) for background correction followed by quantile normalization. Expression values were $\log_2$ transformed. Probe sets were mapped to genes using the *Drosophila*_2.na30.annot.xml annotation file, available from the Affymetrix website. 14,448 of 18,952 (76.2%) probe sets map to gene isoforms—13,016 (90.1%) of which correspond to unique genes (some genes are mapped by ≥1 probe set). Probe sets mapping to the same gene were not combined to minimize technical artifacts. Genes (probe sets) whose expression is not only higher in A cells than P cells ($Fold_{A/P} > 1.2$), but also higher in the A/P boundary adjacent cells than general A cells ($Fold_{B/A} > 1.5$), were selected as potential Hh-induced target genes (*Figure 1*). A total of 61 probe sets (59 unique genes) were identified as potential Hh signaling target genes (*Supplementary file 1*). The heatmap was generated in R with the ggplot2 package, and the genes were ordered by the $Fold_{B/A}$ change.

## Quantitative reverse transcription PCR

Total RNA was extracted from FACS-sorted A, B, and P cells (see details in 'Dissociation and sorting of imaginal disc cells'). Possible contamination of genomic DNA was excluded by treatments of DNAse I (AM2222, Thermo Fisher Scientific). RNA was reverse-transcribed to cDNA using Maxima Reverse Transcriptase (EP0742, Thermo Fisher Scientific) with random hexamers. All samples within an experiment were reverse-transcribed at the same time; the resulting cDNA was stored in aliquots at –80°C until used. cDNA was PCR-amplified using SYBR Green Supermix (1708880; Bio-Rad). qPCR was carried out with an ABI PRISM Sequence Detection System (Applied Biosystems). Reactions were run in triplicate in three independent experiments. Expression data were normalized to the geometric mean of the housekeeping gene *pkg* and were analyzed using the 2–ΔΔCT method. The primer sequences are provided in the Key resources table.

## *In situ* hybridization

*In situ* hybridization of wing discs was performed as previously described (*Hsia et al., 2017*). Briefly, RNA probes were created from *in vitro* transcription of PCR products carrying the T7 RNA polymerase recognition sequence at one end and synthesized by using a digoxigenin (Dig)-labeling kit (Roche). Wing discs of L3 larvae were hybridized with probes overnight at 56°C using standard procedures and visualized using anti-Dig-AP (1:1000; Roche). Primers used for generating PCR templates are listed in the Key resources table.

## Generation of Nord Crispr alleles

The *nord³ᴰ* and *nord²²ᴬ* alleles were generated using the Crispr/Cas9 system. The following guides 5'-GGACCTGTTCGGAATCCACC-3' and 5'-GGGTGAGGTTCTGTCTACCC-3 were separately cloned into the BbsI site of pU6-BbsI-chiRNA plasmid (obtained from Addgene) and both were simultaneously injected by Best Gene into *w¹¹¹⁸; PBac{y[+ mDint2]=vas-Cas9}VK00027* on chromosome 3 (Bloomington Stock Center #51324). G0 flies were crossed to a balancer stock (*w; Pin/Cyoˢᵗᵃʳ*) and then individual males were crossed to *w; Gla, Bc/CyO{GFP}* to establish stocks. DNA from homozygous adults was amplified by PCR using primers that flanked the two Crispr target sites and sequenced. The nord²²ᴬ allele was a 5 bp deletion generated at guide sequence 2 site, while the nord³ᴰ allele was an 11 bp deletion generated at the guide 1 site.

## Cell culture and transfection

*Drosophila* S2 cells (S2- DGRC) were obtained directly from the Drosophila Genomics Resource Center (DGRC) and cultured in *Drosophila* Schneider's medium supplemented with 10% of FBS (Omega Scientific) and 1% Penicillin-Streptomycin-Glutamine (Thermo Fisher) at 25°C in a humidified incubator. S2 cells stably expressing the FLAG-Mad transgene (Mad-S2) cells were generated by co-transfecting pBRAcpA-FLAG-Mad (*Jensen et al., 2009*) and pCoBlast, and then followed by selection with 12.5 µg/mL blasticidin. HEK 293 cells were obtained directly from ATCC and cultured in Dulbecco's Minimal Essential Medium with 10% FBS (Omega Scientific) and 1% Penicillin-Streptomycin-Glutamine (Thermo Fisher) at 37°C in a humidified incubator with 5% $CO_2$. Transfection was performed with FuGENE 6 transfection reagent (Promega). All the cell lines were regularly confirmed to be free of contamination (e.g., mycoplasma) through PCR-based tests as recommended by the NIH.

## Antibodies

Antibodies and dilutions used were mouse anti-Ptc antibody 1:50 (DSHB, Apa1; 1/50); rat anti-Ci antibody 1:50 (DSHB, 2A1; 1/50); rabbit anti-Dpp prodomain (*Akiyama and Gibson, 2015*; 1/100); rabbit anti-GFP (Invitrogen, A-11122; 1/1000); chicken anti-GFP (Abcam Cat# ab13970; 1/2000); rabbit anti-phospho-Smad1/5 (Ser463/465) (Cell Signaling Technology, 9516; 1/100); mouse anti-HA.11 (Covance, Cat#101P; 1/1000); rabbit anti-HA (Thermo Fisher Scientific, Cat# 71-5500; 1/1000); mouse anti-β-tubulin (DSHB, E7; 1/5000); mouse anti-β-galactosidase (Promega, Z378A; 1/100); mouse anti-Myc (Santa Cruz Biotechnology, 9E10; 1/200); mouse anti-FLAG (Sigma-Aldrich, M2; 1/200); rabbit anti-FLAG (Sigma-Aldrich, Cat# F7425; 1/200); HRP-conjugated and fluorophore-conjugated secondary antibodies were from Jackson ImmunoResearch Lab and Thermo Fisher. The antibody information is also listed in the Key resources table.

## Constructs

The coding sequence of Nord or Ndnf from various species was fused in frame with C-terminal GFP tag or HisMyc tag, and cloned into the pAcSV vector for expression in the *Drosophila* S2 cells, or into the pcDNA3.1 vector for expression in the HEK 293 cells. The coding sequence of Nord was fused with a C-terminal HA-GFP tag and cloned into the pUAST vector to generate the transgenic *UAS-Nord* line. Constructs expressing Dpp-FLAG, Gbb-FLAG, Gbb-HA, and FLAG-Mad were generated using PCR methods to tag and amplify the gene of interest from a full-length cDNA and then cloned into the S2 cell constitutive expression vector pBRAcpA (*Cherbas et al., 1994*). The sequences are provided in *Supplementary file 4*.

## Imaginal discs and pupal wing immunostaining and imaging

Wing discs from third instar larvae were dissected, fixed in 4% formaldehyde in PBS, blocked and permeabilized by 5% normal goat serum (NGS) and 0.3% Triton X-100 in PBS, incubated with primary antibody in PBS containing 5% NGS and 0.3% Triton X-100 overnight at 4°C, washed three times with 0.3% Triton X-100/PBS, incubated with secondary antibody, and washed with 0.3% Triton X-100/PBS. To selectively stain the secreted Nord (*Figure 3*), the above immunostaining procedure was carried out in the absence of Triton X-100 (PBS alone for blocking and antibody incubation buffer; 0.01% Tween-20/in PBS for washing buffer). Pupal wings were collected and pre-fixed as previously described (*Classen et al., 2008*), then followed by the procedure described above for immunostaining of the larval wing discs. The stained larval wing discs or pupal wings were mounted and imaged with a Zeiss spinning disc confocal microscope.

## Image collection and quantification of fluorescence intensity

To compare the expression profile of pMad, Dpp, or Ptc in different genotypes, we used wing imaginal discs at the same developmental stage, which were dissected from wandering larvae, corresponding to 1–6 hr before the entry into the pupal stage. Larvae from the control and corresponding experimental group were raised at the same temperature and density. Wing discs were dissected, fixed, immunostained, and mounted by following the same protocol. All images were taken using the same confocal microscope settings. The pixel intensities of pMad, Dpp, or Ptc were obtained within a fixed rectangular region across both ventral and dorsal compartments using the Plot Profile function of Fiji. Then, the average pixel intensities from multiple discs were plotted using the GraphPad

Prism software. The number of wing imaginal discs used in each experiment is provided in the corresponding figure legend.

## Cell immunostaining

48 hr after transfection, NIH 293 cells were washed twice with PBS, fixed in 4% formaldehyde in PBS, blocked by 1.5% NGS in PBS, incubated with the primary antibody in PBS containing 1.5% NGS for overnight at 4° (to stain surface Nord or Ndnf), washed with 0.01% Tween-20/1× PBS, incubated with secondary antibody, and washed with 0.01% Tween-20/PBS. Mad-S2 cells immunostaining was carried out through similar procedures but in the presence of 0.1% Triton X-100 during blocking, antibody incubation, and washing steps. The stained cells were mounted and imaged with a Zeiss spinning disc confocal microscope.

## Immunoprecipitation assay

S2 cells were separately transfected to express Nord with a C-terminal GFP tag, or the FLAG-tagged Dpp, HA-tagged Gbb alone, or Dpp-FLAG/Gbb-HA in combination. >72 hr after transfection, conditioned medium from transfected cells were collected. The medium containing Nord or the BMP ligands were mixed, followed by incubation with anti-FLAG or anti-HA antibody-coupled beads (anti-M2 Affinity Matrix from Sigma; anti-HA Affinity Matrix from Roche) overnight at 4°C. Precipitated proteins were analyzed by Western blotting using anti-GFP, anti-HA, and anti-FLAG antibodies. Beads were washed five times with washing buffer (50 mM Tris-HCl at pH 6.8, 150 mM NaCl, and 1% NP40). Proteins bound to the beads were recovered in the SDS-PAGE sample buffer. Procedures from the medium collection were carried out at 4°C or on ice. Proteins samples were resolved by SDS-PAGE and transferred to PVDF membranes (Millipore) for Western blot analysis.

## S2 cell-based BMP signaling assay

The S2 cell-based BMP signaling assay was adopted from assays as previously described (*Shimmi and O'Connor, 2003*). Recombinant Dpp (159-DP-020, R&D Systems) was diluted in the culture medium according to the manufacturer's recommendations. To prepare BMP ligands secreting source cells, briefly, *Drosophila* S2 cells were transfected with plasmids to express Dpp-FLAG or/and Gbb-FLAG with or without Nord-HA-GFP. >72 hr after transfection, the conditioned medium was collected, and the cells were lysed in lysis buffer (50 mM Tris-HCl at pH 6.8, 150 mM NaCl, 1% NP40, and protease inhibitors). S2 cells stably expressing the FLAG-Mad transgene (Mad-S2) cells were generated and used as BMP responding cells. Alternatively, we also transiently transfected S2 cells or Mad-S2 cells with plasmids to express GFP or Nord-GFP as described in the figure legends. >48 hr after transfection, the responding cells were incubated with the conditioned medium collected from the source cells for 1 hr. After incubation, the responding cells were washed and then lysed in the lysis buffer. Both the conditioned medium and the lysates were clarified by centrifugation, and proteins were recovered directly in the SDS-PAGE sample buffer. Proteins were separated by SDS-PAGE under reducing conditions and then transferred onto PVDF membranes (Millipore). The membranes were blocked and immunostained with primary antibodies and HRP-conjugated secondary antibodies. Blots were developed using Immobilon Forte Western HRP substrate (MilliporeSigma, WBLUF0100) with GeneSys Imaging System (Syngene). In *Figure 8*, Western blot quantification was performed using the Fiji software. Following background subtraction, the intensity of each band was measured by the function of plot lanes (GelAnalyzer). Then, the band intensity was used to calculate the signaling activity from an equal volume of conditioned medium (pMad/Mad) or the relative ligand activity ([pMad/Mad]/BMPs) as indicated in the figure legend.

## Wing size and wing trichome measurements

Adult wings were dissected from animals in 100% ethanol and mounted in 1:1 Wintergreen oil-Canadian balsam medium. The wings were imaged using a ×4 objective and area was measured using Fiji (ImageJ). Measurements were taken from the end of the costa on the anterior portion of the wing hinge to the end of the alula on the posterior. Area in pixels squared was converted to millimeters squared with a calibration value determined using a hemocytometer under the ×4 objective. Wing trichomes from the dorsal wing surface were imaged with a ×40 objective in the region between veins

4 and 5 just distal to the PCV (*Figure 4—figure supplement 6*). Trichomes were counted manually within the imaged area (37,500 $\mu m^2$).

## Statistical analysis

All data in column graphs are shown as mean values with SD and plotted using GraphPad Prism software. As described in the figure legends, one-way ANOVA followed by Sidak's multiple comparison test, unpaired two-tailed *t*-test, or two-sided Fisher's exact test was used for statistical analysis. The sample sizes were set based on the variability of each assay and are listed in the figure legends. Independent experiments were performed as indicated to guarantee reproducibility of findings. Differences were considered statistically significant when p<0.01.

## Acknowledgements

We thank Gibson, MC, the Bloomington Drosophila Stock Center, the Vienna Drosophila RNAi Center, Kyoto Stock Center, Addgene, and the Developmental Studies Hybridoma Bank for fly strains and reagents. MBO thanks Pierre Leopold in whose lab some of this work was carried out while he was on sabbatical. This work was supported by the National Institutes of Health grants R01GM117440 to XZ and R35GM118029 to MBO.

## Additional information

### Funding

| Funder | Grant reference number | Author |
| --- | --- | --- |
| National Institute of General Medical Sciences | R01GM117440 | Xiaoyan Zheng |
| National Institute of General Medical Sciences | R35GM118029 | Michael B O'Connor |

The funders had no role in study design, data collection and interpretation, or the decision to submit the work for publication.

### Author contributions

Shu Yang, Data curation, Formal analysis, Resources, Software, Validation, Visualization, Writing – original draft, Writing – review and editing; Xuefeng Wu, Data curation, Formal analysis, Visualization, Writing – original draft, Writing – review and editing; Euphrosyne I Daoutidou, Ya Zhang, Data curation, Formal analysis, Visualization; MaryJane Shimell, Data curation, Visualization; Kun-Han Chuang, Data curation, Formal analysis; Aidan J Peterson, Data curation, Validation, Writing – review and editing; Michael B O'Connor, Conceptualization, Data curation, Formal analysis, Funding acquisition, Methodology, Supervision, Validation, Visualization, Writing – original draft, Writing – review and editing; Xiaoyan Zheng, Conceptualization, Funding acquisition, Investigation, Methodology, Project administration, Supervision, Validation, Visualization, Writing – original draft, Writing – review and editing

### Author ORCIDs

Ya Zhang http://orcid.org/0000-0001-9060-3777
Kun-Han Chuang http://orcid.org/0000-0003-3241-8586
Aidan J Peterson http://orcid.org/0000-0002-6801-3364
Michael B O'Connor http://orcid.org/0000-0002-3067-5506
Xiaoyan Zheng http://orcid.org/0000-0003-4983-5503

### Decision letter and Author response

Decision letter https://doi.org/10.7554/eLife.73357.sa1
Author response https://doi.org/10.7554/eLife.73357.sa2

## Additional files

### Supplementary files

• Supplementary file 1. Genes that are differentially expressed in A, B, and P cells.
Genes (probe sets) whose expression is not only higher in A cells than P cells ($Fold_{A/P} > 1.2$), but also higher in the A/P boundary adjacent B cells than general A cells ($Fold_{B/P} > 1.5$) were selected as potential Hedgehog (Hh)-induced target genes. A total of 61 probe sets (59 unique genes) were identified as potential Hh signaling target genes.

• Supplementary file 2. Quantification of the posterior crossvein (PCV) phenotypes associated with trans-heterozygotic combinations of $Mi\{MIC\}nord^{MI06414}$ and tested deficiency lines.

• Supplementary file 3. The genotype of larvae, pupae, or adult flies from where wing discs, pupal, or adult wings were collected and imaged in each figure.

• Supplementary file 4. The nucleotide sequences used to express Nord or various Neuron-Derived Neurotrophic Factor (NDNF) fusion proteins.

• Transparent reporting form

### Data availability

The raw microarray data were deposited to the Gene Expression Omnibus public repository (https://www.ncbi.nlm.nih.gov/geo/query/acc.cgi?acc=GSE180120; Gene Expression Omnibus series no. GSE180120).

The following dataset was generated:

| Author(s) | Year | Dataset title | Dataset URL | Database and Identifier |
|---|---|---|---|---|
| Zheng X | 2021 | Genome-wide expression profiling in Drosophila wing imaginal discs | https://www.ncbi.nlm.nih.gov/geo/query/acc.cgi?acc=GSE180120 | NCBI Gene Expression Omnibus, GSE180120 |

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
