## [Editor Report]

This manuscript is of broad interest to readers in the field of Bone Morphogenetic Proteins (BMPs) and *Drosophila* wing development. The identification of a Nord as an indirect regulator of BMP diffusion is a new contribution to our understanding of regulation of BMP function. A combination of genetics, transcriptomics, and biochemistry provide support for many of the conclusions in the paper.

---

## [Decision Letter]

**Decision letter after peer review:**

[Editors’ note: the authors submitted for reconsideration following the decision after peer review. What follows is the decision letter after the first round of review.]

Thank you for choosing to send your work, "The NDNF-like factor Nord is a Hedgehog-induced BMP modulator that regulates *Drosophila* wing patterning and growth", for consideration at *eLife*. Your submission has been assessed by a Senior Editor in consultation with a member of the Board of Reviewing Editors. Although the work is of interest, we regret to inform you that the findings at this stage are too preliminary for further consideration at *eLife*.

Specifically, the reviewers felt that many experiments would be needed to better characterize the nord mutant phenotype, particularly its role in BMP signaling. They wanted the authors to:

1. Measure Dpp levels in nord mutants. This could be accomplished using the anti-pro-Dpp, or enhancer traps that are more accurate than BS3.0 (which not only lack the shortvein enhancers but is overly sensitive to anterior en).

2. Measure Ptc levels in nord mutants.

3. Show a cell autonomous effect of Hh signal reception on Nord expression.

4. Examine pMad at earlier stages of PCV development (20 hr AP) before EGFR and Notch signaling are active in this region.

5. Test whether the posterior defects they see are caused by loss of anterior Nord, or whether they might be caused by loss of low-level Nord expression in the posterior. In other words, what is the result when Nord is depleted from anterior or from posterior cells. Additionally, the authors should determine how far secreted Nord can spread in the disc.

7. Test how much of the PCV loss in en>Nord is caused by loss of the adjacent L5.

8. Add missing controls such as a nub-Gal4/+ 29°C control wing and a control disc stained with anti-GFP.

9. improve quantification for pMad analysis, particularly in Figure 4E, 4F and 6.

The reviewers also suggested extensive editorial changes that would make the manuscript easier for the reader to follow.

*Reviewer #1:*

The authors used cell sorting and genome-wide expression profiling to identify new Hedgehog (Hh) target genes in the developing *Drosophila* wing disc, which has been a powerful system to study Hh signaling. They identify nord, which encodes a secreted factor that associates with cell membranes and the extracellular matrix. Null mutants of nord are adult viable. Paradoxically, these animals have slightly smaller wings and ectopic posterior cross vein (PCV) material. Both phenotypes can be caused by alterations in BMP signaling. They demonstrate that recombinant Nord binds to Dpp and Gbb and restricts their release from cells and diffusion in the media. They use the Gal4/UAS mis-expression to test the model that Nord has both positive and negative effects on BMP signaling depending on where and when Nord is expressed.

The strengths of this paper are the appropriate use of techniques such as cell sorting, transcriptomics, *Drosophila* loss of function genetics, cell culture. These approaches support their model that Nord regulates availability of Dpp, Gbb and/or Dpp-Gbb dimers. However, the use of mis-expression of wild-type Nord in the pouch or posterior compartment lead similar phenotypes as global nord loss-of-function. This is a weakness because it is difficult for the reader to reconcile logically these paradoxical results. This work may impact the mammalian fields because Nord is conserved through evolution, but Nord homologues have not been implicated in BMP signaling.

a) While I appreciate that they authors have done a lot of work and the data are of high quality, I don't follow the reasoning of how similarity in loss- and gain-of-function nord phenotypes supports their model (or Nord having both positive and negative effects on BMP signaling depending on where and when Nord is expressed). Could the authors use available Gbb and Dpp antibodies to determine the expression patterns of these proteins in nord LOF and GOF?

Related to this, the authors do not control for the number of UAS transgenes in Figure 7. If you want to compare 3X UAS-Nord to 1X UAS-Nord, the 1X UAS-Nord sample should have 2X UAS-LacZ, GFP or another neutral protein. Being able to prove that they have not titrated Gal4 is critical to their model about low vs high Nord. These experiments should be performed again with all animals have the same number of UAS transgenes.

b) The authors do not provide a description of micro-array analysis. Additionally, there is no mention of the deposition of these data into public repository like NCBI GEO, to which the reviewers should be provided access. Related to this, there is no information on fold change of nord among the various sorted populations. They discuss top-ranked genes, but none of these data are presented. All of these need to be addressed.

c) We need more information about the Halfon hs-Gal4 technique that they used. Additionally, there are too many "clones" in 1H and we don't know where the "clones" are in Figure 1G. Could the authors use FLP/FRT flip-out approaches to generate a few discrete clones that mis-express Hh or Ptc?

d) The reader needs more information about the nord homologs that were examined in cell culture in Figure 2. For example, what is DrNdnf?

e) Figure 2C, the labels should be M and L, not M and C.

f) The font size is too small in Figure 2A. B, Figure 3D, Figure 3F, Figure 3M. Please enlarge.

*Reviewer #2:*

Wu et al. identified nord as a potential Hh-target gene in the *Drosophila* wing imaginal disc. Nord shares sequence similarity with a family of secreted proteins containing a domain of unknown function (DUF2369) and one or two fibronectin type II-like repeats which contains the neuron-derived neurotrophic factor (NDNF) which has been shown to promote neuronal migration and neurite outgrowth in vertebrates, such as the gonadotropin-releasing hormone producing neurons critical in the regulation of puberty and reproduction. The authors explored the function of nord in wing development. Hh is known to be secreted from cells in the posterior compartment of the wing disc and act on cells across the A/P boundary where an adjacent stripe of anterior cells express the Hh receptor ptc. Interestingly, nord is expressed in a subset of these 'stripe' cells consistent with it being a Hh-target. The dpp gene is a well known Hh-target expressed throughout the 'stripe' like ptc. dpp encodes a BMP2/4 ligand which is known, with the *Drosophila* BMP5/6/7 ligand, Gbb, to play multiple roles in wing growth and patterning. Wu et al. find while flies homozygous for LOF mutations in nord are viable and fertile, their wings are slightly smaller and they contain ectopic vein material at the posterior cross vein (PCV). A reduction in BMP signaling results in smaller wings but an increase in BMP signaling is associated with PCV spurs. Based on these phenotypes, Wu et al. investigate the potential relationship between nord function and BMP signaling. BMP signaling in the larva and its roles in growth and patterning and its later roles in the pupal wing disc in laying down vein material are temporally separable events. Furthermore, both the Blair and O'Connor labs have nicely delineated a mechanistic explanation for BMP involvement in the formation of the PCV. How nord is involved in each of these processes is not completely clear, especially given nord's expression pattern. Must the spatial and temporal requirements for BMPs in these processes be re-evaluated?

Is there a connection between how NDNF influences the migration neurons and its extracellular role in modulating molecules such as BMPs? Does it provide a link with extracellular matrix and other extracellular molecules that also appear to be modulators of BMP activity? Could Nord impact BMP output by influencing wing disc cellular projections?

Strengths:

– Identification of a Hh-target gene that appears to modify BMP signaling output.

– Ectopic vein material near the posterior cross vein (PCV) is attributed to loss of nord.

– A loss of function nord mutants exhibit a reduction in wing size.

– Nord-GFP is shown to bind to Dpp, and less so to Gbb, by co-immunoprecipitations in S2 cells.

Weaknesses:

– The manuscript would benefit from stating at the start that the researcher's findings suggest that nord may both facilitate and antagonize BMP signaling. It will be easier for the reader to follow.

– Better clarification of the different functions for BMP signaling in wing growth, patterning, and development in the larval versus pupal period would strengthen the manuscript.

– The authors can then follow up on these points as to how nord impacts these different functions

– For the most part nord is not expressed in either wing pouch cells or vein primordia. Furthermore, based on expression in cell culture, Nord is not found far from the source cell. How is Nord though to influence pMad, for example, in cells in the anterior wing pouch?

– It is import to reconcile the findings of others that the pMad gradient is not required for wing disc growth.

General points:

It would be helpful to the reader if the authors stated at the outset that they are suggesting that the impact of nord on BMP signaling can be both as a facilitator and an antagonist, and then go on to present data and the contexts in which they see different behaviors.

In the wing disc, pMad represent the output of BMP signaling, both the action of Dpp and Gbb. In most instances it is impossible to separate the effects of the two ligands, whether they act as a heterodimer or as homodimers, therefore, in nearly all places in the text "Dpp signaling" should be replaced with "BMP signaling", unless the authors know for certain they are examining signaling induced solely by Dpp.

Methods:

All *Drosophila* strains should be listed under "*Drosophila* strains".

Method used for overexpression of Hh or Ptc should be presented under some other subheading than *Drosophila* strains.

Third instar larval period is 48hrs at 22°C. If animals were heat shocked in early third and then dissected after two days they should have pupated. More accurate documentation of developmental time should be provided, or the authors should indicate that the larvae were anywhere between X and Y hrs or days old.

Epitomics 1880-1 was raised against phospho-Smad3, not pMad.

Constructs – the nucleotide sequences of the Nord and Ndnf – GFP and 3xMyc fusions should be shown.

Dpp-FLAG, Gbb-FLAG, Gbb-HA, and Mad-FLAG are not described in Serpe et al. 2005 – a description and characterization of activity should be provided

Section "RT-qPCR" needs to be renamed to indicate it contains the description of imaginal disc cell isolation. Were these the cells and RNA used for microarray analysis referred to in the text?

Imaginal discs and gene expression patterns change significantly during the 3rd larval instar. At what time were wing imaginal discs harvested for RNA isolation and immunohistochemistry?

No description of procedure for FACS or microarray was provided.

Results

Figure 1 – when first describing nord gene expression on page 7 Figure 1D & E, its absence from the wing pouch should be noted and that it is expressed anterior to the A/P boundary in the hinge progenitors.

Description of Nord-GFP in discs following over expression of Hh or Ptc is confusing (bottom of page 8) as the tissues are derived from larvae not flies.

hs>hh leads to expansion of Nord-GFP in hinge and wing periphery

hs>ptc breadth of Nord-GFP domain narrowed, only remaining is at A/P boundary.

Figure 1 H – do the author's have an explanation for the uneven expression of ptc and the corresponding response of Nord-GFP expression? Nord-GFP appears to be lacking in cells where Ptc is present in Figure 1G but Nord-GFP expression and Ptc are present in quite a number of cells in the hinge region of a wild type wing disc Figure 1F.

Figure legend: error? – "White brackets indicate the expression range of Ptc (E) or No-GFP (E-H)".

Figure S2 panel A – include a scale in bp.

panel E – why is the expected 732bp band seen when P3 + P4 are used to amplify MI06414-GFSTF.2?

S2 figure legend lacks reference to panels D-F.

Figure S3 – sequencing described?

Residue numbers should be included in figure.

Figure 2 – should note that *melanogaster* was used – provided species name for other genus' clone names should be listed in figure legend, ie which species the Ndnf gene was derived from;

panel B – no size markers.

lanes in figure B marked "M" and "C" but in figure legend noted as "medium (M)" and "cell lysate (L)".

Indicate that the resulting Ndnf proteins were tagged with Myc – in figure legend.

Explanation for difference in size of DmNord found in media versus cell lysate?

What are the predicted sizes for Nord/Ndnf proteins?

Ideas about the differential abundance of Ndnf in media from lysate from different species.

An important control is needed – WT disc with Anti-GFP.

Is it known that MI0641 produces a truncated peptide, or is it predicted?

Does higher prevalence of PCV spur in females suggest a second site enhancer? Dosage compensation?

What is the correlation between ectopic vein formation in the pupal wing and expression in the larval disc?

Authors should be careful when describing protein versus RNA expression patterns, as well as loss of gene function versus loss of gene expression. For example on page 12, neither deCelis 1997 or Yu et al. 1996 examined Dpp (protein) expression They report *in situ*s (endogenous RNA) or lacZ enhancer trap expression patterns.

In the sentence "We asked whether elimination of Nord alters level of pMad" – should be rephrased " We asked if the knock down (or reduction) in nord function …"

pMad is a downstream signal transducer, not a transcription transducer

Harmansa et al. 2015 proposed that dpp spreading is required for medial but not lateral wing growth. How would this finding apply to the propose function of nord with respect to wing growth?

Harmansa 2017 showed that eGFP-Dpp dispersal in basolateral is required for patterning & growth. Does Nord have subcellular localization?

An antibody against the Dpp prodomain is used in Figure 4A. BMP proproteins are cleaved separating the prodomain from the ligand domain proper, with the ligand sometimes secreted with its prodomain and other times, not. Do the authors feel this pattern reflects that of the secreted Dpp ligand?

Nord-GFP is in a subset of Dpp prodomain expression in larval wing disc (Figure 4A) and shows even less overlap in pupal wing disc (Figure 4B) – this needs to be more clearly addressed in the text.

Data not shown for Nord-GFP in later pupal wing when dpp-lacZ transitions to longitudinal veins.

"While dpp is also known to be expressed in the developing longitudinal veins (deCelis, 1997), we see no Nord expression in either the longitudinal vein or crossvein despite the fact that Nord loss of function leads to ectopic crossvein formation." The data referred to in the underlined text top page 13 is not shown. (?)

Figure 4E – Is quantification plot an average of multiple discs? how many discs were measured? Where was the measurement taken? Is it a profile from a line across the disc or a compilation of all signal in the disc? Would the authors expect pMad to be altered in the wing pouch?

Figure 4F – not at all clear what is meant by "maximum pMad intensity in the A and P compartment" in the figure legend. Is this the highest intensity measurement in the entire disc?

Is the proposal that the slight reduction in pMad in the anterior affects only growth and patterning of the position of the wing veins?

If Nord is required to facilitate "Dpp" movement, then why is signaling activation (as measured by pMad) in the majority of the anterior compartment beyond the peak unchanged. If in the absence of Nord, Dpp does not move, then one would expect all signaling in the anterior to be affected.

Instead of facilitating movement, could Nord prevent Dpp-Gbb turnover, thus in its absence Dpp-Gbb is less stable and a drop in signaling (pMad) is observed. Difficult to understand how Nord has an effect on ligands when its not present in wing pouch cells

Top half of pg 16 – authors suggest that media from cells co-expressing Nord and Dpp/Gbb elicit less signaling (pMad) than media produced by cells without Nord -if the same amount of media is added to cells in Figure 5C then it seems that M4 media is less effective at inducing pMad compared to the lower amount of Dpp/Gbb in M5 which elicits nearly the same level of pMad.

What does a normalized plot show?

Using a recombinant Dpp peptide in Figure 5D, the authors find that in the presence of Nord-GFP, equivalent amounts of rDpp induce less pMad but this is the opposite of what is observed *in vivo* in the disc where signaling is reduced in nord mutants. Perhaps the authors can expand on this observation. Is this result related to the fact that a recombinant peptide was used in rather than the entire Dpp proprotein or prodomain + ligand domain?

Figure 5E in WT – seems odd that one cannot see increased concentration of Mad-Flag in the nucleus.

Suggest that Nord-GFP could prevent release of ligand from cell surface and inhibit ligand from media to act on responding cell – what about ligand-receptor binding?

Westerns in Figure 5 are cropped and it is not possible to see the other cleaved forms of ligands. Does Nord show differential interaction with different ligand forms, as has been shown by Anderson 2017 to be the case with different forms of Gbb and receptors?

Figure 6

How were pMad profiles measured? Superimposing profiles in D-F would be helpful.

Authors state ptc levels have not changed – provide measurement of ptc levels.

Loss of L5 and PCV are typical of gbb LOF phenotypes and not seen in dpp LOF genotypes, therefore, it is hard to reconcile these phenotypes with if Nord is thought to preferentially affect dpp.

In Figure 6H – nub/+ label could easily be confused with a nub mutant heterozygote.

A nub-Gal4/+ 29°C control wing should be shown.

In some places nord is indicated as a secreted protein and in others membrane-associated – a little bit confusing.

Akiyama 2015 showed that the pMad gradient in the wing pouch, as well as Dpp were not required for wing growth. How do the authors think changes in the pMad profile may change growth? It is curious that changes in the pMad gradient seen in Figure 6F has no effect on longitudinal vein development in Figure 6K.

The Discussion contains a considerable amount of background information that is not particularly relevant to the findings presented in manuscript. Focusing the discussion a bit more would be helpful to the reader.

To the reader it is not at all clear how expression in the posterior compartment during larval period influences PCV formation in the pupal wing when ligands are thought to be transported from the longitudinal veins.

Does uniform expression in posterior using enGal4 interfer with the normal mechanism?

Suppl Figure 12 – orientation of high mag of trichomes is incorrect.

Overall, please check labelling of figure for units, number of samples (n=?), and accuracy of figure legend description. In some places in the manuscript this information is robust but in other places it is lacking.

*Reviewer #3:*

The study focuses on a possible role for the secreted Nord protein in BMP signaling in *Drosophila*. The authors demonstrate Nord binding to Dpp and Gbb *in vitro*, and suggest that this explains the *in vivo* phenotypes observed. The *in vitro* work is quite suggestive, but some caution may be in order as other other extracellular molecules can affect multiple pathways, and some of these other pathways can have have similar effects on growth and vein formation. While this is a worthwhile publication with strong *in vitro* data, I have a few points below that I think would strengthen the *in vivo* data.

1) Nord as a Hh target- The expression of Nord in the imaginal and pupal wing tissues largely mirrors that of Hh targets, being heightened anterior to the anterior-posterior compartments boundary. Its expression is however excluded from the distal wing blade, something that the authors do not further investigate. The authors can widen or narrow Nord expression by widening or reducing Hh signaling, but this will also widen BMP expression and signaling, so it would have been helpful to have one experiment showing a cell autonomous effect of Hh signal reception on Nord expression.

2) Nord mutants- The authors generate and examine Nord mutants. The effects in the wing are fairly mild, consisting of very slight undergrowth mirrored by slightly reduced BMP signaling (pMad) in the disc, and a few ectopic veins, suggestive of gains in BMP signaling. The only other growth-inducing pathways the authors examine are Hh, looking at a single target, Ptc. They do not however quantify Ptc staining in the same way they did to detect the weak effects on pMad, nor do they look at other Hh targets with different sensitivities.

The ectopic veins in mutants are consistent with defects seen with mild increases in BMP signaling, but there are many other ways of inducing similar ectopic venation, especially through EGF gains or reductions in Notch or its ligands. The authors concentrate almost exclusively on one defect, a vein extension distal to the PCV. This is a common defect seen in many mutant fly stocks as it is something of a “hot-spot” for ectopic venation. The authors show that pMad is high in this ectopic vein, but do this staining at 26hr AP, a fairly late stage of PCV development. By this time LVs and cells near the ACV have all been expressing dpp for some time and have BMP signaling, while dpp expression in the PCV is initiating. I would wager this ectopic vein is expressing dpp, and will also have rhomboid and thus EGFR activity and high Δ and Serrate. The authors could greatly strengthen their case by examining pMad at earlier stages of PCV development (20 hr AP) before EGFR and Notch signaling are active in this region.

The authors also do not test whether the posterior defects they see are caused by loss of anterior Nord, or whether they might be caused by loss of low-level Nord expression in the posterior. Since they have a Nord-RNAi line, what happens with anterior knockdown? Posterior?

3) Nord overexpression and biphasic effects on BMP signaling – Increased Nord expression causes undergrowth and reduced pMad but ectopic venation. In that sense, Nord has biphasic effects, with similar mixed effects (both BMP losses and gains) caused by either too much or too little Nord. The authors extend this by showing that further increases in Nord overexpression change the ectopic PCV extension into PCV loss, the latter often caused by loss of BMP signaling. However, this loss may not always be due to a direct effect on the PCV. While the nub-gal4 example looks ok, the en-gal4 example has lost L5 adjacent to the PCV, likely due to loss of the distal L5 primordium at the disc stages (Ray and Wharton, Bangi and Wharton). That will in turn block or reduce dpp expression in L5 and thus Dpp’s availability for signaling in the early PCV.

4) *In vitro* studies- These provide the strongest proof that Nord can affect BMP signaling, although in this case the observed effects are all inhibitory. Secreted Nord can co-IP with secreted Gbb or Dpp, and co-expression with Nord reduces Gbb or Dpp secretion and thus its ability to signal. Nord can also reduce the BMP signaling induced by addition of exogenous Dpp. The only shortcoming here is that the authors have not apparently tested binding to any of the extracellular modulators of BMP signaling.

I need to see whether there is ectopic pMad in the nord mutant PCV at a stage before dpp, rhomboid and Δ/Serrate expression in the region. Otherwise I am not convinced that the ectopic vein is initiated by ectopic BMP signaling.

Quantify Ptc or some other Hh marker (Dpp would be especially helpful) in Nord mutants.

How much of the PCV loss in en>Nord might be caused by loss of the adjacent L5?

Cell-autonomous effects of Hh signaling on Nord expression would be helpful, as would the effects of anterior vs. posterior Nord RNAi.

[Editors’ note: further revisions were suggested prior to acceptance, as described below.]

Thank you for submitting your article “The NDNF-like factor Nord is a Hedgehog-induced extracellular BMP modulator that regulates *Drosophila* wing patterning and growth” for consideration by *eLife*. Your article has been reviewed by 2 peer reviewers, including Erika A Bach as the Reviewing Editor and Reviewer #1, and the evaluation has been overseen by Utpal Banerjee as the Senior Editor.

Essential revisions:

1) Soften their conclusions about their *in vitro* work (Figure 8) in the results and discussion. Add a statement to the effect that without additional replicates in Figure 8 other conclusions are possible.

2) Provide additional data on the Nord in the posterior compartment using ptc-gal4, UAS-nord-RNAi.

3) Add information about the cross vein phenotype of MS^-1^096 to the manuscript.

4) Make changes to the Figure fonts.

5) Correct the typos and awkward sentences.

*Reviewer #1:*

The authors have made strong efforts to respond to the essential revisions and the reviewers’ critiques. Overall, the manuscript is much improved. However, I have concerns about the conclusions regarding Figure 8, the *in vitro* work, which the authors use conclude that Nord *in vitro* also has biphasic function on BMPs *in vitro*. The authors present results from a single experiment and then normalize these single data points to draw conclusions about Nord function *in vitro*. Specifically, the authors’ new analyses in Figure 8 panels D,E,G show relative pMad levels normalized to Mad (8D,8G) and to Mad and BMP (8E). My concerns are as follows. First, the authors need statistical analyses. Second, BMP released from cells appear to be equally active in the absence (M4) or presence (ME) of Nord, and these data only become supportive of their argument when “normalized” to the amount of ligand. Third, there are additional bands consistent with other Dpp/Gbb isoform in the source blot for Figure 8A. These may be bioactive molecules but are not accounted for in the manuscript. The pMad bands in M4 and M5 also appear to be saturated, again making it difficult to determine differences between the lanes. In panel C, the WB: Flag row is "smiling" with some lanes compacted and others not, again making it difficult to normalize. Without more replicates, the authors statement that “soluble ligand appears to have a higher signaling activity perhaps as a result of an association with Nord” (p.21) is not sufficiently supported by the data. This same issue holds for panel F, with only one replicate. In its current form, the data are not conclusive and other interpretations are also possible. The authors should soften their conclusions about their *in vitro* work in the results and discussion. Additionally, they need add a statement to the effect that without additional replicates in Figure 8 other conclusions are possible.

They are other issues that need to be addressed:

1. The figure in the response to reviewers was too small. In the future, please upload a larger image.

2. P. 7. The authors write “In wing discs, cells near the A/P compartment boundary (B: ptc+) receive the highest level of Hh stimulation while A cells (A: hh-), located further from the border, receive lower levels of stimulation. In contrast, P cells (P: hH^+^) do not respond to Hh due to lack of the receptor Ptc and the transcription factor Ci (Tabata et al., 1992; Eaton and Kornberg, 1990).” It is very important that the authors add “Figure 1” when discussing B, A and P cells.

3. P. 11 and Figure 3C – please clarity what tissue is being used for the flip-out clones.

4. Top of P. 15. Please rephrase “which is consistent with the obviously reduced BMP, but not Hh, signaling activity during wing imaginal disc development (Figure 5B, C)” because it is not clear what is being discussed. I suggest “which is consistent with normal Hh signaling activity during the development of nord mutant wing imaginal discs (Figure 5B, C).”

5. P. 18 The authors cite Robert 1998 but there is no such citation in the references cited section. Please add this reference or remove it from the manuscript.

6. P. 21. As noted, please soften your conclusions about the *in vitro* studies and add a statement to the effect that without additional replicates in Figure 8 other conclusions are possible.

7. P. 25, line 9 “it’s” should be “its”.

8. P. 25, line 13 “Akiyama et al. 2021 accompanying manuscript” is not in the references cited.

9. Add to the Materials and methods the procedure you used for normalization of the data in Figure 8E,D,G.

10. Figure 1B – increase the font size of the colored/highlighted genes. Currently we can’t read them.

11. Figure 2A – increase the font size of the RMCE vectors. Currently we can’t read them.

12. Figure 2D, E, F, G, H – increase the width of the lines outlining regions of the wing disc and clones. Currently we can’t see them.

13. Figure 2C-H, it would be helpful to add C’ and C” etc to the unlabeled panels.

14. Figure 4E – The graphs are too small – I cannot read anything. Please increase the size of the graphs and the font of the labels.

15. Figure 5D – increase the width of the lines of the boxes and the enlargements so that they can be seen on a print out.

16. Figure 7B – increase the width of the lines of the boxes and the enlargements so that they can be seen on a print out.

17. Figure 2, figure supplement 3. Increase the size of the brackets in the Nord-GFP monochrome panels. Currently, it is very hard to see them.

18. Figure 5—figure supplement 2B – label the blue and the red lines.

*Reviewer #2:*

The authors have added some valuable data that have improved the manuscript, enough for acceptance. I do however have some points I would like the authors to seriously consider. (1) and (2) concern data that are problematic, but could be removed from the manuscript without damaging its conclusions.

1) Hh signaling and Nord expression- The autonomous gains in Nord expression caused by Smo overexpression are convincing. The loss caused by Ptc overexpression, shown for a single flpout clone Figure 3D, is not. To be a valid result, the dotted blue line drawn by the authors must accurately follow the A/P. However, outside the Ptc clone at the bottom of the figure, the dotted line does not at all correspond to the line shown by Nord expression, even though the Nord outside the clone should define the A/P. Nor does the line correspond to the A/P defined by Ci staining. Instead, the true A/P boundary is further left, and thus it is not at all clear that the Ptc clone is in the anterior compartment. If the authors want to keep Ptc flpout data in the manuscript, they need a more convincing example.

2) MS1096-gal4- The authors use MS1096 for some of the nord-RNAi data. This is a Gal4 enhancer trap insertion into the X-linked Bx (dLMO) locus, and in hemizygous males has a mild Bx wing mutant phenotype with variable penetrance that includes disruption of the crossveins (see Figure 6 in Milan, Diaz-Benjumea, and Cohen 1998). The crossvein phenotype tends to disappear in stocks, but often reappears when outcrossing to other lines. It is therefore not an appropriate driver to use when assessing crossvein phenotypes in males. If the authors want to use it, they have to explain how they dealt with the crossvein phenotypes that can be caused by MS1096 on its own.

And while MS1096 is expressed in the wing disc, it is not clear that MS1096 is “wing-specific” as the authors state. Bx (dLMO) is expressed in many tissues, including the CNS. If MS1096 follows the expression of Bx, the authors cannot say that this experiment “further validates that the observed phenotypes in the nord mutants are not indirect effects associated with loss-of-nord in tissues other than the wing discs.”

3) Is Nord in the posterior compartment? The authors make several absolute statements about the absence of Nord expression from the posterior of the developing wing, but do not provide any staining of controls that would allow one to distinguish signal from background. While Nord is certainly much higher anterior to the A/P boundary, the data does not rule out low-level Nord expression in the posterior. The authors have addressed this in their revision, not by changing any of their absolute statements about expression, but by added a new experiment showing that “we did not notice any ectopic PCV in the adult flies when UAS-nord-RNAi was selectively expressed in the P compartment of the larval and pupal wing via the hh-Gal4 driver.” But this negative result could simply reflect stronger RNAi expression with MS1096-gal4 or A9-gal4. It would be preferable if they had tested the effects of nord-RNAi when it was limited to the anterior compartment, something that their model predicts should affect the posterior crossvein. This could easily be done with ptc-gal4 or ci-gal4.

The authors also model the action of Nord as if it only diffuses over a short distance, and thus is only affecting BMP movement from neighboring cells, such as those”in L4 in pupal wings, and is never present in the PCV. However, the extracellular staining of Nord-GFP in Figure 3 looks substantially above background throughout the wing disc.

4) Biphasic or multiphasic effects- The authors have added new data that clarify some of the effects of Nord in the PCV, but complicate their story about biphasic effects. In growth Nord appears truly biphasic, promoting growth only at normal levels, and inhibiting growth and BMP signaling with either decreased or increased Nord expression. But in PCV formation things are more complex. Both loss of endogenous Nord and mild gains induce ectopic PCV phenotypes. Since normal levels do not alter the PCV, that also fits with a biphasic effect, albeit the opposite of the growth effect (ectopic signaling with both losses and mild gains). Overexpression also has a biphasic effect, as strong Nord gains do the opposite of weak gains and cause loss of the PCV and signaling. However, if one puts all of this together, the PCV effect is more multiphasic than biphasic, because of the normal signaling observed at wild type. Nord loss increases signaling, wild type is normal, mild Nord overexpression increases, strong Nord decreases.

The only detailed explanation of these effects is in the legend of Figure 9, which is not very complete and leaves some puzzles. If loss of Nord increases long-range BMP diffusion into the PCV, why isn’t the same thing happening in a third instar wing pouch? And there are ideas hidden in the figure itself that are not explained in the legend or the Discussion, especially the switch from BMP presentation to BMP sequestration with a change from low local to high local Nord concentrations. A fuller discussion in the text would be helpful.

5) Biphasic *in vitro*- The *in vitro* work remains strong, especially for Nord’s inhibitory effects on BMP signaling. But the only evidence for a positive and thus biphasic effect of adding Nord relies on a single lane in a single gel: the lowest Nord concentration in Figure 8F, then quantified in 8G. It would be helpful if the authors could state how many times they have repeated this result.

6) Given the long discussion of NDNF, the authors should mention the mechanisms that have been proposed for NDNF activity: by binding integrins, or by modulation of FGF signaling by binding HSPGs.

---

## [Author Response]

[Editors’ note: the authors resubmitted a revised version of the paper for consideration. What follows is the authors’ response to the first round of review.]

Specifically, the reviewers felt that many experiments would be needed to better characterize the nord mutant phenotype, particularly its role in BMP signaling. They wanted the authors to:1. Measure Dpp levels in nord mutants. This could be accomplished using the anti-pro-Dpp, or enhancer traps that are more accurate than BS3.0 (which not only lack the shortvein enhancers but is overly sensitive to anterior en).

The suggested experiment was performed, in which no noticeable change of the Dpp (anti-Dpp prodomain) expression levels were detected in the *nord* mutant wing discs. These data are presented in Figure 5—figure supplement 2 in the revised manuscript.

2. Measure Ptc levels in nord mutants.

The suggested experiment was performed and included in Figure 5C in the revised manuscript.

To further differentiate the effect of *loss-of-nord* on Hh vs. BMP signaling pathways, we measured the dL3-L4 /dL2-L5 ratio in the adult wing (Figure 4C), and these results were described on pages14-15 (below).

“Of note, beside BMPs, Hh signaling also patterns the developing *Drosophila* wing. Hh shortrange activity is responsible for patterning the central L3-L4 region and determining the distance between L3 and L4 longitudinal veins (Mullor et al., 1997; Strigini and Cohen, 1997). When quantifying the distance between distal ends of longitudinal veins L3 and L4 (dL3-L4) and that of L2 and L5 (dL2-L5), we found no specific reduction in the L3-L4 region indicated by comparable or higher dL3-L4 /dL2-L5 ratio in the female and male nord mutant wings (Figure 4B, C), which is consistent with the obviously reduced BMP, but not Hh, signaling activity during wing imaginal disc development (Figure 5B, C). Together, these findings suggested a positive role of endogenous Nord in augmenting BMP signaling activity to promote wing growth.”

3. Show a cell autonomous effect of Hh signal reception on Nord expression.

The suggested experiment was performed and included in Figure 2C-H in the revised manuscript. Briefly, we found that Nord expression is cell-autonomously induced/inhibited by Hh signaling within the anterior compartment in the wing discs’ hinge and outer wing pouch region.

4. Examine pMad at earlier stages of PCV development (20 hr AP) before EGFR and Notch signaling are active in this region.

The suggested experiment was performed and included in Figure 5D-E in the revised manuscript. Expansion of pMad range around the primordial PCV was noticed in the *nord* mutants starting at 19-20 hr after pupation.

5. Test whether the posterior defects they see are caused by loss of anterior Nord, or whether they might be caused by loss of low-level Nord expression in the posterior. In other words, what is the result when Nord is depleted from anterior or from posterior cells.

Our findings show that *nord*, like *ptc* and *dpp*, is a target gene of the Hh signaling pathway and not expressed in the posterior compartment of the wing discs (Figures 1, 2, 5A; Figure 2—figure supplement 3; Figure 5—figure supplement 1). Consistently, when we knocked down *nord* in the posterior compartment using *hh-Gal4*, we did not observe the ectopic PCV phenotype, which is frequently caused by knocking down *nord* throughout the wing discs via *MS1096-Gal4* or *A9- Gal4*. These data are presented in Figure 5—figure supplements 3 and 4 in the revised manuscript.

6. Additionally, the authors should determine how far secreted Nord can spread in the disc.

We expressed *UAS-Nord* in small clones in the wing discs and found that Nord was mainly detected on the expressing cells' surface or nearby extracellular matrix (Figure 3D). We also detected very low levels of secreted Nord-GFP in the wing discs carrying the *nord-GFP* protein trap allele (Figure 3E). Additionally, we observed that ectopic Nord driven by *hh-Gal4* or *en-Gal4* could induce ectopic crossvein formation in the anterior compartment close to the A/P boundary, indicating secreted Nord can spread and act several cell diameters away. These data are presented in Figure 7B and Figure 7—figure supplement 2 in the revised manuscript. Based on these observations, we think a portion of Nord can spread at least several cell diameters away from the source cells.

7. Test how much of the PCV loss in en>Nord is caused by loss of the adjacent L5.

High levels of ectopic Nord driven by *en-Gal4* (or *hh-Gal4*) in the larval stage can sequester BMP ligands and inhibit BMP signaling, thus lead to loss of L5 (original Figure 7). To avoid affecting L5 development, in the revised version, we restricted ectopic Nord expression to the pupal wing and then studied its effect on PCV patterning. The temporal control of *en-Gal4* (*or hh-Gal4*) driven expression of *UAS-Nord* was achieved by including the *tub-Gal80ts* allele. In this new setting, in the presence of normal L5, ectopic Nord caused PCV phenotypes in a dosage-dependent manner. These data are presented in a new Figure 7 in the revised manuscript.

8. Add missing controls such as a nub-Gal4/+ 29°ting and genome-wide expression proC control wing and a control disc stained with anti-GFP.

The suggested experiment was performed and included in Figure 6 in the revised manuscript.

9. Improve quantification for pMad analysis, particularly in Figure 4E, 4F and 6.

The suggested experiment was performed and included in Figure 5 (original Figure 4) and Figure 6 in the revised manuscript.

10. The reviewers also suggested extensive editorial changes that would make the manuscript easier for the reader to follow.

Based on the reviewers' suggestions, we reorganized the Results section to better communicate the idea that Nord is a dosage-dependent biphasic modulator of BMP signaling. We also shortened the discussion by mainly focusing on the mechanistic aspects of Nord/Ndnf-mediated regulation of BMP signaling.

Reviewer #1:The authors used cell sorting and genome-wide expression profiling to identify new Hedgehog (Hh) target genes in the developing *Drosophila* wing disc, which has been a powerful system to study Hh signaling. They identify nord, which encodes a secreted factor that associates with cell membranes and the extracellular matrix. Null mutants of nord are adult viable. Paradoxically, these animals have slightly smaller wings and ectopic posterior cross vein (PCV) material. Both phenotypes can be caused by alterations in BMP signaling. They demonstrate that recombinant Nord binds to Dpp and Gbb and restricts their release from cells and diffusion in the media. They use the Gal4/UAS mis-expression to test the model that Nord has both positive and negative effects on BMP signaling depending on where and when Nord is expressed.The strengths of this paper are the appropriate use of techniques such as cell sorting, transcriptomics, *Drosophila* loss of function genetics, cell culture. These approaches support their model that Nord regulates availability of Dpp, Gbb and/or Dpp-Gbb dimers. However, the use of mis-expression of wild-type Nord in the pouch or posterior compartment lead similar phenotypes as global nord loss-of-function. This is a weakness because it is difficult for the reader to reconcile logically these paradoxical results. This work may impact the mammalian fields because Nord is conserved through evolution, but Nord homologues have not been implicated in BMP signaling.

We want to thank the reviewer for the time and effort invested in reviewing our manuscript, as well as the instructive comments. In the revised manuscript, we have provided additional experimental evidence to support our model that Nord is a biphasic regulator of BMP signaling.

In addition, extensive editorial changes were made to help the reader to reconcile logically the paradoxical results associated with the biphasic nature of Nord in regulating BMP signaling.

a) While I appreciate that they authors have done a lot of work and the data are of high quality, I don't follow the reasoning of how similarity in loss- and gain-of-function nord phenotypes supports their model (or Nord having both positive and negative effects on BMP signaling depending on where and when Nord is expressed). Could the authors use available Gbb and Dpp antibodies to determine the expression patterns of these proteins in nord LOF and GOF?

We have provided additional experimental results to support the model that Nord is a dosage depend biphasic BMP modulator, where low levels of Nord promote, and high levels inhibit BMP signaling. The biphasic feature of Nord was observed both in the larval wing discs during wing growth and in the pupal wing in the process of PCV patterning. Additionally, we show the biphasic activity of Nord on BMP signaling in cultured S2 cells. These data are presented in Figures 6-8 in the revised manuscript.

The suggested experiment was performed, in which no noticeable change of the Dpp (anti-Dpp prodomain) expression levels were detected in the *nord* mutant wing discs. These data are presented in Figure 5—figure supplement 2 in the revised manuscript.

Related to this, the authors do not control for the number of UAS transgenes in Figure 7. If you want to compare 3X UAS-Nord to 1X UAS-Nord, the 1X UAS-Nord sample should have 2X UAS-LacZ, GFP or another neutral protein. Being able to prove that they have not titrated Gal4 is critical to their model about low vs high Nord. These experiments should be performed again with all animals have the same number of UAS transgenes.

The suggested experiment was performed and is included in Figure 7 in the revised manuscript.

b) The authors do not provide a description of micro-array analysis. Additionally, there is no mention of the deposition of these data into public repository like NCBI GEO, to which the reviewers should be provided access. Related to this, there is no information on fold change of nord among the various sorted populations. They discuss top-ranked genes, but none of these data are presented. All of these need to be addressed.

In the method section of the revised manuscript, we now have provided the GEO information related to the array. See Page 30.

We also added new figures (Figure 1; Supplementary file 1) to provide detailed information about how *nord* was selected from the array analysis. See Page 7.

c) We need more information about the Halfon hs-Gal4 technique that they used. Additionally, there are too many "clones" in 1H and we don't know where the "clones" are in Figure 1G. Could the authors use FLP/FRT flip-out approaches to generate a few discrete clones that mis-express Hh or Ptc?

The suggested experiment was performed and included in Figure 2C-H in the revised manuscript. Briefly, we found that Nord expression is cell-autonomously induced/inhibited by Hh signaling within the anterior compartment in the wing discs’ hinge and outer wing pouch region. See Pages 9-10.

d) The reader needs more information about the nord homologs that were examined in cell culture in Figure 2. For example, what is DrNdnf?

As suggested, the definition of Dm, Dr, Hs, Ce, and Xt were added into the legend of Figure 3 (original Figure 2). See Pages 39-40.

e) Figure 2C, the labels should be M and L, not M and C

Suggested changes were made in Figure 3 (original Figure 2) of the revised manuscript.

f) The font size is too small in Figure 2A. B, Figure 3D, Figure 3F, Figure 3M. Please enlarge.

Suggested font size change was made in Figures 3 and 4 (original Figures 2 and 3) in the revised version.

Reviewer #2:Wu et al. identified nord as a potential Hh-target gene in the *Drosophila* wing imaginal disc. Nord shares sequence similarity with a family of secreted proteins containing a domain of unknown function (DUF2369) and one or two fibronectin type II-like repeats which contains the neuron-derived neurotrophic factor (NDNF) which has been shown to promote neuronal migration and neurite outgrowth in vertebrates, such as the gonadotropin-releasing hormone producing neurons critical in the regulation of puberty and reproduction. The authors explored the function of nord in wing development. Hh is known to be secreted from cells in the posterior compartment of the wing disc and act on cells across the A/P boundary where an adjacent stripe of anterior cells express the Hh receptor ptc. Interestingly, nord is expressed in a subset of these 'stripe' cells consistent with it being a Hh-target. The dpp gene is a well known Hh-target expressed throughout the 'stripe' like ptc. dpp encodes a BMP2/4 ligand which is known, with the *Drosophila* BMP5/6/7 ligand, Gbb, to play multiple roles in wing growth and patterning. Wu et al. find while flies homozygous for LOF mutations in nord are viable and fertile, their wings are slightly smaller and they contain ectopic vein material at the posterior cross vein (PCV). A reduction in BMP signaling results in smaller wings but an increase in BMP signaling is associated with PCV spurs. Based on these phenotypes, Wu et al. investigate the potential relationship between nord function and BMP signaling. BMP signaling in the larva and its roles in growth and patterning and its later roles in the pupal wing disc in laying down vein material are temporally separable events. Furthermore, both the Blair and O'Connor labs have nicely delineated a mechanistic explanation for BMP involvement in the formation of the PCV. How nord is involved in each of these processes is not completely clear, especially given nord's expression pattern. Must the spatial and temporal requirements for BMPs in these processes be re-evaluated?Is there a connection between how NDNF influences the migration neurons and its extracellular role in modulating molecules such as BMPs? Does it provide a link with extracellular matrix and other extracellular molecules that also appear to be modulators of BMP activity? Could Nord impact BMP output by influencing wing disc cellular projections?Strengths:– Identification of a Hh-target gene that appears to modify BMP signaling output.– Ectopic vein material near the posterior cross vein (PCV) is attributed to loss of nord.– A loss of function nord mutants exhibit a reduction in wing size.– Nord-GFP is shown to bind to Dpp, and less so to Gbb, by co-immunoprecipitations in S2 cells.Weaknesses:– The manuscript would benefit from stating at the start that the researcher's findings suggest that nord may both facilitate and antagonize BMP signaling. It will be easier for the reader to follow.– Better clarification of the different functions for BMP signaling in wing growth, patterning, and development in the larval versus pupal period would strengthen the manuscript.– The authors can then follow up on these points as to how nord impacts these different functions– For the most part nord is not expressed in either wing pouch cells or vein primordia. Furthermore, based on expression in cell culture, Nord is not found far from the source cell. How is Nord though to influence pMad, for example, in cells in the anterior wing pouch?– It is import to reconcile the findings of others that the pMad gradient is not required for wing disc growth.

We want to thank the reviewer for the very detailed and constructive comments towards improving the quality and presentation of our manuscript. In the revised manuscript, we have provided additional experimental evidence, with better spatial (Figure 5, compared to original Figure 4) and temporal (Figure 7 compared to original Figure 7) resolution, to support our model that Nord can both facilitate and antagonize BMP signaling in a dosage-dependent manner. In addition, as suggested, the abstract and majority of the result section were re-written, and several figures were re-formatted to help the readers follow. The specific changes were explained below in the point-by-point response.

As for the three major questions raised by the reviewer, we commented on the second question “Does Nord provide a link with extracellular matrix and other extracellular molecules that also appear to be modulators of BMP activity?”, which is included in the Discussion section under “Nord is a novel, multi-functional BMP binding protein.” However, at this point, we would not be able to comment on the other two, which will be explored in future studies.

General points:It would be helpful to the reader if the authors stated at the outset that they are suggesting that the impact of nord on BMP signaling can be both as a facilitator and an antagonist, and then go on to present data and the contexts in which they see different behaviors.

In the revised manuscript, the abstract and majority of the result section were re-written by following reviewer 2’s suggestion. See Pages 2, 5, 14-22.

In the wing disc, pMad represent the output of BMP signaling, both the action of Dpp and Gbb. In most instances it is impossible to separate the effects of the two ligands, whether they act as a heterodimer or as homodimers, therefore, in nearly all places in the text "Dpp signaling" should be replaced with "BMP signaling", unless the authors know for certain they are examining signaling induced solely by Dpp.

Changes were made throughout the text as suggested.

Methods:All *Drosophila* strains should be listed under "*Drosophila* strains"Method used for overexpression of Hh or Ptc should be presented under some other subheading than *Drosophila* strains.

In the revised manuscript, we have listed all the used *Drosophila* strains in the Key Resources Table. The original “*Drosophila* strains” section was renamed as “*Drosophila* maintenance.”

Third instar larval period is 48hrs at 22°C. If animals were heat shocked in early third and then dissected after two days they should have pupated. More accurate documentation of developmental time should be provided, or the authors should indicate that the larvae were anywhere between X and Y hrs or days old.

Changes were made as suggested under the method section “*Drosophila* maintenance.”

Epitomics 1880-1 was raised against phospho-Smad3, not pMad.

The description of antibody used to recognize pMad was corrected under Method section under “Antibodies.”

Constructs – the nucleotide sequences of the Nord and Ndnf – GFP and 3xMyc fusions should be shown.

Information about Nord/Ndnf expression constructs is provided in the Key Resources Table and Supplementary file 3.

Dpp-FLAG, Gbb-FLAG, Gbb-HA, and Mad-FLAG are not described in Serpe et al. 2005 – a description and characterization of activity should be provided

Information about the constructs used to express tagged Dpp, Gbb and Mad is provided in the Key Resources Table and Supplementary file 3. We also corrected the related citation in the Method section under “Constructs.”

Section "RT-qPCR" needs to be renamed to indicate it contains the description of imaginal disc cell isolation. Were these the cells and RNA used for microarray analysis referred to in the text? Imaginal discs and gene expression patterns change significantly during the 3rd larval instar. At what time were wing imaginal discs harvested for RNA isolation and immunohistochemistry?

As suggested, we have separated the “Dissociation and sorting of imaginal disc cells” section from the “Quantitative reverse transcription PCR” section. The cells were dissociated from wondering third instar larvae, which is added into the revised method section. The procedure for FACS are included under “Dissociation and sorting of imaginal disc cells” and microarray were carried out under standard Affymetrix labeling and hybridization protocol, which was provided when depositing the GEO raw data.

ResultsFigure 1 – when first describing nord gene expression on page 7 Figure 1D & E, its absence from the wing pouch should be noted and that it is expressed anterior to the A/P boundary in the hinge progenitors.

The suggested change was made in the revised manuscript. See Page 6 (first paragraph).

No description of procedure for FACS or microarray was provided.Description of Nord-GFP in discs following over expression of Hh or Ptc is confusing (bottom of page 8) as the tissues are derived from larvae not flies.hs>hh leads to expansion of Nord-GFP in hinge and wing peripheryhs>ptc breadth of Nord-GFP domain narrowed, only remaining is at A/P boundary.

The suggested change was made in the revised manuscript. See Page 9 (second paragraph).

Figure 1 H – do the author's have an explanation for the uneven expression of ptc and the corresponding response of Nord-GFP expression? Nord-GFP appears to be lacking in cells where Ptc is present in Figure 1G but Nord-GFP expression and Ptc are present in quite a number of cells in the hinge region of a wild type wing disc Figure 1F.

The uneven expression (staining) of Ptc is due to the fact that anti-Ptc detected both endogenous Ptc and exogenous Ptc from UAS-Ptc driven by the Hs-Gal4. In the revised manuscript, we generated a Nord-RFP protein trap allele and examined Nord expression in Flipout clones with increased/reduced Hh signaling activity. In this experiment, the clones were labeled with the expression of mCD8-GFP, which allowed to avoid the use of Ptc antibody. We showed that Nord expression was cell-autonomously induced/inhibited by Hh signaling within the anterior compartment in the wing discs’ hinge and outer wing pouch region. These new data were included in Figure 2C-H in the revised manuscript.

Figure legend: error? – "White brackets indicate the expression range of Ptc (E) or No-GFP (E-H)".

The figure legend error was corrected in the revised manuscript.

Figure S2 panel A – include a scale in bp.

The scale in bp was provided in the revised manuscript in Figure 2—figure supplement 2.

Fig S2 panel E – why is the expected 732bp band seen when P3 + P4 are used to amplify MI06414-GFSTF.2?

It is likely due to alternative splicing by skipping the inserted cassette. Alternatively, it could due to a small portion of heterozygous flies was used during RNA extraction.

S2 figure legend lacks reference to panels D-F.

The figure legend error was corrected in the revised manuscript.

Figure S3 – sequencing described?Residue numbers should be included in figure.

The scale in bp was provided in the revised manuscript in Figure 4—figure supplement 1.

Figure 2 – should note that melanogaster was used – provided species name for other genus' clone names should be listed in figure legend, ie which species the Ndnf gene was derived from;

The species name was added into the legend of Figure 3 (original Figure 2) in the revised manuscript.

panel B – no size markers.

The size marker was added into Figure 3B (original Figure 2B).

Lanes in figure B marked "M" and "C" but in figure legend noted as "medium (M)" and "cell lysate (L)".

The figure label error was corrected in the revised manuscript in Figure 3B (original Figure 2B).

Indicate that the resulting Ndnf proteins were tagged with Myc – in figure legend.

The suggested information was added into the revised legend of Figure 3 (original Figure 2)

Explanation for difference in size of DmNord found in media versus cell lysate?

Secreted Ndnf/nord may undergo additional posttranslational modification, which lead to the slower migration of Ndnf/Nord in the Media compared to that in the cell lysate. Alternatively, the salt concentration in the media vs. lysate may also contribute to the different migration speed.

What are the predicted sizes for Nord/Ndnf proteins?

The predicted size for Nord (587 aa) /NDNF (568 aa) proteins is around 65-70 kDa, however we observed that these proteins migrate a slower than the predicted size, especially the secreted potion, which is likely due to post translational modification.

Ideas about the differential abundance of Ndnf in media from lysate from different species.

We are not sure about why there is differential abundance of Ndnf in the media from lysate from different species.

An important control is needed – WT disc with Anti-GFP.

A WT disc with Anti-GFP surface staining was added as the control in to Figure 3E (original Figure 2D).

Is it known that MI0641 produces a truncated peptide, or is it predicted?

We have modified the text to clarify that the truncate peptide is predicted.

Does higher prevalence of PCV spur in females suggest a second site enhancer? Dosage compensation?

Higher prevalence of PCV in females was also observed in *man1* mutants (Wagner et al., 2010 Dev Bio). It was thought to related to the differences in wing size. We are not entirely clear about the mechanisms underlying the PCV+ associated female preference.

What is the correlation between ectopic vein formation in the pupal wing and expression in the larval disc?

We propose the ectopic vein formation in the pupal wing is related to Nord expression in the early pupal wing discs (Figure 5A lower row; Figure 5—figure supplement 1), not that in the larval wing disc.

Authors should be careful when describing protein versus RNA expression patterns, as well as loss of gene function versus loss of gene expression. For example on page 12, neither deCelis 1997 or Yu et al. 1996 examined Dpp (protein) expression They report *in situ*s (endogenous RNA) or lacZ enhancer trap expression patterns.

We changed “Dpp” to “*dpp*” in the revised manuscript. See Page 13 (bottom line).

In the sentence "We asked whether elimination of Nord alters level of pMad" – should be rephrased " We asked if the knock down (or reduction) in nord function …"pMad is a downstream signal transducer, not a transcription transducer.

The suggested clarification was made in the revised manuscript. See Page 14 (second paragraph).

Harmansa et al. 2015 proposed that dpp spreading is required for medial but not lateral wing growth. How would this finding apply to the proposed function of nord with respect to wing growth?

We measured L2-L5 distance (dL2-L5) in the revised manuscript and found that dL2-L5 is significantly reduced in the *nord* mutant wings (Figure 4C). The reduced dL2-L5 indicates Nord-Dpp interaction is likely required for the medial wing growth. See Figure 4C.

Harmansa 2017 showed that eGFP-Dpp dispersal in basolateral is required for patterning & growth. Does Nord have subcellular localization?

Nord localized to both apical and basolateral side of the wing epithelium*.*

An antibody against the Dpp prodomain is used in Figure 4A. BMP proproteins are cleaved separating the prodomain from the ligand domain proper, with the ligand sometimes secreted with its prodomain and other times, not. Do the authors feel this pattern reflects that of the secreted Dpp ligand?

We could not detect any specific signal when we used anti-Dpp prodomain antibody in the absent of detergent (surface staining protocol). The Dpp signal in the original Figure 4 (Figure 5 in the revised version) should come from the Dpp prodomain that stayed inside of the producing cells.

Nord-GFP is in a subset of Dpp prodomain expression in larval wing disc (Figure 4A) and shows even less overlap in pupal wing disc (Figure 4B) – this needs to be more clearly addressed in the text.

The suggested clarification was made in the revised manuscript. See Page 13-14.

Data not shown for Nord-GFP in later pupal wing when dpp-lacZ transitions to longitudinal veins.

Nord-GFP (co-stained with anti-Dpp prodomain antibody) expression in later pupal wing is provided in Figure 5A (lower row) in the revised manuscript.

"While dpp is also known to be expressed in the developing longitudinal veins (deCelis, 1997), we see no Nord expression in either the longitudinal vein or crossvein despite the fact that Nord loss of function leads to ectopic crossvein formation." The data referred to in the underlined text top page 13 is not shown. (?)

The related Nord expression data is presented in the lower row of Figure 5A of the revised manuscript.

Figure 4E – Is quantification plot an average of multiple discs? how many discs were measured? Where was the measurement taken? Is it a profile from a line across the disc or a compilation of all signal in the disc? Would the authors expect pMad to be altered in the wing pouch?Figure 4F – not at all clear what is meant by "maximum pMad intensity in the A and P compartment" in the figure legend. Is this the highest intensity measurement in the entire disc?Is the proposal that the slight reduction in pMad in the anterior affects only growth and patterning of the position of the wing veins?

The quantification procedure was described with more detail in “Image collection and quantification of fluorescence intensity”. In the new Figure 5C, we present pMad intensity profile plot as the mean ± SD from n = 15 wing imaginal discs. In the *nord* mutants, reduced pMad signal was observed, which we thought is due to increased spreading of Dpp when Nord is absent. The increased spreading will likely cause reduced local Dpp concentration, thus reduce BMP signaling in the medial wing discs (see the proposed model in Figure 9).

If Nord is required to facilitate "Dpp" movement, then why is signaling activation (as measured by pMad) in the majority of the anterior compartment beyond the peak unchanged. If in the absence of Nord, Dpp does not move, then one would expect all signaling in the anterior to be affected.

We propose Nord-mediated binding of Dpp (and/or Dpp-Gbb) interferes with (not facilitate) Dpp movement. This is because Nord is associated with the matrix, and does not spread too far from the producing cells (Figure 3D, Figure 7D). Also, we detected much less Dpp/Dpp-Gbb ligands in the medium when Nord is co-expressed (Figure 8B). We propose that in the absence of endogenous Nord (low level), Dpp spreading will increase, which in turn lead to the reduced medial BMP signaling and associated with decreased wing growth.

Instead of facilitating movement, could Nord prevent Dpp-Gbb turnover, thus in its absence Dpp-Gbb is less stable and a drop in signaling (pMad) is observed. Difficult to understand how Nord has an effect on ligands when its not present in wing pouch cells.

Instead of “impeding (not facilitating)” Dpp movement, we agree endogenous Nord may also prevent Dpp-Gbb internalization and turnover, thus enhancing BMP signaling. However, these models need to be tested in future experiments.

Secreted Nord, although it does not travel too far, may explain its potential effect in the center wing pouch.

Top half of pg 16 – authors suggest that media from cells co-expressing Nord and Dpp/Gbb elicit less signaling (pMad) than media produced by cells without Nord -if the same amount of media is added to cells in Figure 5C then it seems that M4 media is less effective at inducing pMad compared to the lower amount of Dpp/Gbb in M5 which elicits nearly the same level of pMad.What does a normalized plot show?

We thank the reviewer for pointing out an additional interpretation of Figure 8C (original Figure 5C). In the revised manuscript, we added panel D to show the relative signaling activity of an equal volume of conditioned medium (pMad/Mad). The medium signaling activity was then normalized to the ligand amount (band intensity in panel B) to calculate the normalized ligand activity [(pMad/Mad)/ligand amount], which was found higher in the conditioned medium with Nord (Figure 8E), suggesting released Nord may enhance Dpp/Gbb activity. We expanded this observation and showed that Nord could modulate recombinant Dpp activity in a biphasic manner *in vitro* (Figure 8F-I). These *in vitro* results are consistent with what we found *in vivo* (Figures 4, 6, and 7).

Using a recombinant Dpp peptide in Figure 5D, the authors find that in the presence of Nord-GFP, equivalent amounts of rDpp induce less pMad but this is the opposite of what is observed *in vivo* in the disc where signaling is reduced in nord mutants. Perhaps the authors can expand on this observation. Is this result related to the fact that a recombinant peptide was used in rather than the entire Dpp proprotein or prodomain + ligand domain?

In Figure 8H (original Figure 5D), high levels of exogenous Nord were expressed under the actin promoter (pAcSV plasmids) in transiently transfected S2 cells, therefore, only inhibitory effects of Nord were observed here, which is similar to the inhibitory effects of high levels of Nord produced by en>3XUAS-Nord or nub>UAS-Nord at 29°C.

Figure 5E in WT – seems odd that one cannot see increased concentration of Mad-Flag in the nucleus.

This is likely due to (1) FLAG-Mad is expressed at a high level in the stable S2 cells, and (2) very low dose of Dpp (0.625 nM) was used. Thus, only a small portion (for example <5%) of total FLAG-Mad is phosphorylated, and its subcellular localization change will not be evident when visualized with the FLAG antibody.

Suggest that Nord-GFP could prevent release of ligand from cell surface and inhibit ligand from media to act on responding cell – what about ligand-receptor binding?

It is very likely that Nord functions via additional BMP interacting proteins, including the receptors. We propose that low levels of Nord enhance and high levels inhibit BMP signaling likely by promoting or interfering ligand-receptor binding either directly or indirectly, which will be tested in future studies.

Westerns in Figure 5 are cropped and it is not possible to see the other cleaved forms of ligands. Does Nord show differential interaction with different ligand forms, as has been shown by Anderson 2017 to be the case with different forms of Gbb and receptors?

In the medium used in the IP experiment, majority of the ligands, as showing in Figure 8A, are smaller than 25 kDa (see the uncropped blot in Figure 8-source data 1). We also detected a smaller fraction of other isoforms of Dpp /Gbb (*), and further studies are required to figure out the interacting domain on Dpp/Gbb and Nord.

Figure 6How were pMad profiles measured? Superimposing profiles in D-F would be helpful.Authors state ptc levels have not changed – provide measurement of ptc levels.

Suggested changes were made in the revised Figure 6. The procedures used to measure pMad and Ptc was described with more detail in “Image collection and quantification of fluorescence intensity”.

Loss of L5 and PCV are typical of gbb LOF phenotypes and not seen in dpp LOF genotypes, therefore, it is hard to reconcile these phenotypes with if Nord is thought to preferentially affect dpp.

Based on Posakony, et al., 1991 and Ray and Wharton 2001, *dpp* is required for the distal half of L5. In Figure 6 and the original Figure 7 (replaced by a new Figure 7), high levels of ectopic Nord (driven by *nub-Gal4* or *en-Gal4*) indeed only caused loss of the distal half of L5. Thus, these observations are consistent with the model that ectopic Nord impedes Dpp spreading, thus affect L5 growth due to reduced Dpp ligands available in the distal wing discs.

In Figure 6H – nub/+ label could easily be confused with a nub mutant heterozygote.A nub-Gal4/+ 29°C control wing should be shown.

The suggested changes were made and the suggested control (UAS-GFP) were added in the revised Figure 6.

In some places nord is indicated as a secreted protein and in others membrane-associated – a little bit confusing.

Nord is a secreted protein, but associated with the extracellular matrix and not freely diffusible. To avoid confusion, we now changed “membrane-associated” to “matrix-associated” in the revised version.

Akiyama 2015 showed that the pMad gradient in the wing pouch, as well as Dpp were not required for wing growth. How do the authors think changes in the pMad profile may change growth?

In Akiyama 2015, their conclusion is “These results indicate that during the latter half of larval development, the Dpp morphogen gradient emanating from the anterior–posterior compartment boundary is not directly required for wing disc growth.” There are also publications (Barrio and Milan, 2017 *eLife*) demonstrated that “Boundary Dpp promotes growth of medial and lateral regions of the *Drosophila* wing”. We propose that ectopic Nord (driven by *nub-Gal4* throughout larval development) impedes the spreading and inhibits the activity of Dpp (reflected at the level of pMad intensity), which in turn inhibits the growth of the wing.

It is curious that changes in the pMad gradient seen in Figure 6F has no effect on longitudinal vein development in Figure 6K.

Based on Posakony, et al., 1991 and Ray and Wharton 2001, *dpp* is required for the distal half of L5. In the original Figures 6 and 7 (replaced by a new Figure 7 to void the L5 phenotype), high levels of ectopic Nord (driven by *nub-Gal4* or *en-Gal4*) indeed only caused loss of the distal half of L5. Thus, these observations are consistent with the model that ectopic Nord impedes Dpp spreading, thus affecting L5 growth due to reduced Dpp ligands available in the distal wing discs.

The Discussion contains a considerable amount of background information that is not particularly relevant to the findings presented in manuscript. Focusing the discussion a bit more would be helpful to the reader.

As suggested, we shortened the discussion.

To the reader it is not at all clear how expression in the posterior compartment during larval period influences PCV formation in the pupal wing when ligands are thought to be transported from the longitudinal veins.

We propose the ectopic vein formation in the pupal wing is related to Nord expression in the early pupal wing discs (Figure 5A lower row; Figure 5—figure supplement 1), not that in the larval wing disc. The text was extensively reorganized to better communicate this message. See Pages 13-22.

Does uniform expression in posterior using enGal4 interfer with the normal mechanism?

In the original Figure 7, we showed en-Gal4 driven expression of Nord during the larval stage and pupal stage caused reduced wing size, especially the posterior compartment size.

However, en-Gal4 driven expression of Nord during the pupal stage has no noticeable effect on wing size (Figure 7 of the revised version).

Suppl Figure 12 – orientation of high mag of trichomes is incorrect

The orientation of the zoomed area was corrected in Supplementary file 4 of the revised version.

Overall, please check labelling of figure for units, number of samples (n=?), and accuracy of figure legend description. In some places in the manuscript this information is robust but in other places it is lacking.

Suggested changes were made in the revised version.

Reviewer #3:The study focuses on a possible role for the secreted Nord protein in BMP signaling in *Drosophila*. The authors demonstrate Nord binding to Dpp and Gbb *in vitro*, and suggest that this explains the *in vivo* phenotypes observed. The *in vitro* work is quite suggestive, but some caution may be in order as other other extracellular molecules can affect multiple pathways, and some of these other pathways can have have similar effects on growth and vein formation. While this is a worthwhile publication with strong *in vitro* data, I have a few points below that I think would strengthen the *in vivo* data.

We thank the reviewer for the instructive comments and suggestions. As suggested, we have performed additional experiments and analysis to strengthen the *in vivo* data and clarify the model of how Nord functions.

1) Nord as a Hh target- The expression of Nord in the imaginal and pupal wing tissues largely mirrors that of Hh targets, being heightened anterior to the anterior-posterior compartments boundary. Its expression is however excluded from the distal wing blade, something that the authors do not further investigate. The authors can widen or narrow Nord expression by widening or reducing Hh signaling, but this will also widen BMP expression and signaling, so it would have been helpful to have one experiment showing a cell autonomous effect of Hh signal reception on Nord expression.

The suggested experiment was performed and included in Figure 2C-H in the revised

manuscript. Briefly, we found that Nord expression is cell-autonomously induced/inhibited by Hh signaling within the anterior compartment in the wing discs’ hinge and outer wing pouch region.

2) Nord mutants- The authors generate and examine Nord mutants. The effects in the wing are fairly mild, consisting of very slight undergrowth mirrored by slightly reduced BMP signaling (pMad) in the disc, and a few ectopic veins, suggestive of gains in BMP signaling. The only other growth-inducing pathways the authors examine are Hh, looking at a single target, Ptc. They do not however quantify Ptc staining in the same way they did to detect the weak effects on pMad, nor do they look at other Hh targets with different sensitivities.

In the revised manuscript, we measured the Ptc staining profile and examined the expression of a low-threshold Hh signaling target gene *dpp* (anti-Dpp Prodomain) in the *nord* mutant wing discs. No noticeable change of the Ptc or Dpp (anti-Dpp prodomain) expression levels were detected in the *nord* mutant wing discs. These data are presented in Figure 5 and Figure 5-figure supplement 2 in the revised manuscript.

Additionally, we measured the distance between distal ends of longitudinal veins L3 and L4 (dL3-L4) and that of L2 and L5 (dL2-L5), we found no specific reduction in the L3-L4 region indicated by comparable or higher dL3-L4 /dL2-L5 ratio in the female and male *nord* mutant wings (Figure 4B, C), which is consistent with the obviously reduced BMP, but not Hh, signaling activity during wing imaginal disc development (Figure 5B, C). Together, these findings suggested a positive role of endogenous Nord in augmenting BMP signaling activity to promote wing growth. See Page 14-15.

The ectopic veins in mutants are consistent with defects seen with mild increases in BMP signaling, but there are many other ways of inducing similar ectopic venation, especially through EGF gains or reductions in Notch or its ligands. The authors concentrate almost exclusively on one defect, a vein extension distal to the PCV. This is a common defect seen in many mutant fly stocks as it is something of a "hot-spot" for ectopic venation. The authors show that pMad is high in this ectopic vein, but do this staining at 26hr AP, a fairly late stage of PCV development. By this time LVs and cells near the ACV have all been expressing dpp for some time and have BMP signaling, while dpp expression in the PCV is initiating. I would wager this ectopic vein is expressing dpp, and will also have rhomboid and thus EGFR activity and high Δ and Serrate. The authors could greatly strengthen their case by examining pMad at earlier stages of PCV development (20 hr AP) before EGFR and Notch signaling are active in this region.

The suggested experiment was performed and included in Figure 5D-E in the revised manuscript. Expansion of pMad range around the primordial PCV was noticed in the *nord* mutants starting from 19-20 hr AP.

The authors also do not test whether the posterior defects they see are caused by loss of anterior Nord, or whether they might be caused by loss of low-level Nord expression in the posterior. Since they have a Nord-RNAi line, what happens with anterior knockdown? Posterior?

Our findings show that *nord*, like *ptc* and *dpp,* is a target gene of the Hh signaling pathway and not expressed in the posterior compartment of the wing discs (Figures 1, 2; Figure 2-figure supplement 3). Consistently, when we knocked down *nord* in the posterior compartment using *hh-Gal4*, we did not observe the ectopic PCV phenotype, which is frequently caused by knocking down *nord* throughout the wing discs via *MS1096-Gal4* or *A9-Gal4*. These data are presented in Figure 5—figure supplements 3 and 4 in the revised manuscript.

3) Nord overexpression and biphasic effects on BMP signaling – Increased Nord expression causes undergrowth and reduced pMad but ectopic venation. In that sense, Nord has biphasic effects, with similar mixed effects (both BMP losses and gains) caused by either too much or too little Nord. The authors extend this by showing that further increases in Nord overexpression change the ectopic PCV extension into PCV loss, the latter often caused by loss of BMP signaling. However, this loss may not always be due to a direct effect on the PCV. While the nub-gal4 example looks ok, the en-gal4 example has lost L5 adjacent to the PCV, likely due to loss of the distal L5 primordium at the disc stages (Ray and Wharton, Bangi and Wharton). That will in turn block or reduce dpp expression in L5 and thus Dpp's availability for signaling in the early PCV.

High levels of ectopic Nord driven by *nub-Gal4*, *en-Gal4* or *hh-Gal4* in the larval stage can sequester BMP ligands and inhibit BMP signaling, thus leading to loss of L5 (Figure 6; original Figure 7). To avoid affecting L5 development, we restricted ectopic Nord expression to the pupal wing and then studied its effect on PCV patterning. The temporal control of *en-Gal4/hh-Gal4>UAS-Nord* was achieved by adding the *tub-Gal80ts* allele. In this new setting, in the presence of normal L5, ectopic Nord caused PCV phenotypes in a dosage-dependent manner.

These data are presented in a new Figure 7 in the revised manuscript.

4) *in vitro* studies- These provide the strongest proof that Nord can affect BMP signaling, although in this case the observed effects are all inhibitory. Secreted Nord can co-IP with secreted Gbb or Dpp, and co-expression with Nord reduces Gbb or Dpp secretion and thus its ability to signal. Nord can also reduce the BMP signaling induced by addition of exogenous Dpp. The only shortcoming here is that the authors have not apparently tested binding to any of the extracellular modulators of BMP signaling.

Interaction of Nord and the BMP binding protein Dally was presented in a manuscript co-submitting with ours. We discussed the possibility that Nord can bind both BMPs and Dally in regulating wing growth and vein patterning in the Discussion section under “Nord is a novel, multi-functional BMP binding protein.”

I need to see whether there is ectopic pMad in the nord mutant PCV at a stage before dpp, rhomboid and Δ/Serrate expression in the region. Otherwise I am not convinced that the ectopic vein is initiated by ectopic BMP signaling.

The suggested experiment was performed and included in Figure 5D-E in the revised manuscript. Expansion of pMad range around the primordial PCV was noticed in the *nord* mutants starting from 19-20 hr AP.

Quantify Ptc or some other Hh marker (Dpp would be especially helpful) in Nord mutants.

In the revised manuscript, we have measured the Ptc staining profile and examined the expression of a low-threshold Hh signaling target gene *dpp* (anti-Dpp Prodomain) in the *nord* mutant wing discs. No noticeable change of the Ptc or Dpp (anti-Dpp prodomain) expression levels were detected in the *nord* mutant wing discs. These data are presented in Figure 5 and Figure 5—figure supplement 2 in the revised manuscript.

How much of the PCV loss in en>Nord might be caused by loss of the adjacent L5?

High levels of ectopic Nord driven by *en-Gal4* or *hh-Gal4* in the larval stage can sequester BMP ligands and inhibit BMP signaling, thus lead to loss of L5 (original Figure 7). To avoid affecting L5 development, we restricted ectopic Nord expression to the pupal wing and then studied its effect on PCV patterning. The temporal control of *en-Gal4/hh-Gal4>UAS-Nord* was achieved by adding the tub-Gal80ts allele. In this new setting, despite essentially normal L5, ectopic Nord caused PCV phenotypes in a dosage-dependent manner. These data are presented in Figure 7 in the revised manuscript.

Cell-autonomous effects of Hh signaling on Nord expression would be helpful, as would the effects of anterior vs. posterior Nord RNAi.

The suggested experiment was performed and included in Figure 2C-H in the revised manuscript. Briefly, we found that Nord expression is cell-autonomously induced/inhibited by Hh signaling within the anterior compartment in the wing discs’ hinge and outer wing pouch region.

[Editors’ note: what follows is the authors’ response to the second round of review.]

Essential revisions:1) Soften their conclusions about their *in vitro* work (Figure 8) in the results and discussion. Add a statement to the effect that without additional replicates in Figure 8 other conclusions are possible.

We include additional replicates and statistical analysis in Figure 8G. See changes in the Figure 8 legend on Page 45. We also softened the conclusion on Page 22 of the revised version.

2) Provide additional data on the Nord in the posterior compartment using ptc-gal4, UAS-nord-RNAi.

Suggested experiments were performed and included in Figure 5—figure supplements 3 and 4.

3) Add information about the cross vein phenotype of MS^-1^096 to the manuscript.

As suggested by the reviewer, we removed MS1096-Gal4 related data in Figure 4.

4) Make changes to the Figure fonts.

Suggested changes (see below) to the figure founts were made in Figures 1, 2, 4, and 7.

5) Correct the typos and awkward sentences.

Suggested changes (see below) were made.

Reviewer #1:The authors have made strong efforts to respond to the essential revisions and the reviewers' critiques. Overall, the manuscript is much improved. However, I have concerns about the conclusions regarding Figure 8, the *in vitro* work, which the authors use conclude that Nord *in vitro* also has biphasic function on BMPs *in vitro*. The authors present results from a single experiment and then normalize these single data points to draw conclusions about Nord function *in vitro*. Specifically, the authors' new analyses in Figure 8 panels D,E,G show relative pMad levels normalized to Mad (8D,8G) and to Mad and BMP (8E). My concerns are as follows. First, the authors need statistical analyses. Second, BMP released from cells appear to be equally active in the absence (M4) or presence (ME) of Nord, and these data only become supportive of their argument when "normalized" to the amount of ligand. Third, there are additional bands consistent with other Dpp/Gbb isoform in the source blot for Figure 8A. These may be bioactive molecules but are not accounted for in the manuscript. The pMad bands in M4 and M5 also appear to be saturated, again making it difficult to determine differences between the lanes. In panel C, the WB: Flag row is "smiling" with some lanes compacted and others not, again making it difficult to normalize. Without more replicates, the authors statement that "soluble ligand appears to have a higher signaling activity perhaps as a result of an association with Nord" (p.21) is not sufficiently supported by the data. This same issue holds for panel F, with only one replicate. In its current form, the data are not conclusive and other interpretations are also possible. The authors should soften their conclusions about their *in vitro* work in the results and discussion. Additionally, they need add a statement to the effect that without additional replicates in Figure 8 other conclusions are possible.

We want to thank the reviewer for carefully reviewing our manuscript. We agree with the concerns regarding panels B-E, and therefore added the experiments in F and G, where rDpp was used. In the revised version, we include additional replicates and statistical analysis in Figure 8G. See changes in the Figure 8 legend on Page 45. We also softened the conclusion on Page 22.

They are other issues that need to be addressed:1. The figure in the response to reviewers was too small. In the future, please upload a larger image.

Suggestions will be followed in the future.

2. P. 7. The authors write "In wing discs, cells near the A/P compartment boundary (B: ptc+) receive the highest level of Hh stimulation while A cells (A: hh-), located further from the border, receive lower levels of stimulation. In contrast, P cells (P: hH^+^) do not respond to Hh due to lack of the receptor Ptc and the transcription factor Ci (Tabata et al., 1992; Eaton and Kornberg, 1990)." It is very important that the authors add "Figure 1" when discussing B, A and P cells.

Suggested changes were made on Page 7 of the revised manuscript.

3. P. 11 and Figure 3C – please clarity what tissue is being used for the flip-out clones.

Suggested changes were made on Page 11 (Figure 3D) of the revised manuscript.

4. Top of P. 15. Please rephrase "which is consistent with the obviously reduced BMP, but not Hh, signaling activity during wing imaginal disc development (Figure 5B, C)" because it is not clear what is being discussed. I suggest "which is consistent with normal Hh signaling activity during the development of nord mutant wing imaginal discs (Figure 5B, C)."

Suggested changes were made on Page 15 of the revised manuscript.

5. P. 18 The authors cite Robert 1998 but there is no such citation in the references cited section. Please add this reference or remove it from the manuscript.

The missing citation was added to the reference list.

6. P. 21. As noted, please soften your conclusions about the *in vitro* studies and add a statement to the effect that without additional replicates in Figure 8 other conclusions are possible.

We include additional replicates and statistical analysis in Figure 8G. See changes in the Figure 8 legend on Page 45. We also softened the conclusion on Page 22 of the revised version.

7. P. 25, line 9 "it's" should be "its".

Suggested changes were made on Page 25 of the revised manuscript.

8. P. 25, line 13 "Akiyama et al. 2021 accompanying manuscript" is not in the references cited.

The missing citation was added to the reference list.

9. Add to the Materials and methods the procedure you used for normalization of the data in Figure 8E,D,G.

The procedure used for normalization of the data in Figure 8E, D, G was added to the figure legend (Page 45-46) and Materials and methods (Page 37).

10. Figure 1B – increase the font size of the colored/highlighted genes. Currently we can't read them.

The suggested change was made in Figure 1B.

11. Figure 2A – increase the font size of the RMCE vectors. Currently we can't read them.12. Figure 2D, E, F, G, H – increase the width of the lines outlining regions of the wing disc and clones. Currently we can't see them.13. Figure 2C-H, it would be helpful to add C' and C" etc to the unlabeled panels.

The suggested changes were made in Figure 2, as well as the related figure legend.

14. Figure 4E – The graphs are too small – I cannot read anything. Please increase the size of the graphs and the font of the labels.

The suggested change was made in Figure 4E.

15. Figure 5D – increase the width of the lines of the boxes and the enlargements so that they can be seen on a print out.

The suggested change was made in Figure 5D.

16. Figure 7B – increase the width of the lines of the boxes and the enlargements so that they can be seen on a print out.

The suggested change was made in Figure 7B.

17. Figure 2, figure supplement 3. Increase the size of the brackets in the Nord-GFP monochrome panels. Currently, it is very hard to see them.

The suggested change was made in Figure 2—figure supplement 3.

18. Figure 5—figure supplement 2B – label the blue and the red lines.

The suggested change was made in Figure 5—figure supplement 2B.

Reviewer #2:The authors have added some valuable data that have improved the manuscript, enough for acceptance. I do however have some points I would like the authors to seriously consider. (1) and (2) concern data that are problematic, but could be removed from the manuscript without damaging its conclusions.

We want to thank the reviewer for carefully reviewing our manuscript. We have seriously considered and addressed each concern. Below is a point-by-point response.

1) Hh signaling and Nord expression- The autonomous gains in Nord expression caused by Smo overexpression are convincing. The loss caused by Ptc overexpression, shown for a single flpout clone Figure 3D, is not. To be a valid result, the dotted blue line drawn by the authors must accurately follow the A/P. However, outside the Ptc clone at the bottom of the figure, the dotted line does not at all correspond to the line shown by Nord expression, even though the Nord outside the clone should define the A/P. Nor does the line correspond to the A/P defined by Ci staining. Instead, the true A/P boundary is further left, and thus it is not at all clear that the Ptc clone is in the anterior compartment. If the authors want to keep Ptc flpout data in the manuscript, they need a more convincing example.

We agree with the reviewer’s concern and have removed the related data from Figure 4 and Figure 5—figure supplementary 3. We also removed related text, including the statement "further validates that the observed phenotypes in the nord mutants are not indirect effects associated with loss-of-nord in tissues other than the wing discs" from Page 13 and 15.

2) MS1096-gal4- The authors use MS1096 for some of the nord-RNAi data. This is a Gal4 enhancer trap insertion into the X-linked Bx (dLMO) locus, and in hemizygous males has a mild Bx wing mutant phenotype with variable penetrance that includes disruption of the crossveins (see Figure 6 in Milan, Diaz-Benjumea, and Cohen 1998). The crossvein phenotype tends to disappear in stocks, but often reappears when outcrossing to other lines. It is therefore not an appropriate driver to use when assessing crossvein phenotypes in males. If the authors want to use it, they have to explain how they dealt with the crossvein phenotypes that can be caused by MS1096 on its own.And while MS1096 is expressed in the wing disc, it is not clear that MS1096 is "wing-specific" as the authors state. Bx (dLMO) is expressed in many tissues, including the CNS. If MS1096 follows the expression of Bx, the authors cannot say that this experiment "further validates that the observed phenotypes in the nord mutants are not indirect effects associated with loss-of-nord in tissues other than the wing discs."

We agree with the reviewer’s concern and have removed the related data from Figure 4 and Figure 5—figure supplementary 3. We also removed related text, including the statement "further validates that the observed phenotypes in the nord mutants are not indirect effects associated with loss-of-nord in tissues other than the wing discs" from Page 13 and 15.

3) Is Nord in the posterior compartment? The authors make several absolute statements about the absence of Nord expression from the posterior of the developing wing, but do not provide any staining of controls that would allow one to distinguish signal from background. While Nord is certainly much higher anterior to the A/P boundary, the data does not rule out low-level Nord expression in the posterior. The authors have addressed this in their revision, not by changing any of their absolute statements about expression, but by added a new experiment showing that "we did not notice any ectopic PCV in the adult flies when UAS-nord-RNAi was selectively expressed in the P compartment of the larval and pupal wing via the hh-Gal4 driver." But this negative result could simply reflect stronger RNAi expression with MS1096-gal4 or A9-gal4. It would be preferable if they had tested the effects of nord-RNAi when it was limited to the anterior compartment, something that their model predicts should affect the posterior crossvein. This could easily be done with ptc-gal4 or ci-gal4.

As suggested, we analyzed the ectopic venation phenotype in flies carrying ptc-Gal4 driven expression of UAS-nord-RNAi (Figure 5—figure supplements 3 and 4). See the related changes on Page 15-16 of the revised manuscript.

The authors also model the action of Nord as if it only diffuses over a short distance, and thus is only affecting BMP movement from neighboring cells, such as those in L4 in pupal wings, and is never present in the PCV. However, the extracellular staining of Nord-GFP in Figure 3 looks substantially above background throughout the wing disc.

We thank the reviewer for pointing out our incomplete interpretation regarding how far Nord may diffuse. We propose that the secreted Nord/NDNF proteins likely exit in two spatially distinct pools: one pool with more freely diffusible proteins that can reach a longer distance, whereas another pool containing proteins mainly associated with source cells or nearby extracellular matrix. See Pages 23-34 and 47 of the revised manuscript for related changes.

4) Biphasic or multiphasic effects- The authors have added new data that clarify some of the effects of Nord in the PCV, but complicate their story about biphasic effects. In growth Nord appears truly biphasic, promoting growth only at normal levels, and inhibiting growth and BMP signaling with either decreased or increased Nord expression. But in PCV formation things are more complex. Both loss of endogenous Nord and mild gains induce ectopic PCV phenotypes. Since normal levels do not alter the PCV, that also fits with a biphasic effect, albeit the opposite of the growth effect (ectopic signaling with both losses and mild gains). Overexpression also has a biphasic effect, as strong Nord gains do the opposite of weak gains and cause loss of the PCV and signaling. However, if one puts all of this together, the PCV effect is more multiphasic than biphasic, because of the normal signaling observed at wild type. Nord loss increases signaling, wild type is normal, mild Nord overexpression increases, strong Nord decreases.

We agree with the reviewer that we need to be cautious about using the word “biphasic” when describing how Nord modulates BMP signaling in PCV patterning. Therefore, we have replaced “biphasic” with “dosage-dependent” when describing the activity of Nord in the context of PCV patterning.

The only detailed explanation of these effects is in the legend of Figure 9, which is not very complete and leaves some puzzles. If loss of Nord increases long-range BMP diffusion into the PCV, why isn't the same thing happening in a third instar wing pouch?

Loss of nord likely also leads to increased Dpp spreading in the 3rd instar wing pouch. For instance, a recent study reported that nord mutation rescued the L5 defects in dally mutants (Akiyama et al., 2021). However, in nord single mutant, the Dpp spreading increase associated gain of BMP activity is not strong enough to overturn the reduced BMP signaling associated with medial Dpp reduction during wing growth regulation.

And there are ideas hidden in the figure itself that are not explained in the legend or the Discussion, especially the switch from BMP presentation to BMP sequestration with a change from low local to high local Nord concentrations. A fuller discussion in the text would be helpful.

The suggested changes were made on Pages 23-34 and 47 of the revised manuscript.

5) Biphasic *in vitro*- The *in vitro* work remains strong, especially for Nord's inhibitory effects on BMP signaling. But the only evidence for a positive and thus biphasic effect of adding Nord relies on a single lane in a single gel: the lowest Nord concentration in Figure 8F, then quantified in 8G. It would be helpful if the authors could state how many times they have repeated this result.

We include additional replicates and statistical analysis in Figure 8G. See Page 45 for the changes in the legend of Figure 8. We also softened the conclusion on Page 22.

6) Given the long discussion of NDNF, the authors should mention the mechanisms that have been proposed for NDNF activity: by binding integrins, or by modulation of FGF signaling by binding HSPGs.

The suggested changes were made on Page 28.